

# Combination of UAV and terrestrial photogrammetry to assess rapid glacier evolution and conditions of glacier hazards

Fugazza, Davide[1]; Scaioni, Marco[2]; Corti, Manuel[2]; D'Agata, Carlo[3]; Azzoni, Roberto Sergio[3]; Cernuschi, Massimo[4]; Smiraglia, Claudio[1]; Diolaiuti, Guglielmina Adele[3]

[1]Department of Earth Sciences 'A.Desio', Università degli studi di Milano, 20133 Milano Italy

[2]Department of Architecture, Built Environment and Construction Engineering, Politecnico di Milano, 20133 Milano Italy

[3]Department of Environmental science and policy (DESP), Università degli studi di Milano, 20133 Milano Italy

[4]Agricola 2000 S.C.P.A., 20067 Tribiano (MI) Italy

*Correspondence to:* Marco Scaioni (marco.scaioni@polimi.it)

**Abstract**

Tourists and hikers visiting glaciers all year round face hazards such as the rapid formation of collapses at the terminus, typical of such a dynamically evolving environment. In this study, we analysed potential hazards of the Forni glacier, an important geo-site located in Stelvio Park (Italian Alps), by describing local surface features and evaluating the glacier melting rate. The analyses were based on point clouds and digital elevation models (DEMs) from two separate surveys of the glacier tongue carried out in 2014 and 2016 with Unmanned Aerial Vehicles (UAVs), terrestrial photogrammetry (only in 2016) and a DEM obtained in 2007 from an aerial survey. On the area covered by the 2016 survey, average glacier thinning rates of -4.15 ma$^{-1}$ were found in 2007-2016, while the mean thickness change of the glacier tongue in 2014-2016 was -10.40±2.60 m. UAV-based DEMs were thus found to be sufficiently accurate with respect to the rates of glacier down-wasting, while terrestrial photogrammetry allowed the reconstruction of the glacier terminus, presenting several vertical and sub-vertical surfaces whose modelling was difficult to obtain from airborne UAV images. The integration of UAV and terrestrial photogrammetry provided a detailed and accurate 3D model of the glacier tongue, which we used to identify hazard areas.

## 1 Introduction

The effects of climate change due to global warming are increasingly seen on high mountain regions. In the European Alps, temperatures have increased twice the global average over the last century (Auer et al., 2007; Brunetti et al., 2009). Precipitation patterns show contrasting local trends, with an increase in the northern Alps and a decrease on the southern side (Brunetti et al., 2009), while snow cover has reportedly decreased in the last three decades (Bocchiola and Diolaiuti, 2010; Diolaiuti et al., 2012). The



most sensitive indicators of climate change in mountain regions are glaciers and permafrost, both
showing unequivocal signs of involution. In the Italian Alps, glaciers have lost at least about a third of
their area since the 1950s (Smiraglia et al., 2015). A similar retreat has occurred in the Swiss Alps, where
Fischer et al. (2014) report a loss of 28% since 1973, and in the French Alps, with a decrease in glacier
area of 25% since the early 1970s (Gardent et al., 2014). Warming trends have also been reported at
permafrost monitoring sites throughout Europe, with consequent thickening of the active layer (Harris et
al., 2009).
Changes to glacier and permafrost environments, either by climate variations alone or in combination
with anthropogenic activities, have been recognized to promote land-surface instabilities, playing a
significant role in the generation of geomorphological hazards evolving in a downstream direction
(Keiler et al., 2010). In glacial and periglacial regions, the most severe hazards are generally related to
flooding, through the outburst of moraine- or ice-dammed lakes. Climate change has accelerated the
formation of glacial lakes and the expansion of new ones, increasing the risk of devastating glacial lake
outburst floods (GLOFs), which frequently occur in the Himalayas, Karakorum, Chilean Patagonia and
Peruvian Andes (Wang et al., 2015). In recent years, the formation of moraine-dammed lakes has also
been reported in the Swiss Alps, with growing concern of possible overtopping of moraine dams
provoked by ice avalanches (Gobiet et al., 2014). Outbursts of water from the englacial or subglacial
system are equally threatening: in the French Alps, water-filled cavities were recently identified at
Glacier de Tête Rousse, which experienced a deadly rupture of a water pocket in the past (Garambois et
al., 2016). Other recurrent hazard situations may arise from ice avalanches from hanging glaciers
(Vincent et al., 2015), including the complete detachment of sections of the ice body. In Italy, the partial
detachment and fragmentation of the Mount San Matteo serac in Stelvio Park limited spring access to
the Forni Glacier for skiers and mountaineers in 2005 and 2006 (Riccardi et al., 2010). More recently,



Azzoni et al. (submitted) identified two types of collapse features (see fig. 2) on the tongue of Forni
Glacier, namely normal faults and ring faults, both posing serious hazards to mountaineers. The first
occur mainly on the medial moraines and are due to gravitational collapse of debris-laden slopes, whereas
the latter develop as a series of circular or semicircular fractures with stepwise subsidence, caused by
englacial or subglacial meltwater creating voids at the ice-bedrock interface and eventually the collapse
of cavity roofs. The retreat and thinning of glaciers in the Alps, while increasing the likeliness of these
collapses, is also a major cause of slope instabilities in combination with permafrost thawing, uncovering
and debuttressing rock and debris flanks, increasing mass movement and potentially triggering landslides
and rock avalanches (Keiler et al., 2010).
**1.1 Remote sensing of glacier hazards**
The highly dynamic nature of high mountain environments has led to a widespread use of optical remote
sensing for monitoring of glacier-related hazards, with the ability to produce digital elevation models
(DEMs) and evaluate changes on the basis of multispectral images. DEMs are particularly useful to detect
glacial thickness and volume variations and to identify steep areas that are most prone to
geomorphodynamic changes such as mass movements (Blasone et al., 2014). Multispectral images at a
sufficient spatial resolution enable the recognition of most glacial- and permafrost-related hazards,
including glacier lakes and landslides, their geometric properties and kinematics (Kaab et al., 2005).
Indeed, the crucial factors for monitoring of hazard events, which might be localized in small glacial and
periglacial areas and evolve over short-time scales, are the revisit time of the sensor and its spatial
resolution. In practice, sensors with a high-frequency revisit time often have a coarse spatial resolution
(e.g., MODIS), while images from high-resolution optical sensors are costly and with restrictive data
access policies (e.g., Pleiades, Worldview). This issue mostly limits data availability to the Landsat
TM/OLI family of sensors and Terra ASTER with a maximum spatial resolution of 15 m. Although



technological improvements have been made with Sentinel-2, with greater spatial and temporal coverage
and finer spectral resolution, cloud cover is still a major issue affecting satellite optical sensors and
limiting the acquisition of information over an area of interest. In very recent years, the application of
imaging sensors carried by unmanned aerial vehicles (UAVs – Colomina & Molina, 2014, O'Connor et
al., 2017) has started to emerge in the glaciological community as a viable low-cost alternative for multi-
temporal monitoring of small areas, effectively enabling on-demand research and bridging the gap
between field observations, notoriously difficult on glaciers, and coarser resolution satellite data
(Bhardwaj et al., 2016a).
The use of UAV-based remote sensing for glacier research started in polar environments, in small-scale
studies of cryoconite holes (Hodson et al., 2007), and melt ponds (Inoue et al., 2008). During the last
decade, UAV photogrammetry (Remondino et al., 2011) has been slowly gaining pace as a tool for the
generation of high-resolution DEMs (see, e.g., Rippin et al., 2015). Few studies however have explored
the potential of UAVs in high mountain environments, likely due to the following issues:

1.   The reduced operating autonomy due to the limited battery support combined with the effects of

lower air pressure and temperature;

2.   The complexity of mountainous terrain, which may make it difficult to find suitable take-off and

landing sites; and

3.   Potential Problems in the visibility of GNSS (Global Navigation Satellite System) satellites,

which can hamper UAV navigation (Bhardwaj et al., 2016a) and may introduce errors in geo-

referencing (Santise, 2016).




Notable exceptions include the works of Immerzeel et al. (2014), who generated a high-resolution
orthophoto and DEM to study the dynamics of Lirung Glacier (Nepalese Himalaya) and of Fugazza et
al. (2015) in their study of an Alpine glacier. The latter authors produced an orthophoto from a UAV
survey and mapped small- and large-scale supraglacial features of the Forni Glacier (Italian Alps),
including debris cover, crevasses, epiglacial lakes and the medial moraines, via object-based image
analysis (Blaschke, 2010). In Dell'Asta et al. (2017), multiple orthophotos and DEMs were created from
UAV data captured over the Gran Sometta rock glacier (Italian Alps); a semi-global matching technique
for comparing time-series of both types of raster data was developed in order to detect the surface
displacement field. Another technique that has been shown to provide sufficiently accurate point clouds
for studying glacier surfaces is terrestrial photogrammetry, although a necessary requirement in this case
is that the region of interest must completely observed from ground stations (see, e.g., Piermattei et al.,
2015; 2016). An overview of state-of-the-art terrestrial photogrammetry for application in geosciences
can be found in James & Robson (2012), Westoby et al. (2012), Smith et al. (2015), and Eltner et al.

(2016).

In spite of these progresses, an intercomparison of UAV and terrestrial photogrammetry and accuracy
evaluation of point clouds is still lacking in glacial environments. While Gindraux et al. (2016) estimated
the optimal density of GCPs collected with GNSS sensors to produce accurate DEMs from UAV surveys,
comparison against consolidated surveying techniques such as LiDAR (Bhardwaj et al., 2016b) and
theodolite measurements is still missing over glaciers.
In this study, we focused on a rapidly evolving, hazard-prone glacier in a protected area of the Italian
Alps. We compared different platforms and techniques for point cloud, DEM and orthomosaic
generation: UAV photogrammetry (from two distinct aircraft), terrestrial (or close-range)
photogrammetry (Luhmann et al. 2014) and terrestrial laser scanning (TLS - Vosselman & Maas, 2010),



with the aim of: (1) evaluating the accuracy of UAV- and terrestrial photogrammetric products; (2)
investigating ice thickness changes on both long and short-time scales; (3) identifying glacier-related
hazards, particularly the ones representing acute hazardous phenomena posing risk for mountaineers
visiting the glacier during summer.
**1.2 Study Area**
The Forni Glacier (see Fig. 1a, b), in the Ortles-Cevedale group, was the largest Italian valley glacier
(Smiraglia et al., 2015) until 2015, when the easternmost part of its three ice tongues separated from its
accumulation basin. The latest Italian Glacier Inventory (based on 2007 data, i.e., before the separation),
reported the total glacier area as 11.34 km$^2$ (Smiraglia et al., 2015), an altitudinal range between 2501
and 3673 m a.s.l. and a North-North-Westerly aspect. The glacier has retreated markedly since the little
ice age (LIA), when its area was 17.80 km$^2$ (Diolaiuti & Smiraglia, 2010), with an acceleration of the
retreating trend in the last three decades (Diolaiuti et al, 2012, D'Agata et al; 2014). It gained scientific
importance in 2005, when it was chosen as the site of the first Italian supraglacial automatic weather
station (AWS1 Forni, see Citterio et al; 2007), included in the SPICE (Solid Precipitation Inter
Comparison Experiment) and CryoNet networks of the WMO (World Meteorological Organization).
Recent research on this glacier mainly focused on the modeling of the albedo and debris cover via
terrestrial photography (Azzoni et al., 2016), satellite remote sensing (Fugazza et al., 2016), and a UAV
survey (Fugazza et al., 2015). Beside its scientific relevance, the main reasons behind the choice of this
glacier as a study area are:
1.  The significant retreat of the glacier since the LIA, which sets it as an example of the evolution

of valley glaciers in the Alps;

2.  The profound changes in glacier dynamics that have taken place in recent years, including the

loss of ice flow from the eastern accumulation basin towards its tongue and the evidence of



collapsing areas on the eastern tongue (Azzoni et al., submitted). One such area, hosting a large
ring fault (see Fig. 2d) prompted an investigation carried out with Ground Penetrating Radar
(GPR) in October 2015, but little evidence of a meltwater pocket was found under the ice surface
(Fioletti et al., 2016). Since then, a new ring fault appeared on the central tongue, and the terminus
underwent substantial collapse (see Fig. 2a,b,c,e);
3. The touristic and mountaineering importance of the site (Garavaglia et al., 2012). In fact, the
glacier is included in the list of geosites of Lombardy region (see Diolaiuti and Smiraglia, 2010)
and it is located in Stelvio Park, one of Italy's major protected areas. The glacier is frequently
visited during both winter and summer months, often by inexperienced hikers unaware of the
hazards posed by crevasses and collapsing areas.
**2 Data Sources: acquisition and processing**
**2.1 2016 surveys**
At the end of August 2016, a data acquisition campaign was carried out with the specific aim of
reconstructing the glacier tongue of the Forni Glacier. Multiple techniques were adopted and integrated,
to evaluate the performances of different approaches and establish a methodology for future repeat
surveys. A UAV-photogrammetric survey with a quadcopter (see Sec. 2.1.1) was conducted to provide
a DEM of the glacier surface, to be compared with other DEMs dating back to 2007 and 2014. A
photogrammetric survey carried out from ground stations (Sec. 2.1.2) was specifically aimed at
reconstructing the glacier terminus. In order to assess the quality of the photogrammetric point clouds, a
terrestrial laser scanning (TLS) survey of the same area was concurrently conducted (Sec. 2.1.3). In
addition, a set of ground control points (GCPs) was measured with GNSS equipment in order to register
all the previous point clouds into the mapping frame (Sec. 2.1.4).
**2.1.1 UAV Photogrammetry**





The UAV survey took place on two separate days, on 30$^{th}$ August and 1$^{st}$ September 2016, during the
central hours of the day, as weather conditions on the glacier were rather unstable (rain, excessive cloud
cover) and did not allow morning operation or surveying the glacier on consecutive days. Both surveys
were carried out under low cloud cover to avoid direct solar radiation on the glacier surface while
preserving diffuse illumination conditions (Pepe et al., 2017, submitted). The UAV employed in this
survey was a customized quadcopter (see Fig. 3b, Table 1) carrying a Canon Powershot 16 Megapixel
digital camera. During experiments prior to the flights on the glacier tongue, it was noticed that the
quadcopter drew a significant amount of power for vertical ascension and that it was overly sensitive to
vibrations during flight, potentially exposing pictures to motion blur. To deal with the first issue, two
different sites were chosen for taking-off and landing. Both places, at elevations above the glacier surface,
permitted to gain altitude before take-off and maintain line-of-sight operation with flights at low relative
altitude of 50 m, which ensured an average ground sample distance (GSD) of 5.7 cm. The first take-off
site was on the eastern lateral moraine (elevation approx. 2700 m a.s.l.), while the second site was a rock
outcrop on the hydrographic left flank of the glacier (see Fig. 1b) at an elevation of approx. 2750 m a.s.l.
To reduce motion blur, camera shutter speed was set to the lowest possible setting, 1/2000 s, with aperture
at F/2.7 and sensitivity at 200 ISO.
Several individual parallel flights were conducted to cover a small section of the proglacial plain and
different surface types on the glacier surface, including the terminus, a collapsed area on the central
tongue, the eastern medial moraine and some debris-covered parts of the eastern tongue. A 'zig-zag'
flying scheme was followed to reduce the flight time. The UAV was flown in autopilot mode using the
open-source software Mission Planner (Oborne, 2013) to ensure 70% along-strip overlap and sidelap. In
total, two flights were performed during the first survey and three during the second, lasting about 20
minutes each. The surveyed area spanned over 0.59 km$^2$.



Processing of data from the 2016 UAV flight was carried out using Agisoft Photoscan version 1.2.4
(www.agisoft.com), implementing a Structure-from-Motion (SfM) algorithm for image orientation (see
Barazzetti et al., 2011) followed by a multi-view dense-matching approach for surface reconstruction
(Remondino et al., 2014). The availability of GNSS navigation data was exploited to start the SfM
procedure, shortening the time necessary to register the 288 images acquired by the quadcopter. No pre-
calibration was applied, since the block configuration including strips flown along different directions
was optimal for the estimation of camera calibration parameters (Zhang et al., 2017). A total number of
38,506 tie points (TPs) were extracted for image orientation, corresponding to an average number of 892
TPs per image (see Table 2). The large average number of rays per each TP (6.7) combined with the huge
number of TPs offered a sufficient inner reliability for an effective outlier rejection procedure, which is
applied during bundle adjustment (Kraus, 1997; Luhmann et al., 2014) in Agisoft Photoscan. This
package implements a standard photogrammetric bundle adjustment where GCPs are used as regular
weighted observations, unlike most software packages including SfM algorithms where GCPs are only
used for estimating a 3D rigid-body transformation for geo-referencing the final point cloud. Eight GCPs
(see Fig. 4 and Sec. 2.1.4) were measured for the registration of the photogrammetric blocks and its by-
products into the mapping frame. The root mean square error (RMSE) of the GCPs was 40.5 cm, which
can be used as an indicator of accuracy for the geo-referencing of the photogrammetric block (see Table

2).

The point cloud obtained from the 2016 UAV flight was interpolated to produce a grid DEM (see
Immerzeel et al., 2014), with a cell resolution of 60 x 60 cm. While the high global point density of the
point cloud (89 points/m$^2$) could have permitted a higher spatial resolution, the DEM would have to be
subsampled when computing the differences with other grids. This spatial resolution was considered
sufficient for the analysis of volumetric changes. An orthoimage was also generated from UAV oriented





images and the DEM, with a resolution of 15 cm. Both the DEM and the orthoimage were exported in
the ITRS2000 / UTM 32N mapping coordinate frame.

### 2.1.2 Terrestrial photogrammetry

A terrestrial photogrammetric survey was carried out during the 2016 campaign to reconstruct the
topographic surface of the glacier terminus, which presented several vertical and sub-vertical surfaces
whose measurement was not possible from the UAV platform in nadir configuration (see Fig. 2e).
Images were captured from 134 ground-based stations. Most camera stations were located in front of the
glacier, and some on both flanks of the valley in the downstream area, as shown in Fig. 5a. A single-lens-
reflex Nikon D700 camera was used, equipped with a 50 mm lens, a full-frame CMOS sensor (36x24
mm) composed by 4256x2823 pixels resulting in a square pixel size of 8.4 μm. This photogrammetric
block was processed using Agisoft Photoscan version 1.2.4, following a similar pipeline as described in
Sec. 2.1.1. In this case, no preliminary information about approximate camera stations was necessary,
neither pre-calibration. In such a case, when the photogrammetric block has a sparse geometry (i.e.,
images have not been collected along ordered sequences) and no approximate orientation parameters
(e.g., camera station from GNSS navigation, as in UAV-photogrammetry) are available, the SfM
procedure is applied first on a block of images at down-sampled resolution. This process may provide
approximate orientation, limiting the search space for corresponding points in the final SfM, which is
applied to full resolution images (Barazzetti et al., 2010).
The geometric configuration of the photogrammetric block of the glacier terminus, including hyper-
redundant convergent images as well as 90° rolled images, was optimal for the estimation of camera
calibration parameters. Seven natural features visible on the glacier front were used as GCPs to be
included in the bundle adjustment computation in Agisoft Photoscan. Measurement of GCPs in the field





was carried out by means of a high-precision theodolite. The measurement of points previously recorded
with a GNSS geodetic receiver (see Sec. 2.1.4) allowed to register the coordinates of GCPs in the
mapping frame. The RMSE of 3D residual vectors on GCPs was 34.4 cm, which can be considered as
the accuracy of absolute geo-referencing. A very high number (59,157) of tie points (TPs) was found on
the images after SfM (see Table 2). In addition, the large mean number of rays per each TP (5.6) resulted
in a high reliability of the observations, which mitigates the risk of undetected errors. The final point
cloud obtained from the dense matching tool implemented in Agisoft Photoscan covers at a very high
spatial resolution the full glacier terminus, with the exception of a few obstructed parts (see Fig. 5b).
This part of the Forni glacier has a very complex shape, which evolves at a high dynamic rate. Thus,
rather than a quantitative evaluation of the ice bulk, here the main purpose of 3D reconstruction is to
allow the morphological analysis of the ice structures and the fracturing and collapsing processes. One
working day and two people were required for accomplishing the photogrammetric data acquisition,
including operations for measuring GCP coordinates.
**2.1.3 Terrestrial Laser Scanning**
A long-range terrestrial laser scanner Riegl LMS-Z420i was used to scan the glacier terminus frontally.
This instrument works on Time-of-Flight mode (www.riegl.com). One instrumental standpoint located
on the hydrographic right flanks of the glacier terminus was established. Issues related to meteorological
conditions and to the limited access to unstable areas close to the glacier terminus prevented the operation
from a second station on the other flank of the valley. This solution would have resulted in reducing the
obstructed areas, as it is usually planned in TLS surveys (see Giussani & Scaioni, 2004). The horizontal
and vertical scanning resolution were set up to provide a spatial point density of approx. 5 cm on the ice
surface at the terminus. Geo-referencing was accomplished by placing five GCPs consisting in cylinders
covered by retroreflective paper (see Scaioni et al., 2004). The coordinates of GCPs were measured by



using a precision theodolite following the same procedure adopted for terrestrial photogrammetry.
Considering the accuracy of registration and the expected precision of laser point measurement, the
global accuracy of 3D points was estimated in the order of ±7.5 cm. The completion of the TLS survey
required half working day, including the time necessary for GCP measurements. A team of four to five
people was required for the transportation of the instruments (laser scanner, theodolite, at least two
topographic tripods and poles, electric generator and ancillary accessories).
**2.1.4 GNSS ground control points**
Before the 2016 surveys, eight control targets were placed both outside the glacier and on the glacier
tongue (see Fig. 4). Differential GNSS data were acquired at their location for the purpose of accurate
geo-referencing of UAV, terrestrial photogrammetry and TLS data. While for geo-referencing of UAV
data the GCPs were directly visible on the quadcopter images, for terrestrial photogrammetry and TLS
they were adopted for the registration of theodolite measurements (for practical details about standard
surveying operations see Schofield & Breach, 2007). The targets consisted in a piece of white fabric 80
x 80 cm wide, with a circular marker in red paint chosen to provide contrast against the background.
Such GCPs were positioned on stable glacier areas or flat boulders (see Fig. 6).
GNSS data were acquired by means of a pair of Leica Geosystems 1200 geodetic receivers working
in RTK (Real-Time Kinematics) mode, see Hoffman-Wellenhof (2008). One of them was set up as
master on a boulder beside Branca Hut, where a monument had been established to be used as reference
point for GNSS surveys in the Forni Glacier region. The coordinates of this point were already known in
the geodetic/mapping reference frame ITRS2000 / UTM 32N and were used for geo-referencing all other
points measured with GNSS. The second receiver was used as a rover, communicating via radio link
with the master station. The maximum distance between master and rover was less than 1.5 km, but the
local topography prevented broadcasting the differential corrections in a few zones of the glacier.



Unfortunately, no mobile phone services were available and consequently the internet network could not
be accessed, precluding the use of the regional GNSS real-time positioning service. The theoretical
accuracy of GCPs was estimated in the order of 2-3 cm.

**286 2.2 2014 UAV photogrammetric survey**

The first UAV survey conducted over the tongue of Forni Glacier took place on 28[th] August 2014, using
a SwingletCam fixed wing aircraft (see Fig. 3a). This commercial platform developed by SenseFly, with
basic technical features reported in Table 1, carries a Canon Ixus 127 HS compact digital camera. The
UAV was flown in autopilot mode with a relative flying height of approximately 380 m above the average
glacier surface, which resulted in an average GSD of 11.9 cm. The flight plan was organized by using
the proprietary software eMotion, by which the aircraft follows predefined waypoints with a nominal
along-strip overlap of 70%; sidelap was not regular because of the varying surface topography, but ranged
around 60%. Flight operations started at 07:44 AM and ended at 08:22 AM. Early morning operations
were preferred as during this time of day the glacier is not yet directly illuminated by the sun, thus diffuse
illumination predominates over the glacier surface, and wind speed is at its lowest (Fugazza et al., 2015).
These conditions are therefore optimal to avoid saturating the camera pictures due to the high reflectivity
of ice surfaces as well as to minimize blurring effects due to the UAV motion. In addition, the presence
of tourists on the glacier is reduced during this time of the day. Pictures were automatically captured by
the UAV platform, selecting the best combination of sensor aperture (F=2.7), sensitivity (between 100
and 400 ISO) and shutter speed (between 1/125 s and 1/640 s).
Compared to multi-rotor platforms, fixed wing aircraft are capable of longer flight time on glaciers, due
to their simple structure and the ability to exploit aerodynamics to take advantage of gliding and reduce
battery consumption (Bhardwaj et al., 2016a). This allowed covering an area of 2.21 km$^2$ in just two



flight campaigns, with a low altitude take-off (lake Rosole, close to Branca Hut, see Fig. 1b). Both the
terminal parts of the central and eastern ablation tongue were surveyed. The considerable difference in
area covered during the 2014 and 2016 surveys is due to the reduced battery life of the quadcopter and
lower flying height throughout the 2016 survey.
Processing of data from the 2014 UAV flight was carried out using Agisoft Photoscan version 1.2.4 in a
similar approach to the one applied for UAV-photogrammetry data collected in 2016. Since no GCPs
were measured during the 2014 campaign, the registration of this data set into the mapping frame was
based on GNSS navigation data only. Consequently, a global bias in the order of 1.5-2 m resulted after
geo-referencing, and no control on the intrinsic geometric block stability could be possible. After the
generation of the point cloud, a DEM and orthoimage were produced following the methods outlined in
Sec. 2.1.1, with the same spatial resolutions of final products of 60 cm and 15 cm, respectively.
**2.3 2007 DEM**
The 2007 TerraItaly DEM was produced by BLOM C.G.R (Compagnia Generale Riprese Aeree) for
Lombardy region. It is the final product of an aerial survey over the entire region, that was conducted
with a multispectral pushbroom Leica ADS40 sensor acquiring images from a flying height of 6,300 m
with an average GSD of 65 cm. The images were processed to generate a DEM with a cell resolution of
2 m x 2 m, and projected in the former national 'Gauss Boaga - Fuso I' coordinate system based on the
Monte Mario datum (Mugnier, 2005). Heights were converted from ellipsoidal to geodetic using the
official software for datum transformation in Italy (Verto ver. 3), which is distributed by the Italian
Geographic Military Institute (IGMI). The final vertical accuracy reported by BLOM C.G.R. is ± 3 m.
The only processing step performed within this study was the datum conversion to ITRS2000, using a
seven-parameter similarity transformation based on a local parameter set provided by IGMI.





### 2.4 DEM co-registration

Several studies have found that errors in individual DEMs, both in the horizontal and vertical domain,
propagate when calculating their difference leading to inaccurate estimations of thickness and volume
change (Berthier et al., 2007; Nuth & Kaab, 2011). In the present study, different approaches were
adopted for geo-referencing all the DEMs (2007, 2014, 2016) used in the analysis of the volume change
of the Forni Glacier tongue. The 2007 DEM was extracted from a regional data set, which required a
transformation from the old datum 'Gauss-Boaga - Fuso I' to the present datum ITRS2000/UTM 32 N.
This transformation has an absolute positional accuracy at cartographic level in the order of 1-2 m,
depending on the zone. The DEM obtained from 2014 UAV campaign was geo-referenced on the basis
of onboard GNSS navigation data, with an accuracy with respect to the above mentioned mapping datum
in the order of 1.5-2 m. On the other hand, the most recent DEM derived from the UAV flight (2016)
was geo-referenced using a set of GCPs measured with geodetic-grade GNSS receivers. The average 3D
residuals of these GCPs, which is in the order of 40.5 cm, can provide an estimate of the global geo-
referencing accuracy of the 2016 data set.
To compute the relative differences between the DEMs, a preliminary co-registration was therefore
required. The method proposed by Berthier et al. (2007) for the co-registration of two DEMS was
separately applied to each DEM pair (2007-2014; 2007-2016; 2014-2016). Following this method, in
each pair one DEM plays as reference ('master'), while the other is used as 'slave' DEM to be iteratively
shifted along x and y directions by fractions of pixel to minimize the standard deviation of elevation
differences with respect to the 'master' DEM. Only areas assumed to be stable are considered in the
calculation of the co-registration shift. The ice-covered areas were excluded by overlaying the glacier
outlines from D'Agata et al. (2014) for 2007 and Fugazza et al. (2015) for 2014. The oldest DEM, which
is also the widest in each comparison, was always set as the master. To co-register the 2014 and 2016



DEMs with the 2007 DEM, both were resampled to 2 m spatial resolution, whereas the comparison
between 2014 and 2016 was carried out at the original resolution of these data sets (60 cm).
All points resulting in elevation differences larger than 15 m were labelled as unreliable, and
consequently discarded from the subsequent analysis. Such larger discrepancies may denote errors in one
of the DEMs or unstable areas outside the glacier. Values exceeding this threshold however were only
found in a marginal area with low image overlap in the comparison between the 2014 and 2016 DEMs,
with a maximum elevation difference of 36 m. Once the final co-registration shifts were computed (see
Table 3), the coefficients were subtracted from the top left coordinates of the 'slave' DEM; the residual
mean elevation difference was also subtracted from the 'slave' DEM to bring the mean to zero.
**3 Results**
**3.1 Comparison between observations from 2016: UAV/terrestrial photogrammetry and TLS**
The comparison between data sets collected during the 2016 campaign had the aim of assessing the
quality of different data sources to be used for subsequent physical analyses. In addition, these
evaluations were expected to provide some guidelines for the organization of future investigations in the
field at the Forni Glacier and in other Alpine sites.
Specifically, in this case the analysis consists in comparing point clouds. It is out of the scope of this
article to address this topic in an exhaustive manner. While the reader may refer to other pieces of
literature to have a broader view about it (e.g., Eltner et al., 2016), here the aim is to apply some existing
criteria and metrics to find out which techniques among UAV photogrammetry (i), terrestrial
photogrammetry (ii), and TLS (iii) should be privileged for glaciological studies under certain conditions.
Of course, comparing two point clouds, which is the simplest case that may be considered, is more
complex than comparing coordinates of specific points that have been measured, e.g., with theodolites,


GNSS sensors or target-based photogrammetry (Luhmann et al., 2014). In such a case, the analysis is
limited to evaluating their discrepancies by merely differencing corresponding coordinates, provided that
the points to compare are defined into the same reference frame. The maximum degree of complexity in
the case of specific point comparison is to define the minimum departure revealing statistical significance
(Teunissen, 2009). In the case of point clouds, no precise point-to-point correspondence generally exists,
since 3D points are obtained using different techniques, setups and algorithms. In addition, not only the
'distance' between point clouds should be assessed to check out their spatial accuracy, but other
properties need to be considered as well. In particular, point density and completeness of a point cloud
are two important aspects that in general do not deserve consideration when dealing with specific points.
Thus a first important property to analyse is the point density, which allows verifying whether the whole
reconstructed surface may be modelled with sufficient detail on the basis of the surveyed point cloud. Of
course, the same point density may be fine for a certain kind of geomorphometry, whilst it may not be
sufficient for others, mainly depending on roughness. Secondly, the completeness of a reconstruction
indicates if the surface reconstruction presents some holes or missing parts, for example because of
occlusions, sensor out-of-range areas, low-texture or low-reflectivity surfaces, and the like.  Eventually,
the accuracy of a point cloud should be assessed by comparison with a reference surface or with a set of
precise points. Different criteria exist for evaluating the spatial 'distance' between two point clouds (see
Lindenbergh and Pietrzyk, 2015; Scaioni et al., 2015), depending on the surface morphology, as
described at paragraph 3.1.2.
In order to analyse point density, completeness and accuracy of point clouds obtained during 2016
campaign by means of techniques (i), (ii) and (iii), five regions shown in Fig. 7 were selected. These
regions are mainly located on the glacier and characterized by different geomorphological properties. In
addition, they were surveyed by almost all the three techniques. The analysis of local regions was



preferred to the analysis of the entire point clouds for two reasons: (1) the partial overlap between point
clouds obtained from different methods; (2) the opportunity to investigate the performances of the
techniques in diverse geomorphological situations.
A short description of each sample window follows:

1.   Glacial cavity located on the right orographic side of the glacier terminus, composed by sub-

vertical and fractured surfaces over 20 m high, and forming a typical semi-circular shape (clearly

visible from the top);

2.   Glacial cavity located on the left orographic side of the glacier terminus. It is over 10 m high with

the typical semi-circular shape as window 1; on top, it is covered by fine- and medium-size rock

debris;

3.   Vertical fault on the left orographic flank of the glacier terminus, over10 m high;

4.   Highly-collapsed area on the central region of the glacier terminus, covered by fine- and medium-

size rock debris and rock boulders; and

5.   Planar surface with a vertical fault on the left orographic side of the glacier terminus, covered by

fine- and medium-size rock debris and rock boulders.

Table 4 reports the size of each sample window as well as the number of points obtained with different
techniques. In window 1, method (i) could not provide points except on the upper part, because of the
presence of sub-vertical cliffs that could not be reconstructed from airborne images. Window 5 was not
covered by TLS (iii), because it was not included in the field-of-view of the selected standpoint. Looking
at the point number in each window, at a first glance terrestrial photogrammetry resulted in a much
consistent data set than other techniques. This is mostly motivated by the flexibility of this methodology,
which allows carrying out data acquisition from multiple stations, depending only on the terrain
accessibility in front of the glacier.



### 3.1.1 Point density and completeness


Point density describes the number of points per unit of surface or volume. Depending on the adopted
surveying techniques, it always depends upon the distance between sensor and surface and the adopted
spatial resolution. While in the UAV-photogrammetry survey the distance camera-object is almost
constant (approx. 180 m) in all sample windows, in the case of terrestrial sensors (TLS &
photogrammetry) this distance is greater and therefore it can influence the point cloud reconstruction. In
the terrestrial photogrammetry survey, the distances between camera stations and the sample windows
ranged from 85 m (window 2) to 137 m (windows 1, 3 and 4), and 206 m (window 5).
In the case of photogrammetry the point cloud reconstruction relies on dense matching, thus the resulting
point density also depends upon the surface texture.
The evaluation of point density using a global descriptor that is applied to the whole point cloud or on
large portions of it cannot provide a useful output in the case of glaciers with complex morphology, since
point density may largely change from one portion of surface to another. More significant is the use of
local descriptors applied on small windows or in the proximity of each point. Local results can be
displayed on maps and summarized by global statistics.
In this study, the number of neighbours $N$ (inside a sphere of radius $R$=1 meter) divided by the
neighbourhood surface was used to evaluate the local point density $D$:
$$D = \frac{N}{\pi * R^2}$$ (1)
This function is implemented in the open-source software CloudCompare (www.cloudcompare.org).
Point cloud completeness refers to the presence of enough points to completely describe a portion of
surface. A rigorous evaluation of this parameter is possible by interpolating a regular surface and by
searching for the presence of points in any sectors of it. Of course, this approach can be easily applied





when the morphology of the surface to reconstruct is regular, for example in the analysis of terrain
topography. On the other hand, in the case of an Alpine glacier terminus, the geometry is much more
complex, and the recourse to this approach is more difficult. Consequently, in this study a heuristic
evaluation based on the visual inspection of the obtained windows was preferred.
Mean values and standard deviations of point density in the five windows are shown in Table 5 and Fig.
8. The following general considerations can be made. The values of point density obtained from
terrestrial photogrammetry (ii) are much higher than others, except in window 5 that features a gentle
slope. In such a case, UAV photogrammetry provided results comparable to the ones of terrestrial
photogrammetry (only approximately three times smaller). On the other hand, the mean point density
achieved when using technique (ii) has a large variability both between different windows, and inside
each window as witnessed by the standard deviations of $D$. Point densities related to UAV
photogrammetry (i) and TLS (iii) are more regular and constant. In case (i), the regularity is due to the
structure of the airborne photogrammetric block, which is made up of organized parallel strips looking
in nadir direction towards the ground. In case (iii), the regularity is motivated by the constant angular
resolution adopted during scanning. In general, each sensor performs better when the surface is
orthogonal to the average sensor looking direction. Mainly, this means that terrestrial techniques (ii) and
(iii) perform better in vertical and sub-vertical cliffs (windows 1 and 2), and in high-sloped surfaces
(windows 3 and 4); on the contrary, UAV photogrammetry provided the best results in the case of
window 5 that is less inclined and consequently could be well depicted in nadir photos.
In term of absolute values, the mean point density obtained with different techniques in the sample
windows may suffice for a correct representation of the glacier outer surfaces and the surrounding terrain.
In order to understand the effect of point density dispersion, the standard deviations were considered.
Since the normal distribution of the data sets made up of point density computed inside each sample



window cannot be proved, an approach based on the use of Chebichev theorem was applied (see
Teunissen, 2009). Based on this theorem, given a population of $N$ members with mean $\mu$ and standard
deviation $\sigma$, the minimum frequency of the elements comprehended in the interval $\mu \pm 2\sigma$ is 75%. This
means that in both queues, 25% of the population can be found. Since the inferior part of the population
of point density may be too low to guarantee a detailed modelling of the surface, the upper limit
corresponding to the inferior 12.5% percentile was computed and reported in Table 5.
Based on the mere analysis of point density, terrestrial photogrammetry outperformed other techniques.
In windows 1-4, mean values of this parameter ranged between 1384-2297 points/m$^2$, which are
equivalent to a range between approximately 14-23 points/dm$^2$. A lower point density was obtained in
window 5 that is exposed upwards, with approximately 500 points/m$^2$. Looking at the limit of the inferior
12.5% percentile, three windows (1-3) show a very high value between 766-880 points/m$^2$, while in
window 5 a value of 31 points/m$^2$ was obtained. All these values were retained sufficient for the
reconstruction of different surfaces in the sample windows, according to their different geomorphic
complexity, except in the case of window 5.
In the case of UAV photogrammetry (i), similar results about point density were found in all sample
windows, especially for the standard deviations that were always in the range 22-29 points/m$^2$. Mean
values were between 103-109 points/m$^2$ in windows 2-4, while they were higher in window 5 (141
points/m$^2$). Due to the nadir acquisition points, the reconstruction of vertical/sub-vertical cliffs in window
1 was not possible. The limit of the inferior 12.5% percentile was between 49-62 points/m$^2$ because of
sub-vertical orientation of this sample window. A higher value (97 points/m$^2$) was found in the case of
window 5. Results obtained from photogrammetry based on terrestrial and UAV platforms may be
retained quite complementary: the former are suitable for the reconstruction of vertically oriented
regions, the latter for those surfaces looking upwards.





More varying results were obtained from the use of TLS. With the only exception of window 5, where
no sufficient data were recorded due to the position of this region with respect to the instrumental
standpoint, a mean value of point density ranging from 141-391 points/m$^2$ could be found. Standard
deviations ranged between 69-217 points/m$^2$, moderately correlated with respective mean values. On the
other hand, in correspondence of the inferior 12.5% percentile, too low values were found (0-29
points/m$^2$). These results showed that the adopted long-range TLS instrument was not completely
suitable for surveying the glacier terminus.
In Fig. 9 and 10, the maps of point density in windows 2 and 3 are shown, respectively. These windows
depict some typical problems related to the completeness of surface reconstruction that may be obtained
from the adopted techniques. UAV photogrammetry can provide a sufficient point density in all parts of
those regions that are exposed upwards, as can be seen also in the global model of the glacier shown in
Fig. 12. Results are also satisfying in gently sloped areas, as it can be observed in windows 2 and 3, see
Fig. 9 and 10. Vertical and sub-vertical surfaces cannot be investigated, requiring the integration with a
terrestrial sensor or the installation of the payload camera in oblique configuration.
Terrestrial photogrammetry offers the chance to gather images from several positions. This results in
reducing the effect of occlusions with a consequently more complete reconstruction. On the other hand,
this technique is limited when the surface to reconstruct is close to the horizontal orientation. In such a
case, the integration with UAV data is required.
In general, TLS suffers from occlusions as all 3D measurement techniques (see for example results in
window 2 in Fig. 9). Besides, these instruments are still quite complex to be carried and setup. These
limits prevent the acquisition from several viewpoints as it is possible when using photogrammetry. Data
acquisition is also difficult in regions that are close to be parallel to the laser beams and in the presence
of wet surfaces. Another problem with the adopted TLS concerned the angular resolution adopted for





scanning, which was set up to obtain a linear resolution on the ground surface of approximately 1 point
every 5 cm, while keeping the acquisition time at about 40 minutes. Using a smaller angular resolution
would have resulted in much longer acquisition time (for example, using half resolution could be possible
in four times the acquisition time). On the other hand, the adopted laser scanner instrument still has a
slow acquisition speed (approx. 12 kHz) if compared with up-to-date Time-of-Flight lasers which may
work much faster (over 100 kHz).
Finally, internal parts of fractures and faults are usually problematic to reconstruct by means of all
measurement techniques. However, their presence can be easily detected in the point clouds.
**3.1.2 Accuracy**
The evaluation of the accuracy of a point cloud requires a data set of benchmarking observations. When
the geometry of an object is known a priori (for example a planar surface), the accuracy can be evaluated
by comparing the point cloud to the mathematical model of the surface itself. On the other hand, in the
case of terrain or glacier geomorphology, this solution is clearly not viable. In such a case, benchmarking
data are required, for example, another reference point cloud or a set of specific points (see Eltner et al.,
2016). Due to the fast dynamics of the glacier tongue under investigation, the only available data sets to
compare are the ones collected during the 2016 campaign, i.e., point clouds derived from UAV and
terrestrial photogrammetry, and from TLS. The approach applied to estimate the accuracy was to
compare in a pairwise manner the point clouds obtained from different surveying techniques. The
analysis was carried out inside the same five sample windows used for investigating the point cloud
density. Although each point cloud had been already geo-referenced as described in Section 2, some
residual errors could be expected. In order to get rid of these discrepancies that would affect both surfaces
to compare, a preliminary co-registration using the ICP algorithm (Pomerleau et al., 2016) was
conducted. Secondly, point clouds in corresponding sample windows were compared using M3C2



algorithm implemented in CloudCompare (Lague et al., 2013). The advantage of this algorithm is that it
is able to provide a map of signed distances between corresponding and co-registered point clouds. The
positive direction of distances goes outside the 'reference' point cloud. Therefore, when a computed
M3C2 distance is positive, the compared ('slave') point cloud lies outside with respect to the reference
point cloud. Unlike standard algorithm for comparing DEMs that operate along a predefined direction
(see, e.g., Scaioni et al., 2013), here the direction of distance depends on the local normal to the point
cloud. This method is therefore suitable to compare complex point clouds such as the ones in the sample
windows. The point cloud collected using TLS was used as reference, since these measurement sets were
retained to be the most accurate, although their point density and completeness may not be the best ones
as proved in the previous section. When comparing both photogrammetric data sets, the one obtained
from UAV was used as reference because of the even distribution of point density within the sample
windows.
Table 6 reports some statistics on the computed M3C2 distances in terms of mean values and standard
deviations. Where no data are shown, the comparison was not possible since one or both point clouds
were incomplete (for example, in the case of data sets in windows 1 and 5). The comparison between
TLS and terrestrial photogrammetry resulted in a high similarity between the accuracy of both point
clouds, provided that the TLS point cloud may be assumed as benchmarking surface. No large departures
were found between results obtained in different sample windows. In addition, the RMSE are in the order
of theoretical precision achievable with photogrammetry techniques under the actual acquisition
geometry (Luhmann et al., 2014). This result confirms the small differences between point clouds.
Nevertheless, this analysis was carried out after a posteriori ICP-based registration that may have fixed
residual geo-referencing errors. By looking at the residuals on GCPs for TLS and terrestrial
photogrammetry (7.5 cm and 34.4 cm, respectively) a bias larger than the RMSE found for the distances





in different windows were obtained for the latter. Indeed, the identification of natural GCPs on the glacier
surface was quite difficult and resulted in low-precision measurements. A solution to improve the quality
of geo-referencing in the photogrammetric block should be considered, for example by directly
measuring a part of the photo-stations as proposed in Forlani et al. (2014), instead of recurring to GCPs
on the glacier surface.
The comparison between TLS and UAV photogrammetry provided significantly worse results that may
be summarized by the RMSEs in the range 21.1-37.7. These departures may be attributed to two main
reasons: (1) these techniques offer the best performances in opposite situations: flat terrain in the case of
a UAV survey and vertical surfaces in the case of TLS; (2) the UAV flight was geo-referenced on a set
of GCPs obtaining a RMSE of residuals of 40.5 cm, thus the ICP co-registration may have not totally
compensated the existing bias.
The comparison between UAV (assumed as reference) and terrestrial photogrammetry provided similar
results to the ones obtained in the previous analysis. Indeed, the same reasons may still hold in such a
case, since two point clouds obtained from ground-based and airborne camera poses were compared.
This makes it possible to fuse both point clouds from photogrammetry to obtain a complete model of the
glacier tongue, as reported in Sec. 3.3.
**3.2 Glacier Thickness change 2007-2016**
After DEM co-registration, the resulting shifts reported in Table 3 were applied to each 'slave' DEM,
including the entire glacier area. Then the elevations of the 'slave' DEM were subtracted from the
corresponding elevations of the 'master' DEM to obtain the ΔDEM. Each ΔDEM was then clipped within
the glacier outlines to provide pairwise relative estimates of glacier elevation change. First this operation
was carried out by considering the largest possible area in each ΔDEM (see Fig. 11 and Table 7), using



the oldest outlines available. This operation was aimed at investigating ice lost in areas of glacier retreat.
Secondly, a minimum extension common to all three DEMs was analysed as a means of independently
checking the quality of each surface and finding thinning trends over a reference area. Indeed, while the
2007 aerial DEM covers the entire Lombardy region, the coverage of both UAV DEMs (2014 and 2016)
is limited. Although the DEM from 2016 has the smallest extent, it is not completely included within the
extension of 2014 DEM. In practice, however, the reference area almost completely refers to the extent
of the 2014-2016 analysis, covering 0.32 km$^2$. For the second comparison, the volume change over the
glacier tongue and its uncertainty were estimated as well. The method proposed in Howat et al. (2008)
was applied, which expresses the uncertainty of volume change as the combination of the standard
deviation computed from the residual elevation difference over stable areas, and the truncation error
implicit when substituting the integral in volume calculation with a finite sum, according to Jokinen and
Geist (2010).
When comparing over the maximum possible glacier extension, the latter appears clearly inversely
related to the thinning rates. However, the comparison between 2007 and 2014 includes sections of the
central tongue that only lost an average 15 m of ice. Considering a common reference area, an
acceleration of glacier thinning seems to have occurred over recent years over the lower glacier tongue,
from -4.55 in 2007-2014 to 5.20 ma$^{-1}$ in 2014-2016 (see Table 8).
The eastern ablation tongue appears the most affected by glacier thinning between 2007 and 2014, with
ice thickness changes persistently below −30 m over the period and between -40/-50 m between 2014
and 2016. The greatest ice loss between 2007 and 2014 occurs in correspondence with local collapse of
ice cavities, localized in small areas of the eastern tongue (see Fig. 11a), with local thinning generally
above -50 m and a maximum of -66.80 m. Conversely, between 2014 and 2016 glacier thinning is close
to the mean of approximately 10 m on both the central and eastern sections of the tongue. Only in areas



of local collapse is this value greatly exceeded, with a maximum of -38.71 m thinning at the terminus
and local maxima above -25 m on the medial moraine and left margin of the central tongue (see Fig.
11c).

### 3.3 Data fusion of point clouds from UAV and terrestrial photogrammetry

As shown in the previous analysis, data sets obtained from ground-based and UAV photogrammetry are
quite complementary. In order to derive a full 3D model of the terminal part of the tongue of Forni
Glacier, point clouds were fused together. The merged point cloud was subsampled to keep a minimum
distance between adjacent points of 20 cm (see Fig. 12). The size of this point cloud was approximately
4.4 million points. RGB information from photogrammetric data sets were used for colouring the point
clouds before data fusion. The merged point cloud was used for the analysis of glacier hazards and risks
reported in Sec. 4.2.

### 4 Discussion

### 4.1 Evolution of the glacier tongue

The outcomes of the DEM differencing procedure indicate generalized thinning of the Forni Glacier
tongue over the entire study period. Independent validation of the thinning rates found in this study is
available from Senese et al. (2012), who estimated the specific mass balance at the glacier AWS between
2006 and 2009, by calculating ablation via the glacier energy budget and accumulation via a sonic ranger.
The authors reported a mean annual mass balance of -4.70 m w.e. between 2005 and 2009, with minimum
negative of -4.20 and maximum negative of -4.90 m w.e. In comparison, by calculating the geodetic mass
balance using the mean ice density of 0.917 $g/cm^3$, we found mean annual values of -4.17 ± 0.22 m w.e.
between 2007 and 2014 and -4.36 ± 0.27 between 2007 and 2016 over the lower part of the glacier
tongue, slightly lower but encompassing a wider spatial and temporal range. Besides, our data suggests
that thinning over the last two years was higher than between 2007 and 2014.



Although thinning rates are high over the entire tongue, they are not homogeneous, and both the glacier
preexisting surface morphology and debris input from the valley walls (Azzoni et al., under revision)
played an active role in determining the evolution of the glacier tongue that we identified by means of
elevation transects on the three DEM surfaces. In particular, the ice-cored medial moraines changed
dramatically. In 1987, they were 12 m tall and 50 m wide at maximum on the glacier tongue (Smiraglia,
1989), but both width and height gradually increased over the years: the eastern moraine is the more
prominent of the two, widening asymmetrically towards the terminus, with the left flank being the widest
(see Fig. 13). Along of the middle transect, the height of the eastern moraine remained stable at 15 m
between 2007 and 2016, while its width increased from 80 to 100 m. During this period, a new moraine
also formed on the eastern tongue, reaching a height of approximately 7 m in 2016. East of the this newly
formed moraine, ice thinning was above 60 m between 2007 and 2016, likely due to the reduction of
mass input from the eastern icefall and the development of a thin debris cover promoting ice ablation
(see Fig. 13, middle transect).
Further upvalley, as a result of differential ablation, thinning was lower on the medial moraine than on
the exposed ice surface. Thus, the height of the eastern moraine increased from 20 to approximately 26
m between 2007 and 2016, while its width went from 100 to 145 m. A small new moraine developed,
joining the main one in SE-NW direction. The most prominent feature on the central tongue is however
the large collapse at the left margin, with 26 m ice thinning between 2014 and 2016 (Fig. 13, bottom
transect). At the terminus, the height of the eastern medial moraine decreased between 2007, when it was
about 20 m tall, and 2016, when it was approximately 13 m, due to the development of normal faults
subparallel to the main medial moraine direction. Conversely, its width gradually increased from 100 to
130 m in 2016. The glacier surface once flat is now increasingly hummocky both on the central and
eastern sections of the tongue (see Fig. 13 top transect).



### 4.2 Glacier-related hazards and risks

The collapse of sections of the glacier appears to pose the most significant risk to mountaineers. Collapses are more dangerous than crevasses because of the larger size and relief involved. Besides, already collapsed areas could be filled with snow and rendered entirely or partly invisible to mountaineers. Currently, hikers heading to Mount San Matteo during the summer take the trail crossing the Forni Glacier on the central tongue, dangerously close to the collapsing glacier terminus. During wintertime, ski-mountaineers instead access the glacier from the eastern side, crossing the medial moraine and potentially collapsed areas there (see Fig. 14).

While most collapsed areas on the glacier tongue are in fact normal faults, two large ring fault systems can be identified: the first, located on the eastern section (see Fig. 2d and 15a), covered an area of $25.6 \times 10^3$ m$^2$ and showed surface lowering of up to 5 m in 2014. This area was not surveyed in 2016, since field observation did not show evidence of further subsidence. Conversely, the ring fault that only emerged as a few semi-circular fractures in 2014 grew until cavity collapse, with a vertical displacement up to 20 m and further fractures extending south-eastward (see Fig. 2c and 15b), thus potentially widening the extent of collapse in the future. As regards normal faults, those on the eastern moraine developed rapidly in the vertical domain reaching a relief of 12 m in 2016. The collapse was even more rapid at the terminus, leading to the formation of three sub-vertical facies, which could not be analyzed by UAV data alone given the nadir image acquisition. Here, integration of close-range photogrammetry proved necessary to investigate the cliff height, which reaches up to 24 m, while the height of the vault is as low as 10 m. The fast retreat pace of this glacier suggest the terminus will recede along the fault system on the eastern moraine, increasing the occurrence of hazardous phenomena in this area where the vulnerability (occurrence of paths followed by mountaineers) is relatively high, thus making glacier risk particularly significant here. Upvalley, the increased relief of the medial moraine might cause more frequent landslides and rockfalls, which can be dangerous for mountaineers during the summer season.



Finally, the collapse of the glacier tongue at its margins will further compromise access to the glacier for
winter activities.
In these fragile and dynamic areas, the combination of UAV and terrestrial surveys potentially allows
following the evolution of glacial hazards (e.g., ring faults, collapsed zones, glacier sectors with a very
thin ice layer, etc.) over a summer season or with a higher time frequency than previously possible. This
information will be crucial to manage the vulnerability of the area and thus reduce the level of risk. In
fact, based on the orthophotos obtained from UAV surveys, it will be possible to identify safer paths
where mountaineers and skiers can visit the glacier and reach the most important summits (e.g., Mount
San Matteo, etc..) without crossing the most dangerous zones. These safer paths will be identified with
the help of local alpine guides and reported in the webpage of the Stelvio National Park and in the
Geoportale of the Lombardy Region, to increase the number of citizens potentially visiting the area who
will be informed about the dangers and the safest paths. Our surveys also helped describe new categories
of glacier hazards and risk (for a review see RGSL, 2003), such as faults and ring faults, which were not
considered in the guidelines for the management of environmental risk in the past. Their recent
emergence, driven by the present climate change and the subsequent glacier downwasting, requires a
new approach to risk management. In this context, it is at present impossible to reduce the glacier hazards
and the only chance to lower the risk level is to reduce the vulnerability by changing the tourist paths to
safer areas, only possible by applying UAV and terrestrial photogrammetry-based monitoring.
**5 Conclusions**
In our study, we assessed the potential of UAV and terrestrial photogrammetry to map surface features
pertaining to the collapse of a large Alpine glacier (Forni Glacier, Italian Alps), such as ring faults,
representing hazards for mountaineers, and reconstruct the thickness changes and variations in
topography. We assessed the accuracy of surface elevations by comparing point clouds from UAV and



terrestrial photogrammetry against those obtained from TLS and by measuring DEM differences from
repeat UAV surveys on stable areas.
By comparing different DEMs of the glacier tongue, we found an increased rate of glacier ablation in
recent years, reaching $5.20 \pm 1.11$ ma$^{-1}$ between 2014 and 2016, with a maximum surface elevation
change of -38.71 m. At the same time, the eastern medial moraine and terminus underwent major
changes: the first widened and increased in relief, while also experiencing several faults; the second
experienced relevant collapses while the glacier surface became increasingly hummocky. We combined
point clouds obtained from UAV- and terrestrial photogrammetry to investigate the hazards on the glacier
tongue and the risk to mountaineers and skiers following routes to the popular summits of the area. The
glacier terminus is at present the most dangerous area, because it hosts vertical cliffs with a relief up to
24 m and it is the main gateway to the glacier during the summer. Collapses at the margins of the central
tongue also increase the risk for skiers during winter. The scenario of present glacier downwasting,
besides potentially increasing mass movements from currently unstable slopes, might further
compromise the access to Forni Glacier in the future, modifying the surface topography and increasing
the occurrence of collapses.
Our results also show that a sufficient level of accuracy can be achieved by using UAVs to monitor the
glacier topographic changes over yearly timescales, as the variations that take place are larger than the
associated uncertainty. Thus, UAV surveys could be used effectively to investigate the glacier
downwasting. The integration with terrestrial photogrammetry is crucial to establish a valid alternative
to TLS to monitor recurrent glacier hazards with larger impact on downstream populations, allowing the
estimation of volumes involved in the detachment of seracs or hanging glaciers, and measurements of
the height of moraine dams to help manage potential GLOFs. Terrestrial photogrammetry may provide
better results than TLS in term of point density and point cloud completeness, thanks to the chance to



capture images from a high number of camera stations, limiting occlusions. When analyzing the point
cloud accuracy, the comparison of photogrammetric outputs with respect to TLS outputs revealed
average discrepancies in the order of a few centimeters in the case of terrestrial blocks, and a few
decimeters in the case of UAV blocks. This result, although quite promising, is not yet sufficient for
monitoring of intra-seasonal variations of the glacier topography, or very rapid changes occurring on
daily timescales such as those involved in the collapse of ice blocks at the terminus. Beside the
combination with terrestrial photogrammetry, improvements to our UAV survey design might include a
greater number of GCPs sampled in a dense spatial network, but the glacier dynamics evolving towards
a collapse scenario might make this solution highly unpractical over time. As an alternative, our choice
of a custom UAV platform adopted in 2016 should ease a low cost switch to an RTK navigation system,
reducing the number of GCPs necessary for geo-referencing. While fixed-wing UAVs outperform
multicopters in terms of area covered and aircraft stability, the adaptability of our quadcopter platform,
together with the flexibility of terrestrial photogrammetric surveys might eventually enable continuous
monitoring of the Forni glacier and the provision of rapid hazard detection services for mountain guides
and the tourism sector in Stelvio National Park.
**Competing interests**
The authors declare that they have no conflict of interest.
**Acknowledgements**
This study was funded by DARAS, the department for autonomies and regional affairs of the presidency
of the council of the Italian government. The authors acknowledge the central scientific committee of
CAI (Club Alpino Italiano – Italian Alpine Club) and Levissima San Pellegrino S.P.A. for funding the
UAV quadcopter. The authors also thank Stelvio Park Authority for the logistic support and for
permitting the UAV surveys and IIT Regione Lombardia for the provision of the 2007 DEM.



Acknowledgements also go to the GICARUS lab of Politecnico Milano at Lecco Campus for providing
the survey equipment. Finally, the authors would also like to thank Tullio Feifer, Livio Piatta, and Andrea
Grossoni for their help during field operations.

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





**Tables**

|  | **2016 Survey** | **2014 Survey** |
|---|---|---|
| **Aircraft type** | Quadcopter | Fixed wing |
| **Commercial name** | Customized, with Tarot frame 650 size, VR Brain 5.2 Autopilot & APM Arducopter 3.2.1 Firmware | SwingletCam built by SenseFly |
| **Digital camera** | Canon Powershot ELPH 320 HS | Canon Ixus 127 HS |
| **Camera technical features** | 16 Megapixel, focal length 4.3 mm | 16 Megapixel, focal length 4.3 mm |
| **GNSS antenna** | GPS+GLONASS (Galileo compatible) | GPS only |
| **Weight (incl. payload)** | 2.75 Kg | 0.50 Kg |
| **Battery time** | 20-25 minutes | 30 minutes |

*Table 1: Details of UAV platforms employed during the 2016 and 2014 surveys*





| Block | #Images | Total #valid TPs | Mean # projections per TP | Mean/min # TP per image | Mean/max RMSE reprojection [pixel] | Point cloud # points | Mean GSD[cm] | # GCPs | RMSE on GCPs [cm] |
|---|---|---|---|---|---|---|---|---|---|
| *Terrestrial Photogrammetry (Glacier Terminus)* | 134 | 59,157 | 5.6 | 2,455 / 744 | 0.30 / 0.73 | 27.1M | 1.5 | 7 | 34.4 |
| *UAV 2016* | 288 | 38,506 | 6.7 | 892 / 115 | 0.21 / 0.31 | 75.2M | 5.7 | 8 | 40.5 |
| *UAV 2014* | 85 | 76,856 | 4.4 | 3935 / 2231 | 0.17 / 0.19 | 55.7M | 11.9 | 0 | n.a. |

*Table 2: Statistics of photogrammetric blocks (TP: tie points; GCP: ground control points; RMSE: root*
*mean square error).*



| DEM pair | Elevation differences without co-registration shifts ($\mu_{\Delta H} \pm \sigma_{\Delta H}$) [m] | Co-registration shifts | | Elevation differences without co-registration shifts ($\mu_{\Delta H} \pm \sigma_{\Delta H}$) [m] |
|---|---|---|---|---|
| | | X [m] | Y [m] | |
| 2007-2014 | 1.96±2.60 | 1.11 | -1.11 | 0.00±1.70 |
| 2007-2016 | -0.43±3.48 | 2.44 | -1.11 | 0.00±2.60 |
| 2014-2016 | -2.92±3.21 | -0.20 | -1.30 | 0.00±2.22 |

*Table 3: Statistics of the elevation differences between DEM pairs before and after the application of*
*co-registration shifts.*





| Sample Window | Size of sample windows | #points in sample windows | | |
|---|---|---|---|---|
| | Width x depth x height [m] | (i) UAV photogrammetry | (ii) Terrestrial photogrammetry | (iii) TLS |
| 1 | 49 x 57 x 22 | - | 1984k | 141k |
| 2 | 43 x 42 x13 | 76k | 2175k | 130k |
| 3 | 45 x 11 x14 | 43k | 712k | 25k |
| 4 | 24 x 28 x 10 | 62k | 557k | 33k |
| 5 | 55 x 72 x 18 | 406k | 810k | - |


*Table 4: Number of points in each sample window.*



| Sample Window | Mean and standard deviation of point density [points/m$^2$] | | | Number of point above the lower 12.5% percentile | | |
|---|---|---|---|---|---|---|
| | (i) UAV Photogrammetry | (ii) Terrestrial Photogramm. | (iii) TLS | (i) | (ii) | (iii) |
| 1 | - | 1654±637 | 226±100 | - | 880 | 26 |
| 2 | 109±29 | 2297±708 | 391±217 | 61 | 881 | 0 |
| 3 | 103±27 | 1978±606 | 151±60 | 49 | 766 | 31 |
| 4 | 108±22 | 1384±530 | 141±69 | 62 | 324 | 2 |
| 5 | 141±22 | 485±227 | - | 97 | 31 | - |


*Table 5: Mean and standard deviation of point density computed in five sample windows on the Forni*
*Glacier terminus.*



| Sample Window | | Means and Std. Dev.s of M3C2 distances [cm] | | | RMSE of M3C2 distances [cm] | | |
|---|---|---|---|---|---|---|---|
| | Ref. | TLS | TLS | UAV Photogramm. | TLS | TLS | UAV Photogramm. |
| | Slave | Terrestrial Photogramm. | UAV Photogramm. | Terrestrial Photogramm. | Terrestrial Photogramm. | UAV Photogramm. | Terrestrial Photogramm. |
| 1 | | 4.5±7.4 | - | - | 8.7 | - | - |
| 2 | | -1.1±10.5 | 14.8±34.7 | -14.5±26.7 | 10.6 | 37.7 | 30.4 |
| 3 | | 8.4±4.1 | 14.7±15.1 | -8.5±18.9 | 9.4 | 21.1 | 20.7 |
| 4 | | 2.8±5.3 | 9.4±22.2 | -2.3±24.9 | 6.0 | 24.0 | 25.0 |
| 5 | | - | - | -8.5±25.3 | - | - | 26.7 |


*Table 6:  Statistics on computed M3C2 distances.*




| DEM pair | Glacier Area analysed [km$^2$] | Mean thickness change [m] | Mean thinning rates [ma$^{-1}$] |
|---|---|---|---|
| 2007-2014 | 1.03 | -25.06 ± 1.70 | -3.58 ± 0.24 |
| 2007-2016 | 0.46 | -37.39 ± 2.60 | -4.15 ± 0.29 |
| 2014-2016 | 0.32 | -10.40 ± 2.22 | -5.20 ± 1.11 |

*Table 7: Average thickness change and thinning rates from DEM differencing over the maximum glacier*
*areas for each DEM pair, and corresponding uncertainty.*





| DEM pair | Mean thickness change [m] | Mean thinning rates [ma$^{-1}$] | Volume Change [$10^6$ m$^3$] |
|---|---|---|---|
| 2007-2014 | -31.91 ± 1.70 | -4.55 ± 0.24 | -10.00 ± 0.12 |
| 2007-2016 | -42.86 ± 2.60 | -4.76 ± 0.29 | -13.46 ± 0.14 |
| 2014-2016 | -10.41 ± 2.22 | -5.20 ± 1.11 | -3.29 ± 0.05 |

*Table 8: Average ice thickness change, thinning rates and volume loss from DEM differencing over a*
*common reference area of 0.32 km$^2$ for all DEM pairs. Uncertainty of thickness change expressed as 1σ*
*of residual elevation differences over stable areas after DEM co-registration. See text for an explanation*
*of the uncertainty of volume changes.*





**Figures**

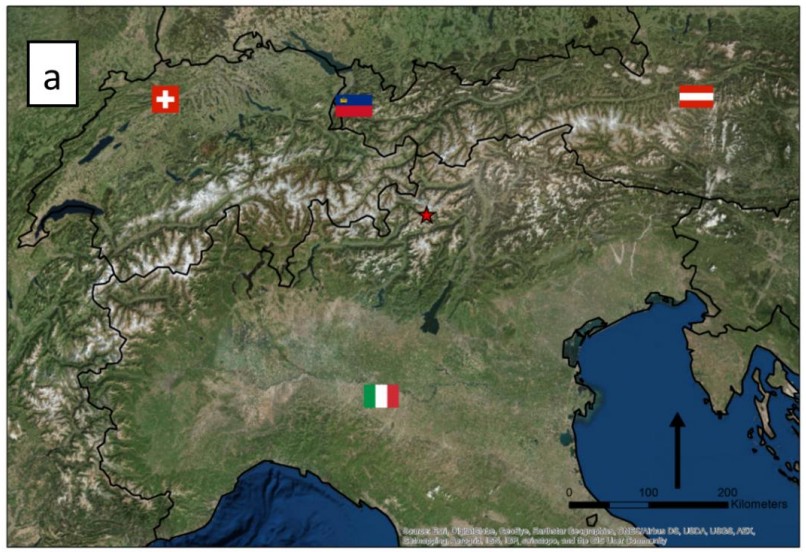

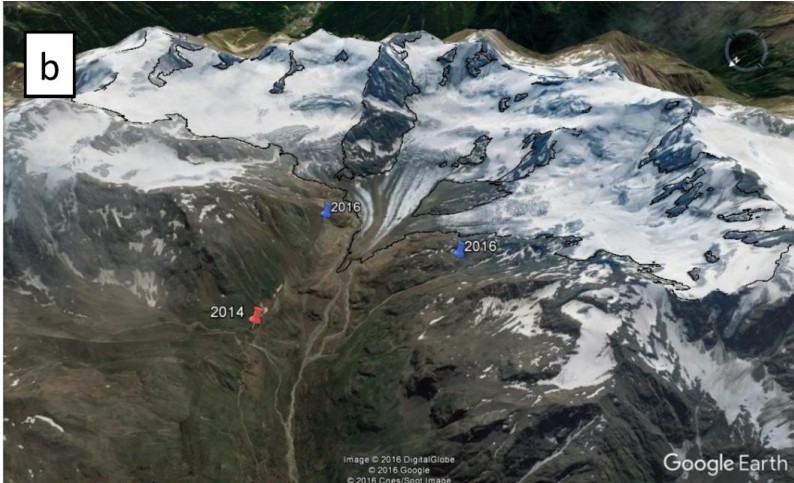

*Figure 1: (a) Location of the Forni Glacier, marked with a red star, within Italy and the Central*
*Alps. (b) Perspective view of the glacier and location of the take-off/landing sites for the 2014*
*and 2016 UAV surveys (in 2016 two different landing sites were used). Base maps courtesy of*
*Bing Maps© and Google Earth©*




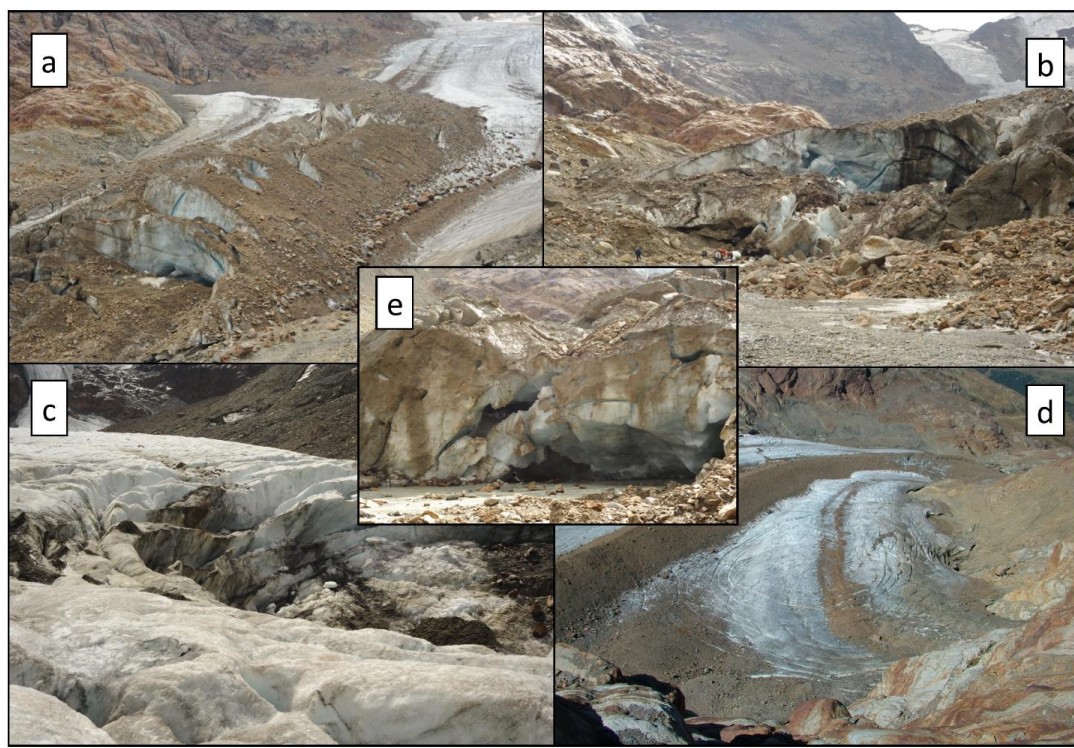

*Figure 2: Collapsing areas on the tongue of Forni Glacier. (a) Faults cutting across the eastern*
*medial moraine; (b) glacier terminus; (c) Near-circular collapsed area on the central tongue;*
*(d) Large ring fault on the eastern tongue at the base of the icefall. Photo courtesy of G.Cola; (e)*
*Close-up of a vertical ice cliff at the glacier terminus.*




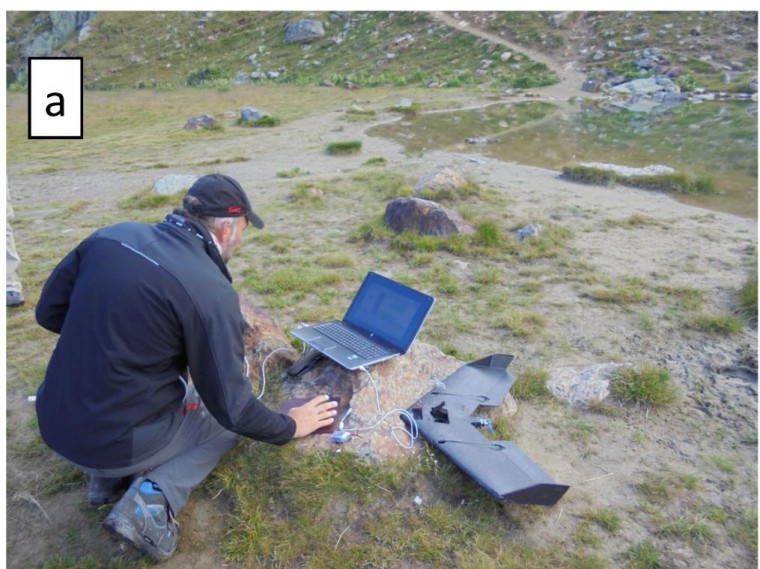

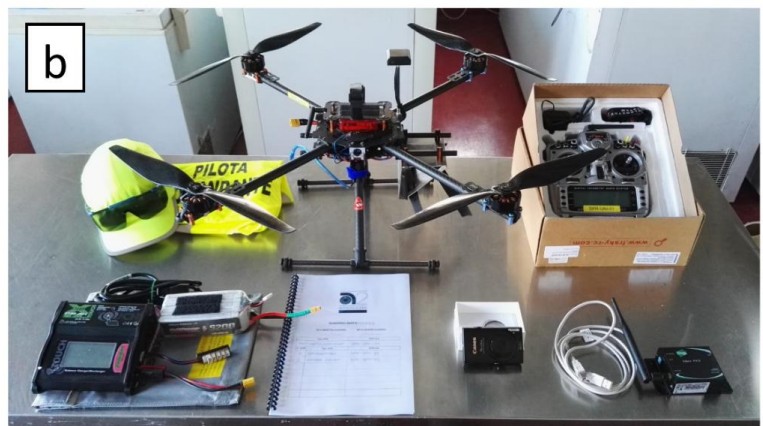


*Figure 3: The UAVs used in surveys of the Forni Glacier. (a) The SwingletCam fixed-wing aircraft*

*employed in 2014, at its take off site by Lake Rosole; (b) The quadcopter used in 2016 in the lab.*




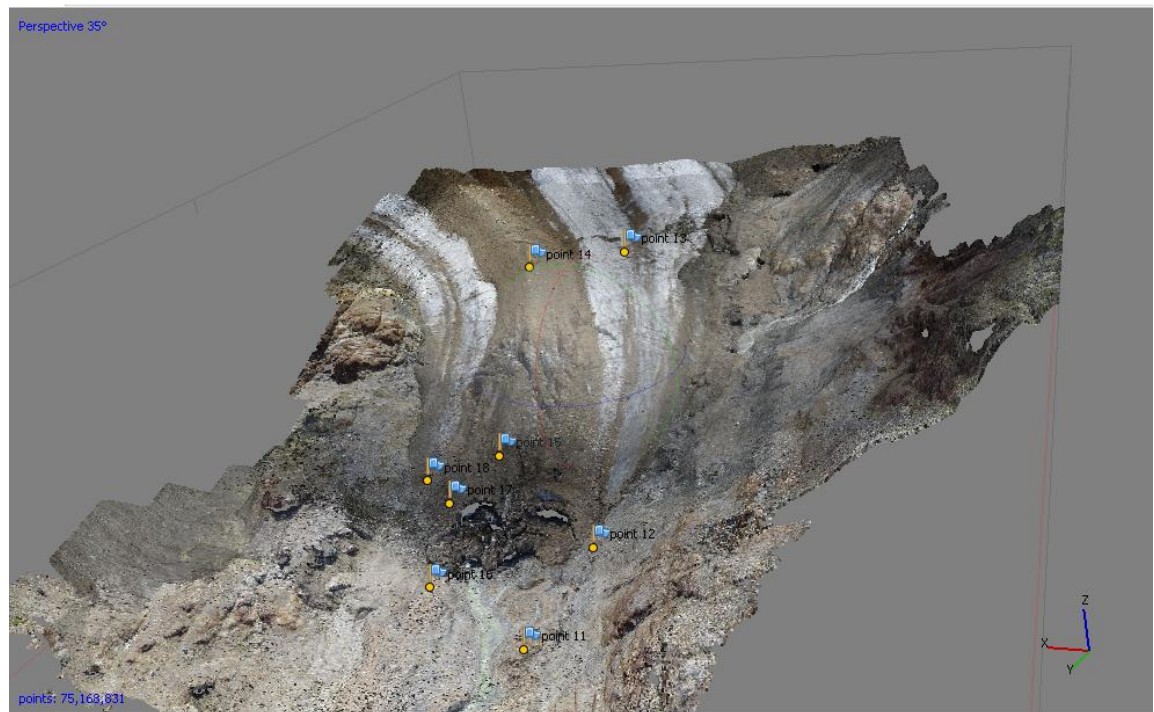


*Figure 4: Dense point cloud of the 2016 survey and location of the GCPs recorded with GNSS*
*equipment.*




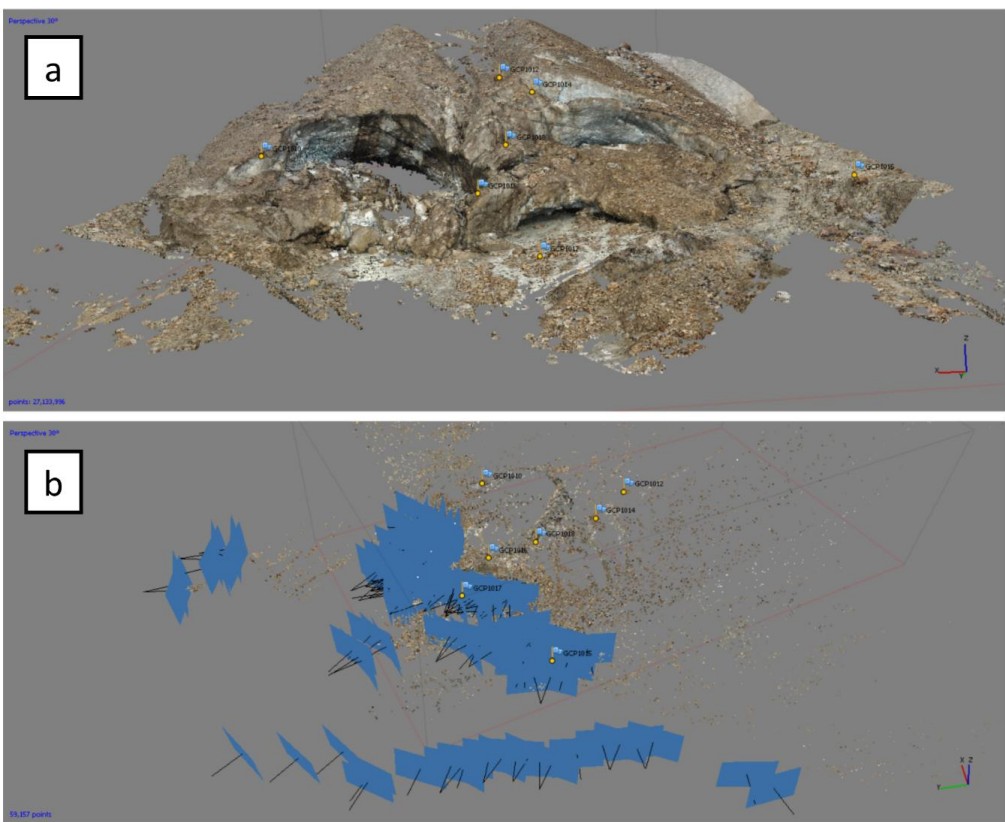


*Figure 5: 3D reconstruction of the glacier terminus using terrestrial photogrammetry: (a) locations of*
*camera stations in front of the glacier and 3D coordinates of tie points extracted during SfM for image*
*orientation; (b) point clouds of the glacier terminus with positions of adopted GCPs.*






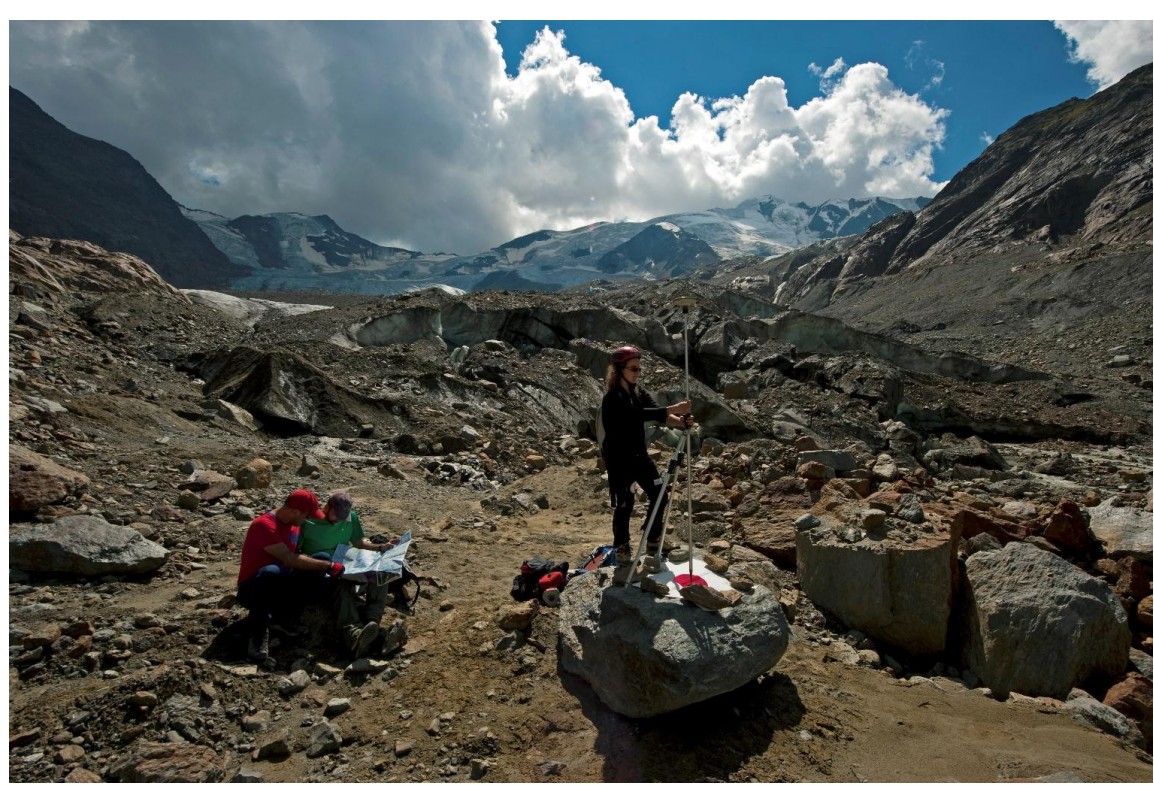

*Figure 6: survey operations of a GCP placed on a flat boulder on the proglacial plain of Forni Glacier.*
*Photo courtesy of Livio Piatta*




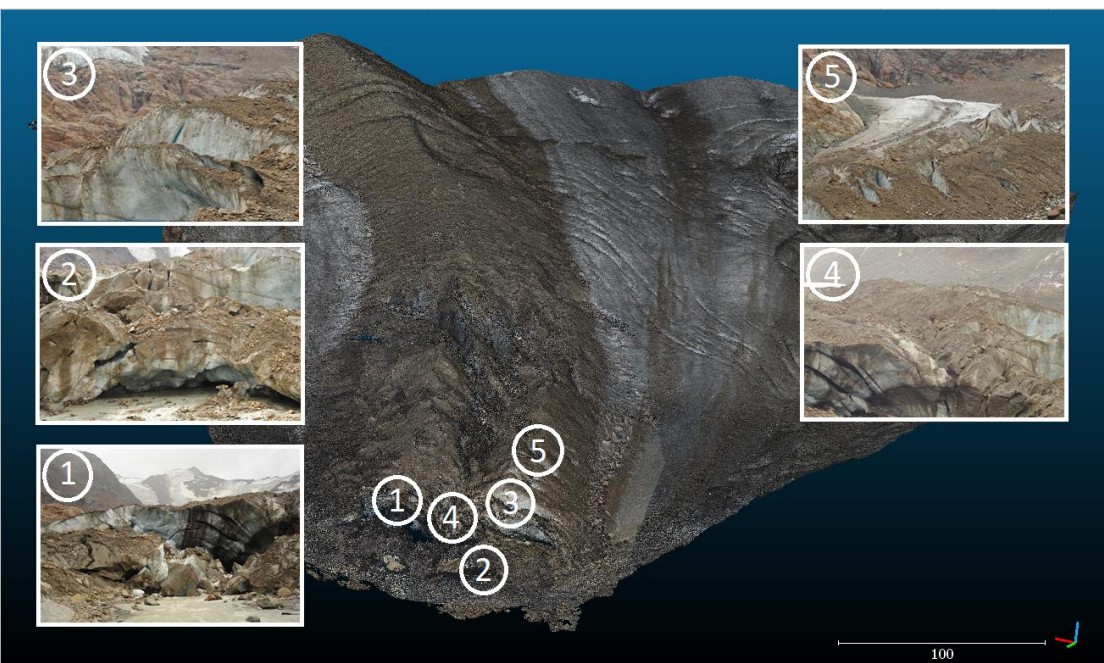

*Figure 7: Sample windows on the glacier terminus area.*



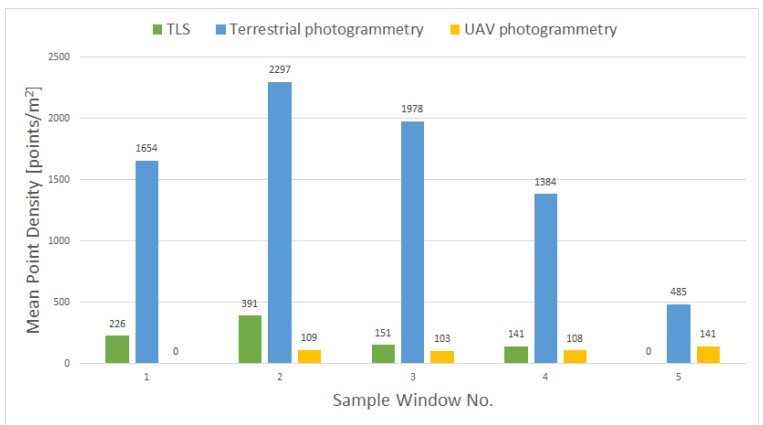


*Figure 8: Bar plot of mean point density computed in the five sample windows on the Forni Glacier*
*terminus.*





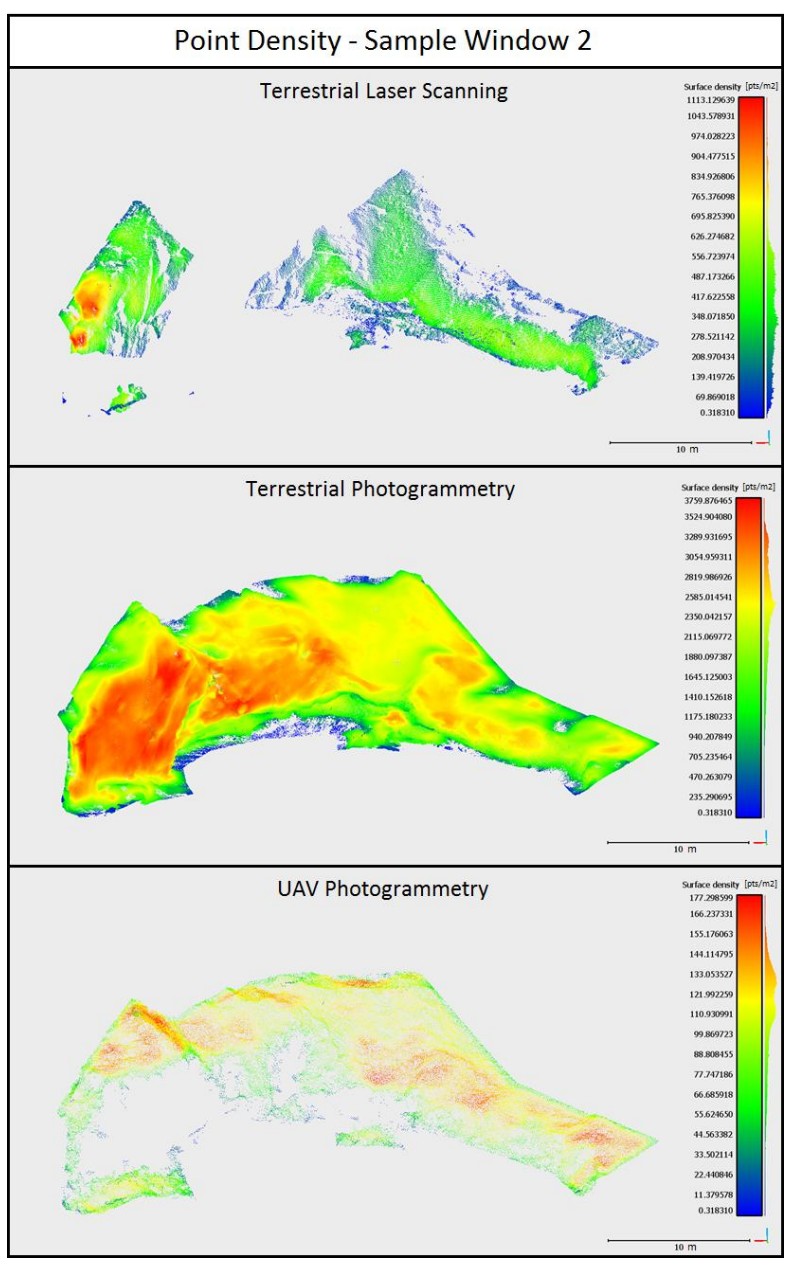


*Figure 9: Maps of point density for Window 2.*







Figure 10: Maps of point density for Window 3.





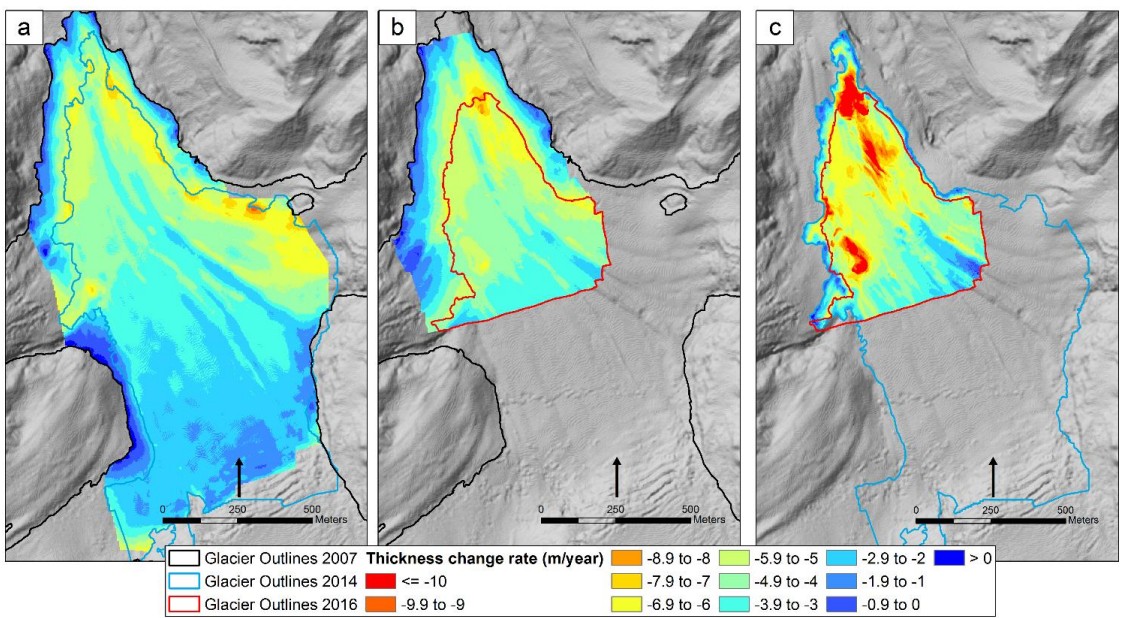

*Figure 11: Ice thickness change rates from DEM differencing over (a) 2007-2014; (b) 2007-2016; (c)*
*2014-2016. Glacier outlines from 2014 and 2016 are limited to the area surveyed during the UAV*
*campaigns. Base map from hillshading of 2007 DEM.*



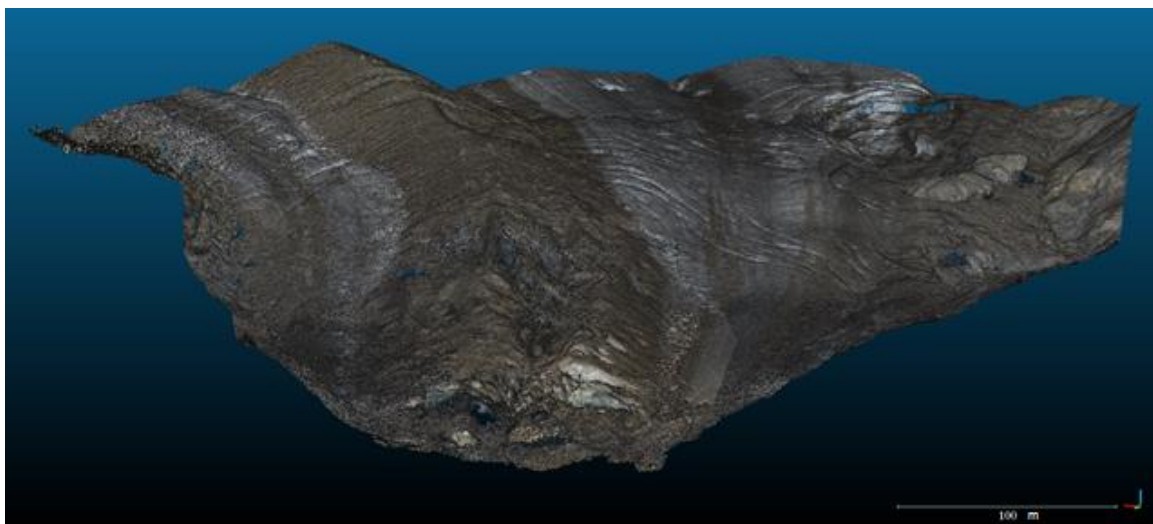


*Figure 12: Merged 3D model of the Forni Glacier tongue, integrating points clouds derived from UAV*
*and terrestrial photogrammetry, subsampled to keep a minimum distance between adjacent points of*
*20 cm, and coloured with RGB information from images.*


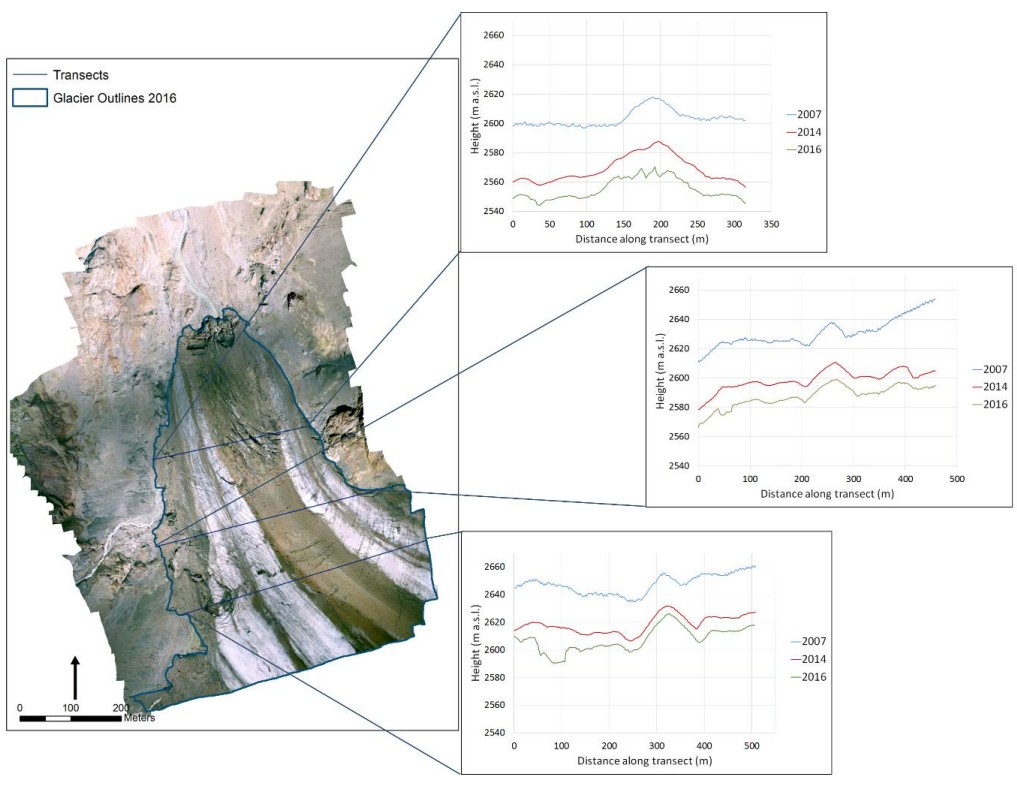

*Figure 13: Across-glacier transects of elevation of the ice surface in 2007, 2014 and 2016, based on*
*the respective DEMs. Base map is the orthomosaic obtained from the 2016 UAV survey.*





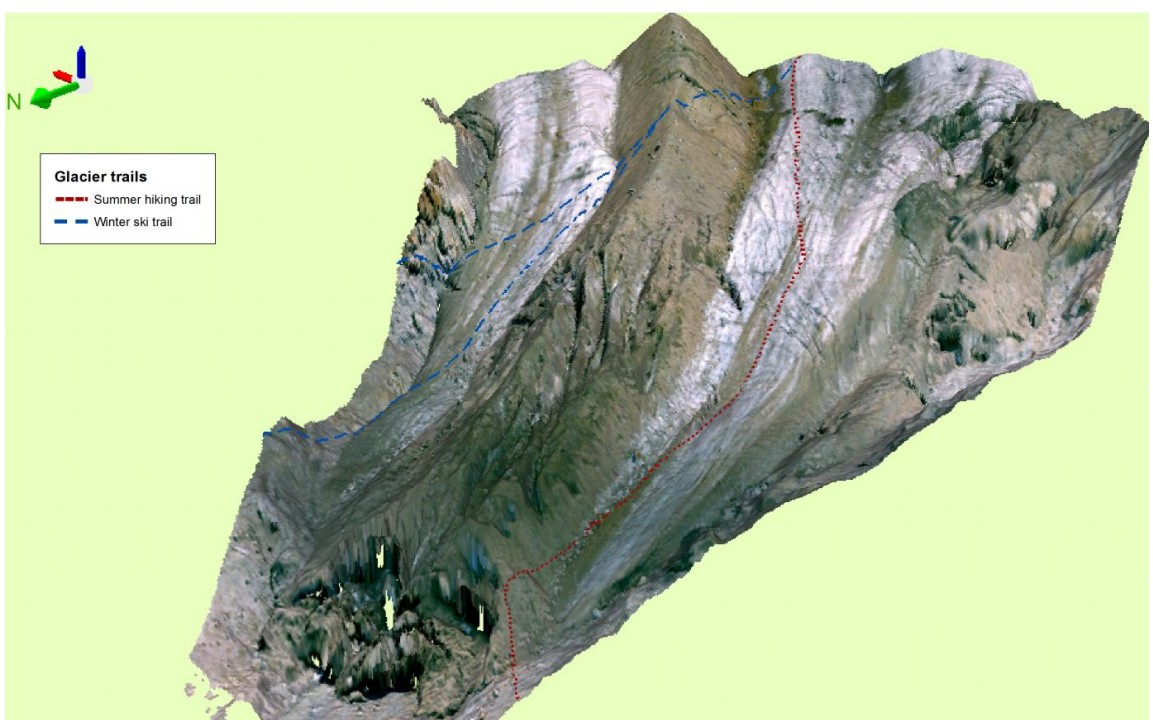


*Figure 14: perspective view of the glacier tongue showing summer and winter trails crossing the*
*glacier. Trails available from Kompass online cartography at https://www.kompass-*
*italia.it/info/mappa-online/. Elevation surface is the merged point cloud obtained from UAV and close-*
*range photogrammetry, with 2x vertical exaggeration.*





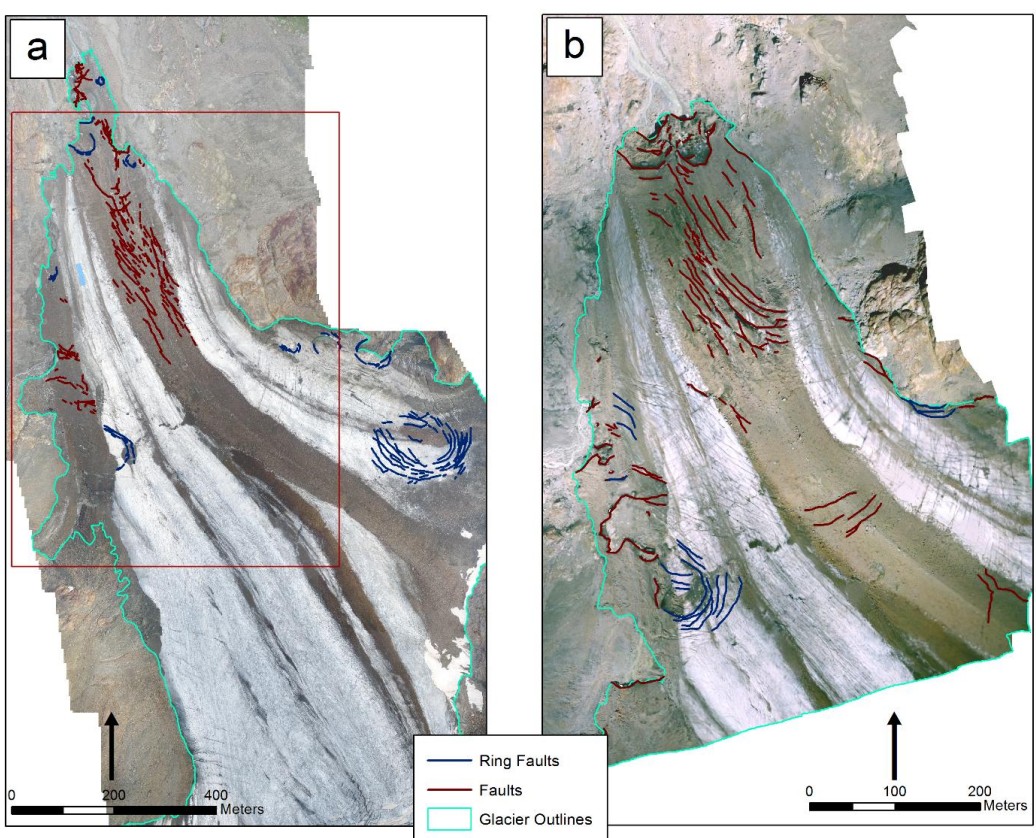


*Figure 15: location of collapse structures on the Forni Glacier, shown on the respective UAV*
*orthophoto. (a) 2014. The red box marks the area surveyed in 2016. (b) 2016.*