# Peer review of "Combination of UAV and terrestrial photogrammetry to assess rapid glacier evolution and conditions of glacier hazards"

_Natural Hazards and Earth System Sciences, 2017_

## Referee Comment (RC1) · Anonymous Referee #1 · 16 Jul 2017

**REVIEW OF THE MANUSCRIPT ENTITLED:**

**Combination of UAV and terrestrial photogrammetry to assess rapid glacier evolution and conditions of glacier hazards**

**By D. Fugazza et al.**

**General comments**

In this paper, Fugazza et al. present the results of photogrammetric surveys carried out on the lower ablation area of the Forni Glacier in 2014 and 2016. The surveys were performed using photographs taken from the ground and from unmanned aerial vehicles, and their results have been used for intercomparisons aimed at evaluating the accuracy of used techniques, for quantification of glacier changes across 9 years, and for the identification of hazards deriving from the current rapid shrinking of the glacier.

The work is interesting and potentially useful as a baseline for future developments of these remote sensing techniques. However, there are several points of the paper that require formal and substantial improvements. In particular:

i) there are several claims of uniqueness, originality, lack of previous work and of scientific knowledge, which are untrue and deserve a careful literature review by the authors. Consequently, the results are not unique, and have to be critically assessed in light of previous findings by other authors

ii) pros and cons of the tested methods require a thorough discussion, as well as their repeatability (e.g. the peculiar cloud cover conditions) and generalizability, costs, logistics and alternative solutions. A very weak point that requires discussion, in my opinion, is the limited areal extent of the surveyed zone, preventing possible applications aimed, for example, at the estimation of the glacier-wide geodetic mass balance. In addition, such a large glacier normally has several other hazardous areas along mountaineers' tracks, which cannot be comprehensively surveyed using the proposed approach. Which improvements (or alternative methods) would be required?

The paper is rather long and contains many descriptive sentences, too generic periods, scholastic explanations. In particular the Results section is difficult to read and wordy. My suggestion is to really focus on the results of the investigations, strongly shortening this part, and moving any (relevant) consideration in the Discussion section.

A careful English proof reading is also required to improve the readability and to make the manuscript appealing. The authors should also consider reducing the self-citations, which currently contribute to one third of the reference list.

In my opinion, the manuscript requires a major revision before being considered for publication in NHESS. A more complete description of the required formal and substantial improvements is reported in the following section.

**Detailed comments**

L30: snow cover thickness and/or duration?

L39: changes of glacier and…

L40: anthropic activities

L39-63: some of the mentioned processes are not strictly depending or worsened by climate variations. They are instead typical of the glacial, periglacial and paraglacial environments. I suggest rewriting this part to clarify which processes are typical and which ones are worsened by the current climatic phase. I also suggest mentioning debris flows

L68: please add some references concerning glacier change detection using DEMs

L72: remove indeed

L92: please replace battery support with e.g. battery life or charge duration

L110: must be completely observable from….

L114-118: previous work reporting comparison between photogrammetry and LiDAR or more traditional survey techniques on glaciers actually exists. Please, see for example Kaufmann and Ladstädter (2008), Piermattei et al., (2015 and 2016), Kaufmann and Seier (2016), Westoby et al., (2016), Seier et al., (2017), and contributions in the book from Pellikka and Gareth Rees (2010).

Kaufmann, V. and Seier, G., 2016. LONG-TERM MONITORING OF GLACIER CHANGE AT GÖSSNITZKEES (AUSTRIA) USING TERRESTRIAL PHOTOGRAMMETRY. International Archives of the Photogrammetry, Remote Sensing & Spatial Information Sciences, 41.

Kaufmann, V. and Ladstädter, R., 2008. Application of terrestrial photogrammetry for glacier monitoring in Alpine environments. Ele, 2700(2800), p.2900.

Petri Pellikka, W. Gareth Rees. Remote Sensing of Glaciers, Taylor & Francis Group, London, UK (2010)

Seier, G., Kellerer-Pirklbauer, A., Wecht, M., Hirschmann, S., Kaufmann, V., Lieb, G.K. and Sulzer, W., 2017. UAS-Based Change Detection of the Glacial and Proglacial Transition Zone at Pasterze Glacier, Austria. Remote Sensing, 9(6), p.549.

Westoby, M.J., Dunning, S.A., Hein, A.S., Marrero, S.M. and Sugden, D.E., 2016. Interannual surface evolution of an Antarctic blue-ice moraine using multi-temporal DEMs. Earth Surface Dynamics, 4(2), p.515.

L119: protected in which sense?

L121: two distinct aircrafts

L124: please provide a length for short and long time scales

L125-126: please try to improve this sentence, which is too long

L129: confluencing ice tongues?

L134: retreating rate? Shrinking rate?

L135-136: AWS1 Forni was not the first. There is at least one precedent, i.e. the AWS that has been operated on the Careser Glacier (10 km from Forni) from 1989 to 1998 (Rossi and Stojkovich, 1992; Novo and Rossi, 1998).

Rossi G. C. and Stojkovic, P. (1992) Scientific programmes for meteo-climatic and environmental observations in Alpine glacial areas. Presented at First Ev-K2-CNR Scientific Conf. on Scientific and Technological Research at High Altitude and Cold Regions. Milano, 10–11 April 1992.

Novo, A. and Rossi, G.C., 1998. A four-year record (1990–94) of snow chemistry at two glacier fields in the Italian Alps (Careser, 3090m; Colle Vincent, 4086m). Atmospheric Environment, 32(23), pp.4061-4073.

L142-155: I suggest shortening these points and possibly moving some of the concepts and references in the discussion section (if relevant). In section 1.2 all references are self-references. I wonder how much they are functional to this work.

L159: the meaning of reconstructing is not fully clear, I suggest writing explicitly that it is a topographic survey (also in the following)

L169-171: it is not clear why morning hours should be preferable to the central hours of the day. Please state it clearly

L171-173: it is not clear how low cloud cover inhibits direct solar radiation, it should be the contrary. Moreover, what to the authors mean with low cloud cover? Which fraction of the sky covered? By which type of clouds? And why should the direct solar radiation be avoided? How often can these ideal meteorological conditions be met in the alpine environment during summer? Which is the impact of ice ablation during the three-day survey period? Is there any measurement? In my opinion this information is of relevance for future applications and repeatability of the proposed survey techniques.

L177: potentially causing motion blur to the acquired imagery

L179: at a relatively low altitude of 50 m

L216: please consider replacing coordinate frame with coordinate system (also in the following)

L218: same days of the UAV survey?

L225: consider replacing pipeline with workflow

L245: evolves rapidly, or is rapidly evolving

L246: it is unclear why a complex shape and a rapid evolution make the glacier terminus not suitable for quantitative evaluation of the ice bulk? What do the authors mean with this sentence?

L249: including GCP surveying

L251: same days of the UAV survey?

L268: remove the purpose of

L274: how much stable has to be considered a GCP placed at the glacier surface, close to the terminus and for more than one day during the ablation season? Please discuss this issue

L284-285: with which consequences? Fewer than planned surveyed GCPs? Why not using post-processing correction?

L287: in my opinion it should be better arranging the methods in chronological order

L294-301: here is the explanation why early morning is preferable. Another reason for moving this part above the 2016 survey, according to me. What about cast shadows? Are they a further reason to avoid direct solar radiation and/or surveys carried out later in the day, with the possible occurrence of shadows from scattered cumulus clouds? What is the repeatability of this method if applied to east-exposed glaciers?

L332-340: this part has some repetitions from previous paragraphs. Please rephrase

L358: what about spatial trends in elevation differences? Are they inexistent, negligible or not taken into account?

L361-390: this part is too long and does not present results

L391-409: why not using the entire overlapping area? The area surveyed by terrestrial photogrammetry is already small, therefore I do not understand why the authors decided to perform a (subjective) sub-sampling taking very small areas, which on the other hand are very similar to each other. I suggest comparing the entire area in common among the different surveys, and then analyse separately glacier areas with peculiar characteristics

L410-411: please avoid describing in the text what figures and table present (their caption already does it)

L414-415: a more dense point cloud? The term consistent has a too general meaning (and here is misleading)

L415-417: the flexibility of terrestrial photogrammetry, compared to UAV photogrammetry, is questionable

L419-432: please summarize this part and avoid too scholastic sentence such as the first. I suggest simply stating which metrics are used and which results they provided

L437-443: here the authors skip to the concept of point cloud completeness, introducing a heuristic evaluation method that is not fully described. Afterwards, they resume with point density. My suggestion is to rearrange paragraphs in a more logic order

L445: please remove the sentence: The following general considerations can be made (and other analogous sentences in the manuscript).

L448: comparable or three times smaller?

L454-455: this is expected and confirmation of findings from previous works. Please add references

L455-458: which are the practical consequences? Which method for which application? Please discuss in the appropriate section

L469: this is another highly-expected result. However this is a very small area, compared to the entire tongue (or the entire glacier). Further considerations are required, e.g. in the discussion

L477: similar point densities were found

L484: the former are more suitable….

L486: please see comment L445. These sentences make the paper boring and difficult to read

L491-492: the suitability of a survey technique depends largely on the final aims of the survey. LiDAR DEMs obtained with point densities as low as 2 pt/m2 are enough for glacier-wide and/or regional scale glacier change assessments, for example. Please comment on that in the discussion

L493: please see comment L410

L496: please replace here and elsewhere "exposed upward" with horizontal, or sub-horizontal, or moderately sloping (maybe adding slope thresholds for improved understanding).

L516: the sections 3.1 and 3.1.1 are very long and can be highly summarized, presenting just the results and moving further considerations in the discussion section.

L518-523: I suggest removing or strongly summarizing this part

L523: do the authors have ablation measurements (or estimates) during the survey period? What is the impact of glacier ablation in calculations?

L540: retained or based on some metrics/methodological constrains?

L576: ΔDEM could be replaced by the more commonly-used dem of difference (DOD)

L575-579: this part is poorly written and hardly readable/understandable. Please reformulate

L579-581: this part is obvious and redundant

L593: please complete numbers with minus sign and measurement units

L594: the eastern part of the ablation tongue

L603-610: I am not fully convinced that the paper deserves section 3.3. My suggestion is to remove it and move concepts above, when the authors write about the complementarity of the two survey techniques.

L613-622: the authors try to validate their geodetic mass balance estimates in the lower glacier tongue, using specific mass balance estimations at the surface, for one point (whose location is not reported). Their approach is not correct, because they are comparing single-point vs. mean areal estimates, which can be highly different in the study area given the high lateral gradients in mass balance and elevation changes (Fig. 11), likely attributable to debris cover and differential ablation. Moreover, local geodetic and glaciological mass balance estimates seldom match on glaciers, because the surface elevation change is the result of a complex combination of surface, internal and basal mass exchanges, and of ice dynamics. In particular, vertical displacements (emergence velocity) have to be quantified for local comparisons of the two methods (see for example Fischer, 2011; Sold et al., 2013).

Fischer, A., 2011. Comparison of direct and geodetic mass balances on a multi-annual time scale. The Cryosphere, 5(1), p.107.

Sold, L., Huss, M., Hoelzle, M., Andereggen, H., Joerg, P.C. and Zemp, M., 2013. Methodological approaches to infer end-of-winter snow distribution on alpine glaciers. Journal of Glaciology, 59(218), pp.1047-1059.

L612-645: in my opinion this is not discussion, but mostly a presentation of results. Here the authors should discuss the accuracy of their results, the problematics in data collection and processing, the generalizability and the added values of the employed techniques. In particular, they should provide a discussion of the pros and cons of the proposed approaches, a comparison of their results with the existing literature, and critical evaluation of local-scale high-resolution surveys vs. glacier-wide surveys, which are required for geodetic mass balance estimates and comprehensive glacier hazard mapping. Which of the used methods has the highest potential for monitoring rapid glacier evolution and deriving hazards? Is there a method that has the potential to become a standard in glacier monitoring strategies, according to the authors? With which improvements/adjustments?

L687: I wonder if there is a more quantitative approach to be used here (such as DOD) to better exploit the new technologies. All the surface features described in this section are so large to be clearly visible by quick field observations and the tourist path can be easily changed accordingly. In my opinion the advantage of AUV and/or terrestrial photogrammetry lies in the possibility of automatically mapping and measuring these features from the DOD. Therefore, I suggest to add this quantitative assessments, starting from elevation changes as displayed in Fig. 11, where the collapse structures are evident.

L695: increased rate of surface lowering (not necessarily equal to surface ablation).

References: the reference list is rather long and, notably, one third of the references are self-citations. Please check if all these references are pertinent and functional to the paper

Table 2: please provide explanation for GSD

Table 3: I guess that the last column shows elevation differences "with" co-registration shifts

Table 8: I suggest showing in a figure the extent and location of the common reference area

Figure 5: I think that a) and b) are inverted

Figure 7, 12 and 14: these figures can be merged in a single image

---

## Referee Comment (RC2) · S. Gindraux (Referee) · 18 Jul 2017

Summary:

In this manuscript, the authors describe and analyse geomorphological features on the tongue of a hazard-prone glacier in the Italian alps with the help of different (close-range) remote sensing methods. They found that the merging of point clouds generated from two methods (UAV- and terrestrial photogrammetry) present the best product in order to map glacier hazards. The idea for this "data fusion" is new and potentially interesting, however it is not sufficiently described. The manuscript has nice Figures, well-displayed tables and is written in an easy-to-follow language style, that I appreci-

ated to read. However, sections are missing and there is a need for a re-shuffling work (i.e. put the information in the right sections). The authors also invested a lot of effort in the text by inserting a great deal of information but the manuscript is overall too long and needs shortening. This work on analysing glacier hazards for the population is surely valid but a stronger emphasis on its scientific relevance is needed. Due to these issues, I think this manuscript needs major revision.

General comments:

The next paragraphs of this review contain the general issues in each manuscript sections.

Introduction (Sec. 1):

- Better define the aim and workflow of your work:

The introduction section is constituted of three parts that are not well linked together. One of the main issue is that there is no clear "story" . I understand what the authors did in term of analysis but how they linked their results to the "evolution and conditions of glacier hazard" question mentioned in the title) was unclear to me. In order to better understand how the authors plan to use the remote sensing products in order to map (or analyse? This is also not clear) the different glacier hazards, including a dedicated method section would be very valuable.

- Link the paragraphs to prepare the reader and move information to other sections:

o In the first part, the two first paragraphs (Lines 27 to 63) explains changes of glacier and permafrost environment to climate change and gives examples of changes and hazards. The GLOFs are also mentioned and not mentioned again until the conclusion, which is unsettling for the reader. Maybe listing the glacier natural hazards that will be analysed in this study would be useful for the reader.

o In the section 1.1, the first paragraph (Lines 65 to 85) is on remote sensing and natural hazards monitoring (what the title promises), while the second (Lines 86 to

113) is on the general use of UAV on glaciers. These two subjects do not link together and the reader is not prepared to read the second paragraph. Maybe it would be good to include it in a new method section? This depends on the "story" you want to tell.

o The third paragraph of subsection 1.1 (Lines 114 to 126) does not include what the title suggests. Instead of a detailed text about remote sensing and glacier hazards, it identifies the research gap and the aim of the study. I suggest to merge everything (all introduction subsections) in one longer introduction and write a text that prepares the reader for the coming section (e.g. data,results,...), as well as states a clear research question and description of the methods used to answer it.

o In my opinion, reading about the study area (Subsection 1.2; Lines 127 to 155), in the introduction is very uncommon. I would merge it in another section (e.g. in the data section or in the new? Method section). This section however is too long and it should be shortened, containing only the information the reader needs to understand your work.

Data Sources: acquisition and processing (Sec. 2):

- Shorten the whole section:

A lot of information in this section is not crucial for the reader that gets lost. I suggest rewriting it in a more succinct way and remove text. See more details in the short comments.

- Re-order the subsections:

It is hard to follow this Sec. 2 because, the reader is starting to read a section about a new UAV survey, then terrestrial survey, TLS, control points (that belong to UAV and TLS), and finally a UAV survey again. I suggest that the different subsections should be divided per surveying method rather than the different datasets. For instance:

———————————————————

2.1 UAV photogrammetry

2.1.1 Dataset 2014

Content example: Type of UAV, flights, GCP network, software to generate products (and eventually workflow), resolution of end product.

2.1.2 Dataset 2016

2.2 Terrestrial photogrammetry

2.3 TLS

2.4 Aerial photogrammetric survey
* * *
Results (Sec. 3):

- Shorten and merge sections:

o The first part of the result section (subsection 3.1 and 3.11) is about statistics describing the point clouds, and is too long. The subsections could be merged and shortened, the number of tables and figures reduced. A large part of the text in these two sections also belong to the discussion section (see short comments).

o Some methodological description seems to be "hidden" in the result section. I suggest that the text is re-shuffled and shortened. More details can be found in the short comments.

o Part of the text in subsubsection 3.1.2 (Lines 517 to 570) belongs to the discussion section (see short comments for more details).

- Clarify "accuracy" and "comparison":

Subsubsection 3.1.2 (Lines 517 to 570), concerns the assessment of the point clouds' accuracy. In principle, the absolute accuracy of such point clouds can only be assessed with perfect validation data (e.g. long-term precise GPS data). Each method has its advantages and drawback and thus, generates products with different kind of errors (i.e. they are all differently imperfect/inexact). Therefore, the accuracy of a point cloud cannot be calculated using a point cloud generated from another method; but a comparison can be made. The analysis performed with the help of cloud compare, looks at the differences between the 3D geometry of point cloud pairs only. I would make a clear distinction and use of these terms in the text.

- Add information:

o It is not clear how the glacier thickness information (Section 3.2 and in general) is used in the assessment of glacier hazards. Could you please provide more information in the text?

o Subsection 3.3 (Lines 571 to 602) also requires more information on how this dataset merging has been made. A method section would be useful, especially when you cite this merging be the best product to monitor glacier hazards in the conclusion. This could be a very interesting point! And maybe the main novelty of this study and should better be highlighted.

o Subsection 3.3 (Lines 571 to 602) present the fusion of two point cloud datasets. It is very confusing for the reader to switch between point cloud (Section 3.1 and 3.3) and DEM sections (Section 3.2). Can you maybe change the section's order?

o It was very unclear to me after reading the results section, why the authors performed all these different analyses (i.e point cloud statistical analysis, point cloud accuracy, point cloud fusion and glacier thickness change), when at the end (Discussion section) you present a map of the glacier hazards (location of collapse, Fig.15) generated with the help of UAV orthophotos?. Could you please better explain their link in the introduction and method section?

Discussion (Sec. 4):

[Figure]

- Link the discussion to the result section:

The discussion section (Lines 611 to 687) is divided into two parts: One on the geomorphological evolution of the glacier tongue and the second about glacier-related hazards and how to risk is reduced through hazard mapping. Although the information is interesting, almost none of the discussion is based on the result section, and this is what the reader expects. Can you please change the text accordingly?

- Discuss your results by comparing them to results of other studies:

Comparing the different point clouds with a) statistical numbers, b) point density and c) completeness, and judging the best mapping method based on them, follow a correct method workflow and give good results but the later are not new. There are many papers that state the drawbacks of the surveying methods in a mountain terrain e.g. that the TLS data have a lot of "holes" and that the UAV data do not represent the vertical geometry well. I would consider making reference to them and compare your results.

Conclusion (Sec. 5):

- Shorten and clarify the main message:

The conclusion (Lines 688 to 730) are a mix of different sections, that are, presently, not well linked together. In particular, a clear conclusive message is missing. My advice would be to revise this part and to include, amongst other, a short summary for how and why this study has been done, which would help to present a better " overall story".

Short comments:

The short comments are listed in a supplement .pdf file.

Comments on Figures and Tables:

I generally enjoyed looking at the figures and tables. The colors, the size and the contast of the Figures are well chosen and their appearance encouraged me to read

the text. Hereafter are a few suggestions of changes.

Figure 1: I suggest to reduce the area of figure 1a and to merge it with Figure 1b (Only one figure for the glacier's location). Can you please specify what are the black outlines and from which year? The location of the TLS standpoint would also be valuable.

Figure 2: It would be helpful to see where these pictures are located on the glacier. Maybe enlarge the glacier on Figure 1 and set the letters (a-e) at the correct location? Or make a new overview map similar to Figure 7.

Figure 3: b) A more exhaustive caption (with UAV name) and presentation of the other objects would be useful. Other that, Figure 3 does not seems to add much information. Consider merging it with Table 1.

Figure 4: Many other figures in the manuscript display the glacier tongue. Would it be possible to put the GCP location on one of them instead of creating a new image just for this? Caption: Add UAV in the caption, such as: "of the 2016 UAV survey".

Figure 5: Please increase the resolution of the image so that the GCP numbers are readable. Consider specifying the year of this survey (2016). Moreover, it would be nice to twist the images so that they have the same view angle (e.g. that on both images the GCP12 is front and GCP10 right).

Figure 6: This is a nice but large image that does not give much information. If you want to show the GCP or measuring device, part of the image can be cropped and merged in Figure 3 or another one.

Figure 7: Please start numbering with 1 on the upper left corner. The background image could be brighter. Caption: Please elaborate (e.g. Location of different glacier features or hazard-prone areas on the tongue of Forni glacier were the point cloud comparison has been performed. The background image is the dense point cloud generated from the 2014 UAV survey).

Figure 8: Figure 8 display part of the information of Table 5. As it does not show new

information, consider removing it.

Figure 9 & 10: I think both images show the same information, so maybe remove one of them? Please enlarge the numbers on the scale bars.

Figure 12: This is the same image than the background image of Figure 7 right? Either remove it and refer the reader to Figure 7 instead of 12, or show an image where the reader can see the difference between the 2014, the 2016 and the merged point cloud.

Figure 15: Please explain the differences between the red and the blue lines on the glacier. Rewrite the caption so that not only a year is given. A "N" close to the arrow would give a meaning to the arrow itself! The year of the glacier outline should be mentioned.

Table 1: This is a nice summary table but most of the useful information are already in the text. The added value to the paper is minor. Consider removing this table or merge it with Figure 3.

Table 2: The # symbols should be removed or indicate that it means "numbers".

Table 3: The last column should display the elevation differences "with" co-registration right?. How do you explain that the standard deviation values are still of several meters? This should be discussed in the discussion section.

Table 4: For the #, same comment as for Table 2. The i, ii, iii are not necessary here, or define them. Giving a volume as size is very uncommon and I suggest using area (m2). Consider merging this table with Table 5.

Table 5: Please specify that the mean and standard deviation is calculated with a function computing local point density. Same note for i, ii and iii as above. Merge with Table 4.

Table 6: Caption: As it is, the reader does not understand what the M3C2 is. Please define so that every image can be understood as stand-alone.

Table 7: The information of Table 8 is more useful in the sense that we can compare the the mean thickness change etc. over the same area of interest (of 0.32 km2). I would not include Table 7 in the manuscript.

Table 8: Remove the last sentence. The reader will usually read the text if he/she wants more information ;-)

Please also note the supplement to this comment:
https://www.nat-hazards-earth-syst-sci-discuss.net/nhess-2017-198/nhess-2017-198-RC2-supplement.pdf

---

## Author Comment (AC1) · 24 Sep 2017

We have prepared a point by point response to the reviewer's comments. In the following text, reviewer's comments are reported as RC, our answers as AC.

RC REVIEW OF THE MANUSCRIPT ENTITLED: Combination of UAV and terrestrial photogrammetry to assess rapid glacier evolution and conditions of glacier hazards By D. Fugazza et al. General comments In this paper, Fugazza et al. present the results of photogrammetric surveys carried out on the lower ablation area of the Forni Glacier in 2014 and 2016. The surveys were performed using photographs taken from the ground and from unmanned aerial vehicles, and their results have been used for intercomparisons aimed at evaluating the accuracy of used techniques, for quantification of glacier changes across 9 years, and for the identification of hazards deriving from the current rapid shrinking of the glacier. The work is interesting and potentially useful as a baseline for future developments of these remote sensing techniques. However, there are several points of the paper that require formal and substantial improvements. In particular: i) there are several claims of uniqueness, originality, lack of previous work and of scientific knowledge, which are untrue and deserve a careful literature review by the authors. Consequently, the results are not unique, and have to be critically assessed in light of previous findings by other authors ii) pros and cons of the tested methods require a thorough discussion, as well as their repeatability (e.g. the peculiar cloud cover conditions) and generalizability, costs, logistics and alternative solutions. A very weak point that requires discussion, in my opinion, is the limited areal extent of the surveyed zone, preventing possible applications aimed, for example, at the estimation of the glacier-wide geodetic mass balance. In addition, such a large glacier normally has several other hazardous areas along mountaineers' tracks, which cannot be comprehensively surveyed using the proposed approach. Which improvements (or alternative methods) would be required? The paper is rather long and contains many descriptive sentences, too generic periods, scholastic explanations. In particular the Results section is difficult to read and wordy. My suggestion is to really focus on the results of the investigations, strongly shortening this part, and moving any (relevant) consideration in the Discussion section. A careful English proof reading is also required to improve the readability and to make the manuscript appealing. The authors should also consider reducing the self-citations, which currently contribute to one third of the reference list. In my opinion, the manuscript requires a major revision before being considered for publication in NHESS. A more complete description of the required formal and substantial improvements is reported in the following section.

AC Dear Reviewer, thank you for your comments. We have rewritten the introduction section reconsidering research gaps and aims of our work in view of the wider literature. We have also rewritten the discussion section entirely, comparing our results

with findings from previous studies and discussing advantages and disadvantages of the techniques used in this study, including the small size of the area investigated. We have greatly shortened the manuscript, with particular attention to the results section, by summarizing key points, and moving considerations to the discussion section. We have carefully proofread the manuscript to improve its clarity and appeal and reduced self-citations and the number of citations in general. The answers to your major and minor comments are provided below.

RC Detailed comments

RC Line 30 Page 1: snow cover thickness and/or duration?

AC We have deleted this sentence to shorten the introduction section.

RC Line 39 Page 2: changes of glacier and. . .

AC We have deleted this sentence to shorten the introduction section.

RC Line 40 Page 2: anthropic activities

AC We have deleted this sentence to shorten the introduction section.

RC Lines 39-63 Pages 2-3: some of the mentioned processes are not strictly depending or worsened by climate variations. They are instead typical of the glacial, periglacial and paraglacial environments. I suggest rewriting this part to clarify which processes are typical and which ones are worsened by the current climatic phase. I also suggest mentioning debris flows

AC: We have rewritten the introduction section which now has a sharper focus on glacier hazards, distinguishing between those typical of glacial environments and those that are worsened by climate variations. We also mentioned debris flows following your suggestion. The paragraph now reads: “Glacier and permafrost-related hazards can be a serious threat to humans and infrastructure in high mountain regions (Carey et al., 2014). The most catastrophic cryospheric hazards are generally related to the

outburst of water, either through breaching of moraine- or ice-dammed lakes or from the englacial or subglacial system, causing floods and debris flows. Ice avalanches from hanging glaciers can also have serious consequences for downstream populations (Vincent et al., 2015), as well as debris flows caused by the mobilization of accumulated loose sediment on steep slopes (Kaab et al., 2005a). Less severe hazards, but still particularly threatening for mountaineers are the detachment of seracs (Riccardi et al., 2010) or the collapse of ice cavities (Gagliardini et al., 2011; Azzoni et al., submitted). While these processes are in part typical of glacial and periglacial environments, there is evidence that climate change is increasing the likelihood of specific hazards (Kaab et al., 2005a). In the European Alps, accelerated formation and growth of proglacial moraine-dammed lakes has been reported in Switzerland, amongst concern of possible overtopping of moraine dams provoked by ice avalanches (Gobiet et al., 2014). Ice avalanches themselves can be more frequent as basal sliding is enhanced by the abundance of meltwater in warmer summers (Clague, 2013). Glacier and permafrost retreat, which have been reported in all sectors of the Alps (Smiraglia et al., 2015; Fischer et al., 2014; Gardent et al, 2014; Harris et al., 2009), are a major cause of slope instabilities which can result in debris flows, by debuttressing rock and debris flanks and promoting the exposure of unconsolidated and ice-cored sediments (Keiler et al., 2010; Chiarle et al., 2007). Glacier downwasting is also increasing the occurrence of structural collapses and while not directly threatening human lives, sustained negative glacier mass balance can also cause shortages of water for industrial, agricultural and domestic use and energy production, negatively affecting even populations living away from glaciers. Finally, glacier retreat and the increase in glacier hazards negatively impacts on the tourism sector and the economic prosperity of high mountain regions (Palomo, 2017)."

RC Line 68 Page 3: please add some references concerning glacier change detection using DEMs

AC: We have added "Fischer et al. (2015); Berthier et al. (2016)" accordingly.

RC Line 72 Page 3: remove indeed

AC: We have deleted this sentence to shorten the introduction section.

RC Line 92 Page 4: please replace battery support with e.g. battery life or charge duration

AC: We have deleted this sentence to shorten the introduction section.

RC: Line 110 Page 5: must be completely observable from. . ..

AC: We have deleted this sentence to shorten the introduction section.

RC: Lines 114-118 Page 5: previous work reporting comparison between photogrammetry and LiDAR or more traditional survey techniques on glaciers actually exists. Please, see for example Kaufmann and Ladstädter (2008), Piermattei et al., (2015 and 2016), Kaufmann and Seier (2016), Westoby et al., (2016), Seier et al., (2017), and contributions in the book from Pellikka and Gareth Rees (2010). Kaufmann, V. and Seier, G., 2016. LONG-TERM MONITORING OF GLACIER CHANGE AT GÖSSNITZKEES (AUSTRIA) USING TERRESTRIAL PHOTOGRAMMETRY. International Archives of the Photogrammetry, Remote Sensing & Spatial Information Sciences, 41. Kaufmann, V. and Ladstädter, R., 2008. Application of terrestrial photogrammetry for glacier monitoring in Alpine environments. Ele, 2700(2800), p.2900. Petri Pellikka, W. Gareth Rees. Remote Sensing of Glaciers, Taylor & Francis Group, London, UK (2010) Seier, G., Kellerer-Pirklbauer, A., Wecht, M., Hirschmann, S., Kaufmann, V., Lieb, G.K. and Sulzer, W., 2017. UAS-Based Change Detection of the Glacial and Proglacial Transition Zone at Pasterze Glacier, Austria. Remote Sensing, 9(6), p.549. Westoby, M.J., Dunning, S.A., Hein, A.S., Marrero, S.M. and Sugden, D.E., 2016. Interannual surface evolution of an Antarctic blue-ice moraine using multi-temporal DEMs. Earth Surface Dynamics, 4(2), p.515.

AC: Thank you for the interesting list of articles which we had overlooked. We have rewritten this paragraph, mentioning studies conducted with terrestrial and UAV photogrammetry in high mountain glacial environments. The paragraph now reads: "Recent years have seen a resurgence of terrestrial photogrammetric surveys for the generation of DEMs (Piermattei et al., 2015; Kaufmann and Seier, 2016) due to important technological advancements including the development of Structure-from-Motion (SfM) Photogrammetry and its implementation in fully automatic processing software, as well as the improvements in the quality of camera sensors (Westoby et al., 2012). In parallel, unmanned aerial vehicles (UAVs – Colomina & Molina, 2014, O'Connor et al., 2017) have started to emerge as a viable alternative to TLS for multi-temporal monitoring of small areas. UAVs promise to bridge the gap between field observations, notoriously difficult on glaciers, and coarser resolution satellite data (Bhardwaj et al., 2016a). Although the number of studies employing them in high mountain environments is slowly increasing (see e.g. Fugazza et al., 2015; Gindraux et al., 2016; Seier et al, 2017), their full potential for monitoring of glaciers and particularly glacier hazards has still to be explored. In particular, the advantages of UAV and terrestrial SfM-Photogrammetry, and the possibility of data fusion to support hazard management strategies in glacial environments needs to be investigated and assessed.".

RC Line 119 Page 5: protected in which sense?

AC: Forni Glacier lies in Stelvio National Park, which is a protected area under the Italian law. We have added "(Stelvio National Park)" to clarify this point.

RC Line 121 Page 5: two distinct aircrafts

AC: The words within parentheses have been deleted as suggested by the other reviewer.

RC: Line 124 Page 5: please provide a length for short and long time scales

AC: We have rewritten this sentence as "investigating ice thickness changes between 2014-2016 and 2007-2016 by comparing the two UAV DEMs and a third DEM obtained from stereo-processing of aerial photos captured in 2007." to clarify the length of scales

involved.

RC: Lines 125-126 Page 5: please try to improve this sentence, which is too long

AC: We have rewritten this sentence as: “identifying glacier-related hazards and their evolution between 2014-2016 using the merged point cloud from UAV and terrestrial photogrammetry and UAV orthophotos”. The description of glacier hazards mapped in this study was moved at the start of the paragraph.

RC Line 129 Page 6: confluencing ice tongues?

AC: This part has been deleted as suggested by reviewer 2.

RC Line 134 Page 6: retreating rate? Shrinking rate?

AC: The papers describe an acceleration in the shrinking rate. We have changed the sentence accordingly.

RC Lines 135-136 Page 6: AWS1 Forni was not the first. There is at least one precedent, i.e. the AWS that has been operated on the Careser Glacier (10 km from Forni) from 1989 to 1998 (Rossi and Stojkovich, 1992; Novo and Rossi, 1998). Rossi G. C. and Stojkovic, P. (1992) Scientific programmes for meteo-climatic and environmental observations in Alpine glacial areas. Presented at First Ev-K2-CNR Scientific Conf. on Scientific and Technological Research at High Altitude and Cold Regions. Milano, 10–11 April 1992. Novo, A. and Rossi, G.C., 1998. A four-year record (1990–94) of snow chemistry at two glacier fields in the Italian Alps (Careser, 3090m; Colle Vincent, 4086m). Atmospheric Environment, 32(23), pp.4061-4073.

AC: The sentences concerning the AWS have been deleted as not strictly relevant to this study.

RC: Lines 142-155 Page 6: I suggest shortening these points and possibly moving some of the concepts and references in the discussion section (if relevant). In section 1.2 all references are self-references. I wonder how much they are functional to this

work.

AC: We have deleted the sentences about recent glacier changes, the AWS (see previous comment) and previous research on the site (and related references), deleted bullet points and merged their content with the previous sentence, as suggested by reviewer 2. The paragraph now reads: "The Forni Glacier (see Fig. 1) has an area of 11.34 km2 based on the 2007 data from the Italian Glacier Inventory (Smiraglia et al., 2015), an altitudinal range between 2501 and 3673 m a.s.l. and a North-North-Westerly aspect. The glacier retreated markedly since the little ice age (LIA), when its area was 17.80 km2 (Diolaiuti & Smiraglia, 2010), with an acceleration of the shrinking rate in the last three decades, typical of valley glaciers in the Alps (Diolaiuti et al, 2012, D'Agata et al; 2014). It has also undergone profound changes in dynamics in recent years, including the loss of ice flow from the eastern accumulation basin towards its tongue and the evidence of collapsing areas on the eastern tongue (Azzoni et al., submitted). One such area, hosting a large ring fault (see Fig. 2d) prompted an investigation carried out with Ground Penetrating Radar (GPR) in October 2015, but little evidence of a meltwater pocket was found under the ice surface (Fioletti et al., 2016). Since then, a new ring fault appeared on the central tongue, and the terminus underwent substantial collapse (see Fig. 2a,b,c,e). Continuous monitoring of these hazards is important as the site is highly touristic (Garavaglia et al., 2012), owing to its location in Stelvio Park, one of Italy's major protected areas, and its inclusion in the list of geosites of Lombardy region (see Diolaiuti and Smiraglia, 2010). The glacier is in fact frequently visited during both summer and winter months. During the summer, hikers heading to Mount San Matteo take the trail along the central tongue, accessing the glacier through the left flank of the collapsing glacier terminus. During wintertime, ski-mountaineers instead access the glacier from the eastern side, crossing the medial moraine and potentially collapsed areas there (see Fig. 1). "

RC Line 159 Page 7: the meaning of reconstructing is not fully clear, I suggest writing explicitly that it is a topographic survey (also in the following)

AC: We have revised in the paper the use of the term “reconstruction,” changing or integrating this word to make it clear we mean the generation of a faithful digital 3D model of the object. The term “topographic survey” is generally used when geodetic methods are applied, so we preferred to use of “3D surface reconstruction”, “3D modelling” or “point cloud acquisition.”

RC Lines 169-171 Page 7: it is not clear why morning hours should be preferable to the central hours of the day. Please state it clearly.

AC: The explanation is provided in the description of the 2014 survey, which has now been moved to the top of the data section as suggested by you and the other reviewer.

RC Lines 171-173 Page 7: it is not clear how low cloud cover inhibits direct solar radiation, it should be the contrary. Moreover, what to the authors mean with low cloud cover? Which fraction of the sky covered? By which type of clouds? And why should the direct solar radiation be avoided? How often can these ideal meteorological conditions be met in the alpine environment during summer? Which is the impact of ice ablation during the three-day survey period? Is there any measurement? In my opinion this information is of relevance for future applications and repeatability of the proposed survey techniques.

AC: The reason why direct solar radiation should be avoided is that it can lead to image saturation on highly reflective surfaces such as ice or snow, as explained in the paragraph describing the 2014 survey. On both surveys in 2016, the weather was too unstable in the morning (i.e. chance of rain). When we actually undertook the surveys, the sky was overcast, i.e. 8/8 of the sky were covered by stratocumulus clouds. We thus found that these peculiar cloud cover conditions are suitable for UAV flights, while also common in Alpine environments during the summer. We have rephrased this sentence as “both around midday with 8/8 of the sky covered by stratocumulus clouds” and further discuss meteorological conditions in the discussion section where we have added a paragraph that reads:”We conducted UAV surveys under different

meteorological scenarios, and obtained adequate results with early-morning operations with 0/8 cloud cover and midday flights with 8/8 cloud cover. Both scenarios can provide diffuse light conditions allowing to collect pictures suitable for photogrammetric processing, but camera settings need to be carefully adjusted beforehand (O'Connor et al., 2017). If early morning flights are not feasible in the study area for logistical reasons or when surveying east-exposed glaciers, the latter scenario should be considered.". As concerns the impact of ice ablation, measurements from ablation stakes collected in summers 2009-2010 (Senese et al., 2012) and 2015 (unpublished data) indicate values of 3-5 cm day-1. Additionally, Ice flow ranges between 1 and 4 cm day-1 (Urbini et al., 2017). This mostly affected the photogrammetric reconstruction of the UAV dataset from 2016 as surveys were performed two days apart and the last one 3 days since GCP placement, and the comparison between the UAV point cloud and other techniques. Measurements of the vertical displacement of stakes taken with GNSS in 2006 also show similar values ranging between 2.8 and 4.6 cm day-1 (unpublished data). We can thus hypothesize a combined effect on the uncertainty of UAV photogrammetric reconstruction between 10 and 20 cm, and lower on GCPs as they were placed on boulders where ablation is reduced. have added a paragraph in the discussion section that reads: "In this study, the uncertainty of the 2016 UAV dataset (40.5 cm RMSE on GCPs and 21.1-37.7 cm RMSE when compared against TLS) was slightly higher than previously reported in high mountain glacial environments (Immerzeel et al., 2014; Gindraux et al, 2017; Seier et al., 2017). Contributing factors might include the sub-optimal distribution and density of GCPs (Gindraux et al., 2017), the delay between the UAV surveys as well as between UAV and other surveys and the lack of coincidence between GCP placement and the UAV flights. This means the UAV photogrammetric reconstruction was affected by ice ablation and glacier flow, which on Forni Glacier range between 3-5 cm day-1 (Senese et al., 2012) and 1-4 cm day-1, respectively (Urbini et al., 2017). We thus expect a combined 3-day uncertainty on the 2016 UAV dataset between 10 and 20 cm, and lower on GCPs considering reduced ablation owing to their placement on boulders. A further contribution to the error

budget of GCPs might stem from the intrinsic precision of GNSS/theodolite measurements and image resolution. The comparison between close-range photogrammetry, and TLS, being only one day apart, was less affected by glacier change and the RMSE of 6-10.6 cm is in line with previous findings by Kaufmann and Landstaedter (2008). To improve the accuracy of UAV photogrammetric blocks, a better distribution of GCPs or switching to an RTK system should be considered, while close-range photogrammetry could benefit from measuring a part of the photo-stations as proposed in Forlani et al. (2014), instead of placing GCPs on the glacier surface.".

RC Line 177 Page 8: potentially causing motion blur to the acquired imagery

AC: This sentence was shortened as suggested by Reviewer 2.

RC Line 179 Page 8: at a relatively low altitude of 50 m

AC: By "Relative" in this sentence we meant "relative to ground". We have rephrased the sentence to clarify this point, from "with flights at low relative altitude of 50 m" to "with a flying altitude of 50 m above ground"

RC Line 216 Page 9: please consider replacing coordinate frame with coordinate system (also in the following)

AC: We have replaced the word frame with system accordingly throughout the manuscript.

RC: Line 218 Page 10: same days of the UAV survey?

AC: We have specified the exact dates when the surveys took place. We discuss potential issues due to ice ablation between surveys in the discussion section, as described in the answer to your comment at lines 171-173.

RC Line 225 Page 10: consider replacing pipeline with workflow

AC: We have replaced this word accordingly.

RC Line 245 Page 11: evolves rapidly, or is rapidly evolving

AC: This sentence was deleted as it was connected to the following one, see next comment.

RC Line 246 Page 11: it is unclear why a complex shape and a rapid evolution make the glacier terminus not suitable for quantitative evaluation of the ice bulk? What do the authors mean with this sentence?

AC: We have removed this sentence as it lacked clarity and was unnecessary for the reader.

RC Line 249 Page 11: including GCP surveying

AC: We have modified the manuscript accordingly. The sentence was moved to the Discussion section as suggested by Reviewer 2.

RC: Line 251 Page 11: same days of the UAV survey?

AC: The TLS survey was conducted on the same day as the first UAV survey. We have added "On the same days as the first UAV survey of 2016," at the start of the paragraph to clarify this point.

RC Line 268 Page 11: remove the purpose of

AC: We have modified the manuscript accordingly

RC Line 274 Page 12: how much stable has to be considered a GCP placed at the glacier surface, close to the terminus and for more than one day during the ablation season? Please discuss this issue

AC: All GCPs at the terminus were actually located on large boulders, whereas only one GCP at the highest site on the central part of the tongue was placed directly on the glacier surface. Large boulders are known to shield the underlying ice from ablation, often leading to the formation of glacier tables. Thus, the effect of ice ablation on

GCPs is reduced. We have added a paragraph concerning this issue in the discussion section, see the answer to your comment at lines 171-173.

RC: Lines 284-285 Page 12: with which consequences? Fewer than planned surveyed GCPs? Why not using post-processing correction?

AC: Two of the points had to be collected and post-processed in fast-static mode due to the loss of radio connection. This effect could have not been planned in advance. We have added: “Non-RTK points were processed in fast-static mode, requiring a longer measurement time of approx. 12 minutes. ”

RC Line 287 Page 13: in my opinion it should be better arranging the methods in chronological order

AC: The paragraph about the 2014 survey was moved to the top of the data section, in accordance with your comment and the other reviewer’s. We have followed the other reviewer’s suggestion as to the order of the data section, leaving the 2007 aerial photogrammetric data at the bottom as it is the only dataset we did not collect ourselves and we believe it should be separated from the others.

RC Lines 294-301 Page 13: here is the explanation why early morning is preferable. Another reason for moving this part above the 2016 survey, according to me. What about cast shadows? Are they a further reason to avoid direct solar radiation and/or surveys carried out later in the day, with the possible occurrence of shadows from scattered cumulus clouds? What is the repeatability of this method if applied to east-exposed glaciers?

AC: We did not experience cast shadows from cumulus clouds during the 2016 survey. Based on our experience with UAVs, it is generally possible to adjust camera settings (ISO, aperture and shutter speed) before each flight to account for different light conditions, and produce pictures that are suitable for photogrammetric processing (see also O’Connor et al., 2017), although cast shadows will decrease the image dynamic range

and might complicate the matching process owing to the lack of contrast. As a rule of thumb, early morning flights with 0/8 cloud cover might generate the best images for photogrammetric processing, but are not always possible due to logistical constraints and meteorological conditions. As you also mention, it might not be possible to obtain these conditions when monitoring east-exposed glaciers. However, we also demonstrate how UAV flights with overcast conditions under stratocumulus clouds produce suitable images. We have added a paragraph where we discuss issues related to meteorological conditions in the discussion section, as described in the answer to your comment at lines 171-173.

RC Lines 332-340 Page 15: this part has some repetitions from previous paragraphs. Please rephrase

AC: This part was deleted to avoid repetitions.

RC Line 358 Page 16: what about spatial trends in elevation differences? Are they inexistent, negligible or not taken into account?

AC: We did not take into account spatial trends in elevation differences but when calculating the uncertainty of volume changes, we assumed the uncertainty of elevation differences as totally correlated in space. This is unlike other approaches where errors in elevation differences are assumed as random and the final uncertainty of volume change is smaller (Fischer et al., 2015). Thus, our estimates of volume change uncertainty represent a worst-case scenario.

Fischer, M.; Huss, M. and Hoelzle, M (2015). Surface elevation and mass changes of all Swiss glaciers 1980–2010, The Cryosphere, 9, 525-540

RC Lines 361-390 Page 16: this part is too long and does not present results

AC: This part has been condensed as follows: “The analysis of point clouds generated during the 2016 campaign had the aim of assessing their geometric quality before their application for the analysis of hazards. These evaluations were also expected to

provide some guidelines for the organization of future investigations in the field at the Forni Glacier and in other Alpine sites. ” and moved to subsection 4.1 in the methods section.

RC Lines 391-409 Page 17: why not using the entire overlapping area? The area surveyed by terrestrial photogrammetry is already small, therefore I do not understand why the authors decided to perform a (subjective) sub-sampling taking very small areas, which on the other hand are very similar to each other. I suggest comparing the entire area in common among the different surveys, and then analyse separately glacier areas with peculiar characteristics

AC: We have considered this suggestion. However, there are some independent registration errors in the data sets from UAV photogr., terrestrial photogr. and TLS. While these errors do not have any influence when analysing the point density and completeness, they do when computing the distances between point clouds. Therefore, we preferred to perform this analysis in individual sample locations, so that the errors due to registration could be compensated by a local refinement of the co-registration between point clouds. We have therefore rewritten the paragraph about the comparison between point clouds as: “Finally, we compared the point clouds in a pairwise manner within the same sample locations. Since no available benchmarking data set (e.g. accurate static GNSS data) was concurrently collected during the 2016 campaign, the TLS point cloud was used as a reference, as it less influenced by controlling factors (network geometry, object texture, lighting conditions). When comparing both photogrammetric data sets, the one obtained from UAV was used as reference because of the even distribution of point density within the sample locations. The presence of residual, non-homogenous geo-referencing errors in the data sets required a specific fine registration of each individual sample location, which was conducted in CloudCompare using the ICP algorithm (Pomerleau et al., 2016). Then, point clouds in corresponding sample areas were compared using the M3C2 algorithm implemented in CloudCompare (Lague et al., 2013). This solution allowed us to get rid of registration

errors from the analysis, which could then be focused on the capability of the adopted techniques to reconstruct the local geometric surface of the glacier in an accurate way." and moved it to the methods section.

RC Lines 410-411 Page 18: please avoid describing in the text what figures and table present (their caption already does it)

AC: This sentence has been removed accordingly.

RC Lines 414-415 Page 18: a more dense point cloud? The term consistent has a too general meaning (and here is misleading)

AC: We have modified the manuscript accordingly.

RC Lines 415-417 Page 18: the flexibility of terrestrial photogrammetry, compared to UAV photogrammetry, is questionable

AC: We have modified the sentence as follows: "Considering point density, terrestrial photogrammetry resulted in a denser data set than the other techniques. This is mostly motivated by the possibility to acquire data from several stations with this methodology, only depending on the terrain accessibility, reducing the effect of occlusions with a consequently more complete 3D modelling. ". The sentence was also moved to the discussion section.

RC Lines 419-432 Pages 18-19: please summarize this part and avoid too scholastic sentence such as the first. I suggest simply stating which metrics are used and which results they provided.

AC: This part has been shortened as follows: "Specifically, we analysed point density (points/m2) and completeness, i.e. % of area in the ray view angle. Point density partly depends upon the adopted surveying technique, since it is controlled by the distance between sensor and surface and the obtainable spatial resolution. In SfM-Photogrammetry, the latter property is affected by dense matching, while in TLS it can be set up as data acquisition input parameter. In this study, the number of neighbours

N (inside a sphere of radius R=1 meter) divided by the neighbourhood surface was used to evaluate the local point density D in CloudCompare (www.cloudcompare.org). To understand the effect of point density dispersion (Teunissen, 2009), the inferior 12.5 percentile of the standard deviation of point density was also calculated. The use of these local metrics allowed to distinguish between point density in different areas, since this may largely change from one portion of surface to another. A further metric in this sense was point cloud completeness, referring to the presence of enough points to completely describe a portion of surface. In this study, the visual inspection of selected sample locations was used to identify occlusions and areas with lower point density."

RC Lines 437-443 Page 19: here the authors skip to the concept of point cloud completeness, introducing a heuristic evaluation method that is not fully described. Afterwards, they resume with point density. My suggestion is to rearrange paragraphs in a more logical order

AC: This paragraph has been reorganized by introducing first point density and then point completeness. Visual inspection of the sample locations was used to identify areas of occlusions or with lower point density. We now specify this in the text as: "A further metric in this sense was point cloud completeness, referring to the presence of enough points to completely describe a portion of surface. In this study, the visual inspection of selected sample locations was used to identify occlusions and areas with lower point density."

RC Line 445 Page 20: please remove the sentence: The following general considerations can be made (and other analogous sentences in the manuscript).

AC: We have removed this and similar sentences throughout the manuscript.

RC Line 448 Page 20: comparable or three times smaller?

AC: We have modified these sentence as follows since Table 5 already displays the results: "Terrestrial photogrammetry featured the highest point density, while UAV photogrammetry had the lowest.”.

RC Lines 454-455 Page 20: this is expected and confirmation of findings from previous works. Please add references

AC: We have reorganized this part of the text, which has been moved in the Discussion Section. This sentence has been modified as follows: “Since any techniques may perform better when the surface to survey is approximately orthogonal to the sensor looking direction, terrestrial photogrammetry is more efficient for reconstructing vertical and subvertical cliffs (Sample areas 1 and 2) and high-sloped surfaces (Sample areas 3 and 4). On the contrary, airborne UAV photogrammetry provided the best results in location 5 which is less inclined and consequently could be well depicted in vertical photos. In general, point clouds from terrestrial photogrammetry provide a better description of the vertical and subvertical parts (see e.g. Winkler et al., 2012), while point clouds obtained from UAV photogrammetry are more suitable to describe the horizontal or sub-horizontal surfaces on the glacier tongue and periglacial area (Seier et al., 2017), unless the camera is tilted to an off-nadir viewpoint (Dewez et al., 2016; Aicardi et al., 2016). ”

RC Lines 455-458 Page 20: which are the practical consequences? Which method for which application? Please discuss in the appropriate section

AC: These sentences have been moved in the Discussion Section, where the optimal type of terrain per each method is described. The main practical consequence is that to have an exhaustive 3D model of the whole surface topography, both point clouds from terrestrial and UAV photogrammetry should to be merged. We have thus added a sentence in the Discussion section, which describes our practical suggestions and reads: “While our integrated approach using a multicopter and terrestrial photogrammetry should be preferred to investigate small individual ice bodies, fixed-wing UAVs, ideally equipped with an RTK system and ability to tilt the camera off-nadir, might be the platform of choice to cover large distances (see e.g. Ryan et al., 2017), potentially

reducing the number of flights and solving issues with GCP placement. ”

RC Line 469 Page 21: this is another highly-expected result. However this is a very small area, compared to the entire tongue (or the entire glacier). Further considerations are required, e.g. in the discussion.

AC: This part was condensed as suggested by Reviewer 2, as: “The analysis of point density shows significant differences between the three techniques for point cloud generation (see Table 2). Values range from 103 to 2297 points/m2 depending on the surveying method, but the density was generally sufficient for the reconstruction of the different surfaces shown in Fig. 5, except for location 5. Terrestrial photogrammetry featured the high point density, while UAV photogrammetry had the lowest. ”. We have added further considerations in the Discussion section as follows: “In our pilot study, we covered part of the Forni glacier tongue, and only investigated hazards related to the glacier collapse. Our maps can help identify safer paths where mountaineers and skiers can visit the glacier and reach the most important summits. However, the increase in collapse structures owing to climate change requires multi-temporal monitoring. A comprehensive risk assessment should also cover the entire glacier, to investigate the probability of serac detachment and provide an estimate of the glacier mass balance with the geodetic method. ”

RC Line 477 Page 21: similar point densities were found

AC: We have modified the manuscript accordingly.

RC Line 484 Page 21: the former are more suitable. . ..

AC: We have rephrased this sentence as described in the answer to your comment at lines 454-455

RC Line 486 Page 21: please see comment L445. These sentences make the paper boring and difficult to read

AC: We have modified the manuscript as follows: “ In relation to TLS, a mean value

of point density ranging from 141-391 points/m2 was found, with the only exception of location 5, where no sufficient data were recorded due to the position of this region with respect to the instrumental standpoint."

RC Lines 491-492 Page 22: the suitability of a survey technique depends largely on the final aims of the survey. LiDAR DEMs obtained with point densities as low as 2 pt/m2 are enough for glacier-wide and/or regional scale glacier change assessments, for example. Please comment on that in the discussion

AC: These lines belong to a part that has been cancelled to shorten the manuscript. We discuss the suitability of the techniques employed in our study in the discussion section. While it is outside the scope of this manuscript to present a comprehensive comparison of aerial LiDAR vs UAV for natural hazard management and glaciological purposes, UAVs have already been used to cover distances up to 280 km2, e.g. by Ryan et al. (2017). We thus believe they could eventually replace this technique for the purposes mentioned in the study. We have added a paragraph in the Discussion section that reads: "In our pilot study, we covered part of the Forni glacier tongue, and only investigated hazards related to the glacier collapse. Our maps can help identify safer paths where mountaineers and skiers can visit the glacier and reach the most important summits. However, the increase in collapse structures owing to climate change requires multi-temporal monitoring. A comprehensive risk assessment should also cover the entire glacier, to investigate the probability of serac detachment and provide an estimate of the glacier mass balance with the geodetic method. While our integrated approach using a multicopter and terrestrial photogrammetry should be preferred to investigate small individual ice bodies, fixed-wing UAVs, ideally equipped with an RTK system and ability to tilt the camera off-nadir, might be the platform of choice to cover large distances (see e.g. Ryan et al., 2017), potentially reducing the number of flights and solving issues with GCP placement. Such platforms could help collect sufficient data for hazard management strategies up to the basin scale in Stelvio National Park and other sectors of the Italian Alps, eventually replacing aerial LiDAR surveys. Cost

analyses (Matese et al., 2015) should also be performed to evaluate the benefits of improved spatial resolution and DEM accuracy of UAVs compared to aerial and satellite surveys and choose the best approach for individual cases."

RC Line 493 Page 22: please see comment L410

AC: We have modified the manuscript as follows: "The analysis of the completeness of surface reconstruction also revealed some issues related to the adopted techniques (see Fig. 6). Specifically, TLS suffered from severe occlusions which prevented acquisition of data in the central part of the sample area, while UAV photogrammetry was able to reconstruct the upper portion of the sample area but not the vertical cliff. Only terrestrial photogrammetry acquired a large number of points in all areas."

RC Line 496 Page 22: please replace here and elsewhere "exposed upward" with horizontal, or sub-horizontal, or moderately sloping (maybe adding slope thresholds for improved understanding).

AC: The term has been modified accordingly throughout the manuscript.

RC Line 516 Page 23: the sections 3.1 and 3.1.1 are very long and can be highly summarized, presenting just the results and moving further considerations in the discussion section.

AC: We have shortened these sections. Relevant considerations have been moved to the Discussion Section following your comments.

RC Lines 518-523 Page 23: I suggest removing or strongly summarizing this part

AC: These lines were deleted to shorten the manuscript.

RC Line 523 Page 23: do the authors have ablation measurements (or estimates) during the survey period? What is the impact of glacier ablation in calculations?

AC: We have added a paragraph concerning this issue in the Discussion section, as described in your comment to lines 171-173.

RC Line 540 Page 24: retained or based on some metrics/methodological constraints?

AC: The accuracy of TLS is less influenced by controlling factors (network geometry, object texture, lighting conditions) than the accuracy of photogrammetry. For this reason we have decided to adopt TLS point clouds as benchmarks. We have thus added:"Since no available benchmarking data set (e.g. accurate static GNSS data) was concurrently collected during the 2016 campaign, the TLS point cloud was used as a reference, as it less influenced by controlling factors (network geometry, object texture, lighting conditions). "

RC Line 576 Page 25: ∆DEM could be replaced by the more commonly-used dem of difference (DOD)

AC: We have replaced the term accordingly throughout the manuscript

RC Lines 575-579 Page 25: this part is poorly written and hardly readable/understandable. Please reformulate

AC: We have deleted this part as suggested by Reviewer 2.

RC Lines 579-581 Page 25: this part is obvious and redundant

AC: We have removed this sentence accordingly.

RC Line 593 Page 26: please complete numbers with minus sign and measurement units

AC: We have modified the manuscript accordingly. We use minus signs when we use the term "changes" but no sign when we use the term "thinning" and related since thinning already implies a loss. We have added measurement units wherever needed.

RC Line 594 Page 26: the eastern part of the ablation tongue

AC: We have rephrased as "the eastern section of the glacier tongue"

RC Lines 603-610 Pages 26-27: I am not fully convinced that the paper deserves

section 3.3. My suggestion is to remove it and move concepts above, when the authors write about the complementarity of the two survey techniques.

AC: Merging of the two datasets required a fine coregistration which was important to mention. We have therefore moved Subsection 3.3 to the methods section and provided more information on the merging procedure, as suggested by Reviewer 2.

RC Lines 613-622 Page 27: the authors try to validate their geodetic mass balance estimates in the lower glacier tongue, using specific mass balance estimations at the surface, for one point (whose location is not reported). Their approach is not correct, because they are comparing single-point vs. mean areal estimates, which can be highly different in the study area given the high lateral gradients in mass balance and elevation changes (Fig. 11), likely attributable to debris cover and differential ablation. Moreover, local geodetic and glaciological mass balance estimates seldom match on glaciers, because the surface elevation change is the result of a complex combination of surface, internal and basal mass exchanges, and of ice dynamics. In particular, vertical displacements (emergence velocity) have to be quantified for local comparisons of the two methods (see for example Fischer, 2011; Sold et al., 2013).

Fischer, A., 2011. Comparison of direct and geodetic mass balances on a multi-annual time scale. The Cryosphere, 5(1), p.107. Sold, L., Huss, M., Hoelzle, M., Andereggen, H., Joerg, P.C. and Zemp, M., 2013. Methodological approaches to infer end-of-winter snow distribution on alpine glaciers. Journal of Glaciology, 59(218), pp.1047-1059.

AC: We have deleted this paragraph accordingly.

RC Lines 612-645 Pages 27-28: in my opinion this is not discussion, but mostly a presentation of results. Here the authors should discuss the accuracy of their results, the problematics in data collection and processing, the generalizability and the added values of the employed techniques. In particular, they should provide a discussion of the pros and cons of the proposed approaches, a comparison of their results with the existing literature, and critical evaluation of local-scale high-resolution surveys vs.

glacier-wide surveys, which are required for geodetic mass balance estimates and comprehensive glacier hazard mapping. Which of the used methods has the highest potential for monitoring rapid glacier evolution and deriving hazards? Is there a method that has the potential to become a standard in glacier monitoring strategies, according to the authors? With which improvements/adjustments?

AC: We have moved the description of glacier hazards to the results section and deleted the paragraph on geomorphological evolution of the glacier tongue as not relevant to this study. In addition, the new Discussion section has been rewritten by rearranging content from the results section and providing more information to discuss the issues mentioned in your comment.

RC Line 687 Page 30: I wonder if there is a more quantitative approach to be used here (such as DOD) to better exploit the new technologies. All the surface features described in this section are so large to be clearly visible by quick field observations and the tourist path can be easily changed accordingly. In my opinion the advantage of AUV and/or terrestrial photogrammetry lies in the possibility of automatically mapping and measuring these features from the DOD. Therefore, I suggest to add this quantitative assessments, starting from elevation changes as displayed in Fig. 11, where the collapse structures are evident.

AC: We agree that tourist paths can be changed but to do so, one requires a comprehensive mapping of hazard features and an insight into their evolution which can not be obtained by simple field observation. While the areas that underwent substantial collapse can be easily mapped with an automatic approach from the DoD, in other areas manual interpretation is required to map newly opened fractures whose vertical displacement is too low to be effectively recognized and to distinguish them from crevasses. Recent fractures are particularly important to map to predict the future evolution of the glacier. Therefore, we preferred manually mapping the hazard features. We now clarify the methodological basis for this mapping in the methods, section 4.2 and added further information concerning the vertical displacement of features in the

results section. Section 4.2 now reads: “The investigation of glacier hazards was conducted by considering datasets from 2014 and 2016. In 2014, only the point cloud and UAV orthophoto were available, while in 2016 the point cloud obtained by merging UAV and close-range photogrammetric data sets was used in combination with the UAV orthophoto. In this study, we focused on ring faults and normal faults, which were manually delineated by using geometric properties from the point clouds while color information from orthophotos was used as a cross-check. On point clouds, mapping is based on visual inspection of vertical displacements following faulting or subsidence. On orthophotos, both types of structures also generally appear as linear features in contrast with their surroundings. As these structures may look similar to crevasses, further information concerning their orientation and location needs to be assessed for discrimination. The orientation of fault structures is not compatible with glacier flow, with ring faults also appearing in circular patterns. Their location is limited to the glacier margins, medial moraines and terminus (Azzoni et al., submitted). After delineation, we also analysed the height of vertical facies using information from the point clouds. ”

RC Line 695 Page 30: increased rate of surface lowering (not necessarily equal to surface ablation).

AC: We now use the term “thinning” or “thinning rate” throughout the manuscript.

RC References Page 32: the reference list is rather long and, notably, one third of the references are self-citations. Please check if all these references are pertinent and functional to the paper

AC: The list of references has been shortened from 72 to 61 references, of which 10 are self-citations.

RC Table 2 Page 40: please provide explanation for GSD

AC: The table was removed to shorten the manuscript.

RC Table 3 Page 41: I guess that the last column shows elevation differences “with”

co-registration shifts

AC: We have modified the column header accordingly.

RC Table 8 Page 46: I suggest showing in a figure the extent and location of the common reference area

AC: The extent and location of the reference area is now provided in Figure 1

RC Figure 5 Page 51: I think that a) and b) are inverted

AC: We have modified the panel order accordingly.

RC Figure 7, 12 and 14 Page 53: these figures can be merged in a single image AC: We have deleted Figure 12 and Figure 14. The location of trails is now shown in Fig. 1 and Fig. 7 where the hazards are shown as well.

---

## Author Comment (AC2) · 24 Sep 2017

We have prepared a point by point response to the reviewer's comments. In the following text, reviewer's comments are reported as RC, our answers as AC.

RC: Interactive comment on "Combination of UAV and terrestrial photogrammetry to assess rapid glacier evolution and conditions of glacier hazards" by Davide Fugazza et al. S. Gindraux (Referee) saskia.gindraux@wsl.ch Summary: In this manuscript, the authors describe and analyse geomorphological features on the tongue of a hazard-prone glacier in the Italian alps with the help of different (closerange) remote sensing methods. They found that the merging of point

clouds generated from two methods (UAV- and terrestrial photogrammetry) present the best product in order to map glacier hazards. The idea for this "data fusion" is new and potentially interesting, however it is not sufficiently described. The manuscript has nice Figures, well-displayed tables and is written in an easy-to-follow language style, that I appreci-ated to read. However, sections are missing and there is a need for a re-shuffling work (i.e. put the information in the right sections). The authors also invested a lot of effort in the text by inserting a great deal of information but the manuscript is overall too long and needs shortening. This work on analysing glacier hazards for the population is surely valid but a stronger emphasis on its scientific relevance is needed. Due to these issues, I think this manuscript needs major revision.

AC: Dear Reviewer, thank you for your detailed comments. We have greatly shortened the manuscript, by selecting only the most important information for the reader. We have rewritten the introduction section to focus on glacier hazards and reorganized the results and discussion section. The data section has been shortened and a new methods section is now provided which explains the criteria used in the analysis of point clouds and the methodological basis for glacier hazard mapping. Finally, we have rewritten the conclusion section by summarizing the main findings of our work. The glacier hazards analyzed in this study are caused by the glacier collapse which is linked to climate change. Thus, they provide a dramatic evidence of this phenomenon in high mountain regions and we have highlighted this information in the manuscript..

RC: General comments:

The next paragraphs of this review contain the general issues in each manuscript sections. Introduction (Sec. 1): - Better define the aim and workflow of your work: The introduction section is constituted of three parts that are not well linked together. One of the main issue is that there is no clear "story" . I understand what the authorsdid in term of analysis but how they linked their results to the "evolution and conditions of glacier hazard" question mentioned in the title) was unclear to me. In order to better understand how the authors plan to use the remote sensing products in order to map

(or analyse? This is also not clear) the differeùnt glacier hazards, including a dedicated method section would be very valuable.

- Link the paragraphs to prepare the reader and move information to other sections: o In the first part, the two first paragraphs (Lines 27 to 63) explains changes of glacier and permafrost environment to climate change and gives examples of changes and hazards. The GLOFs are also mentioned and not mentioned again until the conclusion, which is unsettling for the reader. Maybe listing the glacier natural hazards that will be analysed in this study would be useful for the reader. o In the section 1.1, the first paragraph (Lines 65 to 85) is on remote sensing and natural hazards monitoring (what the title promises), while the second (Lines 86 to113) is on the general use of UAV on glaciers. These two subjects do not link together and the reader is not prepared to read the second paragraph. Maybe it would be good to include it in a new method section? This depends on the "story" you want to tell. o The third paragraph of subsection 1.1 (Lines 114 to 126) does not include what the title suggests. Instead of a detailed text about remote sensing and glacier hazards, it identifies the research gap and the aim of the study. I suggest to merge everything (all introduction subsections) in one longer introduction and write a text that prepares the reader for the coming section (e.g. data,results,: : :), as well as states a clear research question and description of the methods used to answer it.

AC: We have rewritten and shortened the introduction which now focuses more on glacier hazards, especially those exacerbated by climate change as suggested by Reviewer 1. We have kept a description of hazards that we did not study in this article to widen its scope, as we believe that our approach could be useful to study all types of glacier hazards. We have deleted subsections and added links between paragraphs. Now, the first paragraph deals with glacier hazards and the second with remote sensing of glacier hazards, especially proximal remote sensing, i.e. terrestrial photogrammetry, UAVs and TLS. In the third paragraph, we state the research gap and our aims, and we specifically describe here the hazards we investigated in this study. We then

briefly describe the data and methods we used to address our research question to prepare the reader for the data and methods section, which has also been widened by moving content from the former results section and adding information on mapping of glacier hazards and point cloud merging. The introduction section now reads: "Glacier and permafrost-related hazards can be a serious threat to humans and infrastructure in high mountain regions (Carey et al., 2014). The most catastrophic cryospheric hazards are generally related to the outburst of water, either through breaching of moraine- or ice-dammed lakes or from the englacial or subglacial system, causing floods and debris flows. Ice avalanches from hanging glaciers can also have serious consequences for downstream populations (Vincent et al., 2015), as well as debris flows caused by the mobilization of accumulated loose sediment on steep slopes (Kaab et al., 2005a). Less severe hazards, but still particularly threatening for mountaineers are the detachment of seracs (Riccardi et al., 2010) or the collapse of ice cavities (Gagliardini et al., 2011; Azzoni et al., submitted). While these processes are in part typical of glacial and periglacial environments, there is evidence that climate change is increasing the likelihood of specific hazards (Kaab et al., 2005a). In the European Alps, accelerated formation and growth of proglacial moraine-dammed lakes has been reported in Switzerland, amongst concern of possible overtopping of moraine dams provoked by ice avalanches (Gobiet et al., 2014). Ice avalanches themselves can be more frequent as basal sliding is enhanced by the abundance of meltwater in warmer summers (Clague, 2013). Glacier and permafrost retreat, which have been reported in all sectors of the Alps (Smiraglia et al., 2015; Fischer et al., 2014; Gardent et al, 2014; Harris et al., 2009), are a major cause of slope instabilities which can result in debris flows, by debuttressing rock and debris flanks and promoting the exposure of unconsolidated and ice-cored sediments (Keiler et al., 2010; Chiarle et al., 2007). Glacier downwasting is also increasing the occurrence of structural collapses and while not directly threatening human lives, sustained negative glacier mass balance can also cause shortages of water for industrial, agricultural and domestic use and energy production, negatively affecting even populations living away from glaciers. Finally, glacier retreat and the increase in

glacier hazards negatively impacts on the tourism sector and the economic prosperity of high mountain regions (Palomo, 2017). The increasing threat from cryospheric hazards under climate change calls for the adoption of mitigation strategies. Remote Sensing has long been recognized as an important tool to produce supporting data to this purpose, owing to the ability to generate digital elevation models (DEMs) and multispectral images. DEMs are particularly useful to detect glacier thickness and volume variations (Fischer et al., 2015; Berthier et al., 2016) and to identify steep areas that are most prone to geomorphodynamic changes such as mass movements (Blasone et al., 2014). Multispectral images at a sufficient spatial resolution enable the recognition of most cryospheric hazards (Quincey et al, 2005; Kaab et al., 2005b). While satellite images from Landsat and ASTER sensors (15-30 m ground sample distance - GSD) are practical for regional-scale mapping (Rounce et al, 2017), the assessment of hazards at the scale of individual glaciers or basins requires higher spatial resolution, which in the past could only be achieved via dedicated field campaigns with terrestrial laser scanners (TLS) (Bodin et al., 2008; Riccardi et al., 2010). Recent years have seen a resurgence of terrestrial photogrammetric surveys for the generation of DEMs (Piermattei et al., 2015; Kaufmann and Seier, 2016) due to important technological advancements including the development of Structure-from-Motion (SfM) Photogrammetry and its implementation in fully automatic processing software, as well as the improvements in the quality of camera sensors (Westoby et al., 2012). In parallel, unmanned aerial vehicles (UAVs – Colomina & Molina, 2014, O'Connor et al., 2017) have started to emerge as a viable alternative to TLS for multi-temporal monitoring of small areas. UAVs promise to bridge the gap between field observations, notoriously difficult on glaciers, and coarser resolution satellite data (Bhardwaj et al., 2016a). Although the number of studies employing them in high mountain environments is slowly increasing (see e.g. Fugazza et al., 2015; Gindraux et al., 2016; Seier et al, 2017), their full potential for monitoring of glaciers and particularly glacier hazards has still to be explored. In particular, the advantages of UAV and terrestrial SfM-Photogrammetry, and the possibility of data fusion to support hazard management strategies in glacial
environments needs to be investigated and assessed. In this study, we investigated a rapidly downwasting glacier in a protected area and highly touristic sector of the Italian Alps, Stelvio National Park. We focused on the glacier terminus and the hazards identified there, i.e., the formation of normal faults and ring faults. The former occur mainly on the medial moraines and glacier terminus and are due to gravitational collapse of debris-laden slopes. The latter develop as a series of circular or semicircular fractures with stepwise subsidence, caused by englacial or subglacial meltwater creating voids at the ice-bedrock interface and eventually the collapse of cavity roofs. While often overlooked, these collapse structures are particularly hazardous for mountaineers and likely to increase under a climate change scenario (Azzoni et al., submitted). They are more dangerous than crevasses because of the larger size and because they could be filled with snow and rendered entirely or partly invisible to mountaineers. We conducted our first UAV survey of the glacier in 2014; then, through a dedicated field campaign carried out in summer 2016, we compared different platforms and techniques for point cloud, DEM and orthomosaic generation to assess their ability to monitor glacier hazards: UAV photogrammetry, terrestrial photogrammetry and TLS. The aims were: (1) comparing UAV- and terrestrial photogrammetric products acquired in 2016 against the TLS point cloud; (2) identifying glacier-related hazards and their evolution between 2014-2016 using the merged point cloud from UAV and terrestrial photogrammetry and UAV orthophotos; and 3) investigating ice thickness changes between 2014-2016 and 2007-2016 by comparing the two UAV DEMs and a third DEM obtained from stereoprocessing of aerial photos captured in 2007."

RC: o In my opinion, reading about the study area (Subsection 1.2; Lines 127 to 155), in the introduction is very uncommon. I would merge it in another section (e.g. in the data section or in the new? Method section). This section however is too long and it should be shortened, containing only the information the reader needs to understand your work.

AC: The study area has been moved to a dedicated section. Furthermore, we have

shortened this section by deleting the sentences concerning recent glacier changes, the AWS and other research performed on the glacier as not strictly relevant to this study. We have deleted bullet points and rephrased the paragraph as follows: "The Forni Glacier (see Fig. 1) has an area of 11.34 km2 based on the 2007 data from the Italian Glacier Inventory (Smiraglia et al., 2015), an altitudinal range between 2501 and 3673 m a.s.l. and a North-North-Westerly aspect. The glacier retreated markedly since the little ice age (LIA), when its area was 17.80 km2 (Diolaiuti & Smiraglia, 2010), with an acceleration of the shrinking rate in the last three decades, typical of valley glaciers in the Alps (Diolaiuti et al, 2012, D'Agata et al; 2014). It has also undergone profound changes in dynamics in recent years, including the loss of ice flow from the eastern accumulation basin towards its tongue and the evidence of collapsing areas on the eastern tongue (Azzoni et al., submitted). One such area, hosting a large ring fault (see Fig. 2d) prompted an investigation carried out with Ground Penetrating Radar (GPR) in October 2015, but little evidence of a meltwater pocket was found under the ice surface (Fioletti et al., 2016). Since then, a new ring fault appeared on the central tongue, and the terminus underwent substantial collapse (see Fig. 2a,b,c,e). Continuous monitoring of these hazards is important as the site is highly touristic (Garavaglia et al., 2012), owing to its location in Stelvio Park, one of Italy's major protected areas, and its inclusion in the list of geosites of Lombardy region (see Diolaiuti and Smiraglia, 2010). The glacier is in fact frequently visited during both summer and winter months. During the summer, hikers heading to Mount San Matteo take the trail along the central tongue, accessing the glacier through the left flank of the collapsing glacier terminus. During wintertime, ski-mountaineers instead access the glacier from the eastern side, crossing the medial moraine and potentially collapsed areas there (see Fig. 1)."

RC: Data Sources: acquisition and processing (Sec. 2): - Shorten the whole section: A lot of information in this section is not crucial for the reader that gets lost. I suggest rewriting it in a more succinct way and remove text. See more details in the short comments. - Re-order the subsections: It is hard to follow this Sec. 2 because, the reader is starting to read a section about a new UAV survey, then terrestrial survey,

TLS, control points (that belong to UAV and TLS), and finally a UAV survey again. I suggest that the different subsections should be divided per surveying method rather than the different datasets. For instance: 2.1 UAV photogrammetry 2.1.1 Dataset 2014 Content example: Type of UAV, flights, GCP network, software to generate products (and eventually workflow), resolution of end product. 2.1.2 Dataset 2016 2.2 Terrestrial photogrammetry 2.3 TLS 2.4 Aerial photogrammetric survey

AC: We have reordered the data section according to your suggestion and shortened it following your minor comments.

RC:

Results (Sec. 3): - Shorten and merge sections: o The first part of the result section (subsection 3.1 and 3.11) is about statistics describing the point clouds, and is too long. The subsections could be merged and shortened, the number of tables and figures reduced. A large part of the text in these two sections also belong to the discussion section (see short comments).

AC: This subsection has been shortened. Part of it was moved to the methods section and part to the Discussion Section following your short comments. The results sections concerning point clouds comparison was merged into one section and tables and figures reduced according to your short comments. Now results section 5.1 reads: "The analysis of point density shows significant differences between the three techniques for point cloud generation (see Table 2). Values range from 103 to 2297 points/m2 depending on the surveying method, but the density was generally sufficient for the reconstruction of the different surfaces shown in Fig. 5, except for location 5. Terrestrial photogrammetry featured the highest point density, while UAV photogrammetry had the lowest. In relation to UAV photogrammetry, similar point densities were found in all sample locations, especially for the standard deviations that were always in the range 22-29 points/m2. Mean values were between 103-109 points/m2 in locations 2-4, while they were higher in location 5 (141 points/m2). Due to the nadir acquisition points, the

3D modelling of vertical/sub-vertical cliffs in location 1 was not possible. In relation to TLS, a mean value of point density ranging from 141-391 points/m2 was found, with the only exception of location 5, where no sufficient data were recorded due to the position of this region with respect to the instrumental standpoint. Standard deviations ranged between 69-217 points/m2, moderately correlated with respective mean values. The analysis of the completeness of surface reconstruction also revealed some issues related to the adopted techniques (see Fig. 6). Specifically, TLS suffered from severe occlusions which prevented acquisition of data in the central part of the sample area, while UAV photogrammetry was able to reconstruct the upper portion of the sample area but not the vertical cliff. Only terrestrial photogrammetry acquired a large number of points in all areas. In terms of point cloud distance (see Table 3), the comparison between TLS and terrestrial photogrammetry resulted in a high similarity between point clouds, with no large differences between different sample areas. Conversely, the comparison between TLS and UAV photogrammetry and terrestrial and UAV photogrammetry provided significantly worse results, which may be summarized by the RMSEs in the range 21.1-37.7 cm and 20.7-30.4 cm, respectively. The worse values were both obtained in the analysis of location 2, which mostly represents a vertical surface, while the best agreement was found within location 3 which is less inclined. As the UAV flight was geo-referenced on a set of GCPs with an RMSE of 40.5 cm, the ICP co-registration may have not totally compensated the existing bias. "

RC: o Some methodological description seems to be "hidden" in the result section. I suggest that the text is re-shuffled and shortened. More details can be found in the short comments.

AC: We have moved relevant parts of the result section to the methods section following your short comments. Methods section 4.1 now is dedicated to the comparison of point clouds and reads: "The comparison between point clouds generated during the 2016 campaign had the aim of assessing their geometric quality before their application for the analysis of hazards. These evaluations were also expected to provide

some guidelines for the organization of future investigations in the field at the Forni Glacier and in other Alpine sites. Specifically, we analysed point density (points/m2) and completeness, i.e. % of area in the ray view angle. Point density partly depends upon the adopted surveying technique, since it is controlled by the distance between sensor and surface and the obtainable spatial resolution. In SfM-Photogrammetry, the latter property is affected by dense matching, while in TLS it can be set up as data acquisition input parameter. In this study, the number of neighbours N (inside a sphere of radius R=1 meter) divided by the neighbourhood surface was used to evaluate the local point density D in CloudCompare (www.cloudcompare.org). To understand the effect of point density dispersion (Teunissen, 2009), the inferior 12.5 percentile of the standard deviation ðÍÍJŐ of point density was also calculated. The use of these local metrics allowed to distinguish between point density in different areas, since this may largely change from one portion of surface to another. A further metric in this sense was point cloud completeness, referring to the presence of enough points to completely describe a portion of surface. In this study, the visual inspection of selected sample locations was used to identify occlusions and areas with lower point density. To analyse these properties, five regions were selected (see Fig. 5), located on the glacier topographic surface and characterized by different glacier features and the presence of hazards: 1) Glacial cavity composed by subvertical and fractured surfaces over 20 m high, and forming a typical semicircular shape; 2) glacial cavity over 10 m high with the same typical semi-circular shape as location 1, covered by fine- and medium-size rock debris; 3) normal fault over 10 m high; 4) highly-collapsed area covered by fine- and medium-size rock debris and rock boulders; and 5) planar surface with a normal fault covered by fine- and medium-size rock debris and rock boulders. The analysis of local regions was preferred to the analysis of the entire point clouds for the following reasons: 1) the incomplete overlap between point clouds obtained from different methods; 2) the opportunity to investigate the performances of the techniques in diverse geomorphological situations. Finally, we compared the point clouds in a pairwise manner within the same sample locations. Since no available benchmarking data set (e.g. accurate static

GNSS data) was concurrently collected during the 2016 campaign, the TLS point cloud was used as a reference, as it less influenced by controlling factors (network geometry, object texture, lighting conditions). When comparing both photogrammetric data sets, the one obtained from UAV was used as reference because of the even distribution of point density within the sample locations. The presence of residual, non-homogenous geo-referencing errors in the data sets required a specific fine registration of each individual sample location, which was conducted in CloudCompare using the ICP algorithm (Pomerleau et al., 2016). Then, point clouds in corresponding sample areas were compared using the M3C2 algorithm implemented in CloudCompare (Lague et al., 2013). This solution allowed us to get rid of registration errors from the analysis, which could then be focused on the capability of the adopted techniques to reconstruct the local geometric surface of the glacier in an accurate way."

RC: o Part of the text in subsubsection 3.1.2 (Lines 517 to 570) belongs to the discussion section (see short comments for more details).

AC: We have moved relevant considerations in subsect. 3.1.2 to the discussion section and methodological descriptions to the methods section. More information is provided in the answer to your major comments about the discussion section and short comments.

RC: - Clarify "accuracy" and "comparison": Subsubsection 3.1.2 (Lines 517 to 570), concerns the assessment of the point clouds' accuracy. In principle, the absolute accuracy of such point clouds can only be assessed with perfect validation data (e.g. long-term precise GPS data). Each method has its advantages and drawback and thus, generates products with different kind of errors (i.e. they are all differently imperfect/inexact). Therefore, the accuracy of a point cloud cannot be calculated using a point cloud generated from another method; but a comparison can be made. The analysis performed with the help of cloud compare, looks at the differences between the 3D geometry of point cloud pairs only. I would make a clear distinction and use of these terms in the text.

AC: We now avoid using the word "accuracy" during the analysis of point clouds and clarify that no available accurate reference data set was available. We chose TLS as the reference point cloud because it is less influenced by controlling factors (network geometry, object texture, lighting conditions). According to the International vocabulary of metrology JCGM 200:2012, accuracy is a qualitative term that indicates whether the uncertainty is lower than a threshold value identified as suitable for the purposes of a study. We therefore state that "The final accuracy of our UAV photogrammetric products was nevertheless adequate to investigate ice thickness changes over 2 years" in the discussion section following this definition. JCGM 200:2012, see http://www.bipm.org/utils/common/documents/jcgm/JCGM_200_2012.pdf

RC: - Add information: o It is not clear how the glacier thickness information (Section 3.2 and in general) isused in the assessment of glacier hazards. Could you please provide more information in the text?

AC: The information on glacier thickness change provides evidence concerning the processes of glacier downwasting that are linked to glacier hazards. It shows the extent and volume of collapsed areas and the acceleration of thinning rates that is linked to the increase in collapsed areas via higher availability of englacial and subglacial meltwater, which create voids at the ice-bedrock interface and eventually the collapse of cavity roofs. Glacier thinning is also a major cause of slope instabilities which can result in debris flows, by debuttressing rock and debris flanks and promoting the exposure of unconsolidated and ice-cored sediments (see e.g. Keiler et al., 2010; Chiarle et al., 2007). Thus, the information on glacier thinning is useful to provide evidence of increased susceptibility of high mountain areas to hazards related to climate change. Finally, glacier thinning can be considered a hazard by itself as it affects the availability of water resources for industrial and domestic use and the prosperity of high mountain regions, in view of the touristic value of glaciers. We now specify the reasons why we conducted the analysis in the introduction section, as: "Glacier and permafrost retreat, which have been reported in all sectors of the Alps (Smiraglia et al., 2015; Fischer et al.,

2014; Gardent et al, 2014; Harris et al., 2009), are a major cause of slope instabilities which can result in debris flows, by debuttressing rock and debris flanks and promoting the exposure of unconsolidated and ice-cored sediments (Keiler et al., 2010; Chiarle et al., 2007). Glacier downwasting is also increasing the occurrence of structural collapses and while not directly threatening human lives, sustained negative glacier mass balance can also cause shortages of water for industrial, agricultural and domestic use and energy production, negatively affecting even populations living away from glaciers. Finally, glacier retreat and the increase in glacier hazards negatively impacts on the tourism sector and the economic prosperity of high mountain regions (Palomo, 2017)." and "in this study, we investigated a rapidly downwasting glacier in a protected area and highly touristic sector of the Italian Alps, Stelvio National Park. We focused on the glacier terminus and the hazards identified there, i.e., the formation of normal faults and ring faults. The former occur mainly on the medial moraines and glacier terminus and are due to gravitational collapse of debris-laden slopes. The latter develop as a series of circular or semicircular fractures with stepwise subsidence, caused by englacial or subglacial meltwater creating voids at the ice-bedrock interface and eventually the collapse of cavity roofs. While often overlooked, these collapse structures are particularly hazardous for mountaineers and likely to increase under a climate change scenario (Azzoni et al., submitted). They are more dangerous than crevasses because of the larger size and because they could be filled with snow and rendered entirely or partly invisible to mountaineers. ". In the conclusion, we have added: "The analysis of glacier thickness changes suggests a feedback mechanism which should be further analysed, with higher thinning rates leading to increased occurrence of collapses, with additional release of meltwater. Glacier downwasting is also of relevance for risk management in the protected area, providing valuable data to assess the increased chance of rockfalls following glacier retreat and to improve forecasts of glacier meltwater production."

RC: o Subsection 3.3 (Lines 571 to 602) also requires more information on how this dataset merging has been made. A method section would be useful, especially when you cite this merging be the best product to monitor glacier hazards in the conclusion.

This could be a very interesting point! And maybe the main novelty of this study and should better be highlighted.

AC: We have added information on the dataset merging and moved Subsection 3.3. to the methods section. The section now reads: "To improve coverage of different glacier surfaces, including planar areas and normal faults, photogrammetric point clouds from the 2016 campaign were merged. Prior to point cloud merging, a preliminary co-registration was performed on the basis of the ICP algorithm in CloudCompare. Regions common to both point clouds were used to minimize the distances between them and find the best co-registration. The point cloud from UAV photogrammetry, which featured the largest extension, was used as reference during co-registration, while the other was rigidly transformed to fit with it. After this task, both original point clouds resulted aligned into the same reference system. In order to get rid of redundant points and to obtain a homogenous point density, the merged point cloud (see Fig. 5) was subsampled keeping a minimum distance between adjacent points of 20 cm. The final size of this data set is approximately 4.4 million points, which represents a manageable data amount on up-to-date computers. The colour RGB information associated to each point in the final point cloud was derived by averaging the RGB information of original points in the subsampling volumes. While this operation resulted in losing part of the original RGB information, it helped provide a realistic visualization of the topographic model, which can aid the interpretation of glacier hazards."

RC: o Subsection 3.3 (Lines 571 to 602) present the fusion of two point cloud datasets. It is very confusing for the reader to switch between point cloud (Section 3.1 and 3.3) and DEM sections (Section 3.2). Can you maybe change the section's order?

AC: We have changed the section order in the methods and results sections.Subsection 3.3. was also moved to the methods section. In methods, section 4.1 now deals with point cloud analysis, 4.2. with glacier hazards and 4.3 DEM coregistration in the methods section. In results, section 5.1 deals with point cloud analysis, 5.2 with glacier hazards and 5.3 with glacier thickness change.

RC: o It was very unclear to me after reading the results section, why the authors performed all these different analyses (i.e point cloud statistical analysis, point cloud accuracy, point cloud fusion and glacier thickness change), when at the end (Discussion section) you present a map of the glacier hazards (location of collapse, Fig.15) generated with the help of UAV orthophotos?. Could you please better explain their link in the introduction and method section?

AC: We have conducted the analysis again by using primarily the information from point clouds to map glacier hazards, while UAV orthophotos were used as a cross-check. On point clouds, normal faults and ring faults are visible due to the vertical displacement caused by faulting or subsidence. On orthophotos, they can instead be identified owing to the contrast with their surroundings. Glaciological information (orientation and location of features) is also necessary to distinguish these features from crevasses. The new procedure actually allowed us to recognize more features. We now describe in the methods section the procedures used in mapping glacier hazards, as: "The investigation of glacier hazards was conducted by considering datasets from 2014 and 2016. In 2014, only the point cloud and UAV orthophoto were available, while in 2016 the point cloud obtained by merging UAV and close-range photogrammetric data sets was used in combination with the UAV orthophoto. In this study, we focused on ring faults and normal faults, which were manually delineated by using geometric properties from the point clouds while color information from orthophotos was used as a cross-check. On point clouds, mapping is based on visual inspection of vertical displacements following faulting or subsidence. On orthophotos, both types of structures also generally appear as linear features in contrast with their surroundings. As these structures may look similar to crevasses, further information concerning their orientation and location needs to be assessed for discrimination. The orientation of fault structures is not compatible with glacier flow, with ring faults also appearing in circular patterns. Their location is limited to the glacier margins, medial moraines and terminus (Azzoni et al., submitted). After delineation, we also analysed the height of vertical facies using information from the point clouds. ".

RC: Discussion (Sec. 4): - Link the discussion to the result section: The discussion section (Lines 611 to 687) is divided into two parts: One on the geomorphological evolution of the glacier tongue and the second about glacier-related hazards and how to risk is reduced through hazard mapping. Although the information is interesting, almost none of the discussion is based on the result section, and this is what the reader expects. Can you please change the text accordingly?

AC: We have removed the section about the evolution of the glacier tongue as not relevant to this study. We have also moved the glacier hazards mapping section to the results section. The discussion section now reports the advantages of different techniques for hazard mapping and risk assessment as suggested by you and the other Reviewer.

RC: - Discuss your results by comparing them to results of other studies: Comparing the different point clouds with a) statistical numbers, b) point density and c) completeness, and judging the best mapping method based on them, follow a correct method workflow and give good results but the later are not new. There are many papers that state the drawbacks of the surveying methods in a mountain terrain e.g. that the TLS data have a lot of "holes" and that the UAV data do not represent the vertical geometry well. I would consider making reference to them and compare your results.

AC: We have moved relevant considerations from the results to the discussion section and added references to other studies conducted in glacial environments, where available, to investigate the advantages of the different techniques. The discussion section now reads: "The choice of a technique to monitor glacier hazards and the glacier geodetic mass balance can depend on several factors, including the size of the area, the desired spatial resolution and accuracy, logistics and cost. In this study, we focused on spatial metrics, i.e. point density, completeness and distance between point clouds to evaluate the performance of UAV, close-range photogrammetry and TLS in a variety of conditions. Considering point density, terrestrial photogrammetry resulted in a denser data set than the other techniques. This is mostly motivated by the possibil-

[revised manuscript text omitted]

RC: Conclusion (Sec. 5): - Shorten and clarify the main message: The conclusion (Lines 688 to 730) are a mix of different sections, that are, presently, not well linked

together. In particular, a clear conclusive message is missing. My advice would be to revise this part and to include, amongst other, a short summary for how and why this study has been done, which would help to present a better "overall story".

AC: We have rewritten this section by including a short introductory paragraph summarizing the reason of this study and methods. We have added a bullet point to highlight the main finding of our work and add a final conclusive message at the end, as: "In our study, we compared point clouds generated from UAV photogrammetry, close-range photogrammetry and TLS to assess their quality and evaluate the potential in mapping and describing glacier hazards such as ring faults and normal faults, by carrying out a specific campaign in summer 2016. In addition, we employed orthophotos and point clouds from a UAV survey conducted in 2014 to analyze the evolution of glacier hazards and a DEM from an aerial photogrammetric survey conducted in 2007 to investigate glacier thickness changes between 2014 and 2016. The main findings of our study include: UAVs and terrestrial photogrammetric surveys provide reliable performances in glacial environments, outperform TLS in terms of logistics and costs, and are more flexible in relation to meteorological conditions. UAV and terrestrial photogrammetric blocks can be easily integrated providing more information than individual techniques to help identify glacier hazards. UAV-based DEMs can be employed to estimate thickness changes but improvements are necessary in terms of area covered and accuracy to calculate the glacier geodetic mass balance of large glaciers. The Forni Glacier is rapidly collapsing with an increase in ring faults size, providing evidence of climate change in the region. The glacier thinning rate increased owing to collapses to 5.20 ma-1 between 2014 and 2016. The maps produced from the combined analysis of UAV and terrestrial photogrammetric point clouds can be made available through GIS web portals of Stelvio National Park or Lombardy region (http://www.geoportale.regione.lombardia.it/). A permanent monitoring programme should be setup to help manage risk in the area, issuing warnings and assisting mountain guides in changing hiking and ski routes as needed. The analysis of glacier thickness changes suggests a feedback mechanism which should be further analysed, with higher thinning rates leading to increased occurrence of collapses, with additional release of meltwater. Glacier downwasting is also of relevance for risk management in the protected area, providing valuable data to assess the increased chance of rockfalls and to improve forecasts of glacier meltwater production. While our test was conducted on one of the largest glaciers in the Italian Alps, the integrated photogrammetric approach is easily transferrable to similar sized and much smaller glaciers, where it would be able to provide a comprehensive assessment of hazards and mass balance and become useful in decision support systems for natural hazard management. In larger regions, UAVs hold the potential to become the platform of choice but their performances and cost-effectiveness compared to aerial and satellite surveys need to be further evaluated."

RC: Comments on Figures and Tables: I generally enjoyed looking at the figures and tables. The colors, the size and the contast of the Figures are well chosen and their appearance encouraged me to read the text. Hereafter are a few suggestions of changes.

RC Figure 1: I suggest to reduce the area of figure 1a and to merge it with Figure 1b (Only one figure for the glacier's location). Can you please specify what are the black outlines and from which year? The location of the TLS standpoint would also be valuable.

AC: We now show only one figure for the glacier map, with a small inset illustrating the glacier location within Italy. The figure includes the location of features reported in Fig.2, UAV take-off/landing sites, TLS standpoint, GCPs, hiking/ski trails which determine the vulnerability to glacier hazards. Finally, we show the reference area for volume change calculation

RC Figure 2: It would be helpful to see where these pictures are located on the glacier. Maybe enlarge the glacier on Figure 1 and set the letters (a-e) at the correct location? Or make a new overview map similar to Figure 7.

AC: We now show the location of these features in Figure 1.

RC Figure 3: b) A more exhaustive caption (with UAV name) and presentation of the other objects would be useful. Other that, Figure 3 does not seems to add much information. Consider merging it with Table 1.

AC: We have merged Table 1 and Figure 3 accordingly. The UAV full names are now provided in the table within the Figure.

RC Figure 4: Many other figures in the manuscript display the glacier tongue. Would it be possible to put the GCP location on one of them instead of creating a new image just for this? Caption: Add UAV in the caption, such as: "of the 2016 UAV survey".

AC: The location of GCPs is now shown in Figure 1.

RC Figure 5: Please increase the resolution of the image so that the GCP numbers are readable. Consider specifying the year of this survey (2016). Moreover, it would be nice to twist the images so that they have the same view angle (e.g. that on both images the GCP12 is front and GCP10 right).

AC: We have replaced the figure by adopting the same view for the upper and lower panel and adding labels over GCPs to improve readability. The caption now reads: "3D reconstruction of the glacier terminus from the terrestrial photogrammetric survey of 2016 : (a) locations of camera stations in front of the glacier and 3D coordinates of tie points extracted during SfM for image orientation; (b) point cloud of the glacier terminus with positions of GCPs."

RC Figure 6: This is a nice but large image that does not give much information. If you want to show the GCP or measuring device, part of the image can be cropped and merged in Figure 3 or another one.

AC: We have removed Figure 6 accordingly.

RC Figure 7: Please start numbering with 1 on the upper left corner. The background image could be brighter. Caption: Please elaborate (e.g. Location of different glacier features or hazard-prone areas on the tongue of Forni glacier were the point cloud

comparison has been performed. The background image is the dense point cloud generated from the 2014 UAV survey).

AC: We have modified the image by numbering sample windows as suggested. We have rephrased the caption and moved here the description of sample windows. The caption now reads: "Figure 5: Location of different glacier features or hazard-prone areas on the tongue of Forni glacier were the point cloud comparison was performed. The background image is the merged point cloud generated from the 2016 UAV and terrestrial photogrammetry survey."

RC Figure 8: Figure 8 display part of the information of Table 5. As it does not show new information, consider removing it.

AC: We have removed figure 8 accordingly.

RC Figure 9 & 10: I think both images show the same information, so maybe remove one of them? Please enlarge the numbers on the scale bars.

AC: we have removed Figure 10 accordingly.

RC Figure 12: This is the same image than the background image of Figure 7 right? Either remove it and refer the reader to Figure 7 instead of 12, or show an image where the reader can see the difference between the 2014, the 2016 and the merged point cloud.

AC: We have removed figure 12 accordingly.

RC Figure 15: Please explain the differences between the red and the blue lines on the glacier. Rewrite the caption so that not only a year is given. A "N" close to the arrow would give a meaning to the arrow itself! The year of the glacier outline should be mentioned.

AC: The difference between normal faults and ring faults is explained in the introduction section, and the methods used to map them are now described in section 4.3. The

glacier outlines were those from 2014 in panel a and 2016 in panel b, respectively. We have split the legend to clarify the year of glacier outlines and added an "N" close to the north arrow. We have rewritten the caption as: "Figure 7: location of collapse structures, i.e. normal faults and ring faults and trails crossing the Forni Glacier (a) 2014, with 2014 UAV ortophoto as basemap. The red box marks the area surveyed in 2016. (b) 2016, with 2016 UAV orthophoto as basemap."

RC Table 1: This is a nice summary table but most of the useful information are already in the text. The added value to the paper is minor. Consider removing this table or merge it with Figure 3.

AC: We have merged this table with figure 3 accordingly.

RC Table 2: The # symbols should be removed or indicate that it means "numbers".

AC: We have replaced the # symbol with "number" accordingly.

RC Table 3: The last column should display the elevation differences "with" co-registration right?. How do you explain that the standard deviation values are still of several meters? This should be discussed in the discussion section.

AC: We have replaced "with" with "without". The coregistration method is not expected to cancel out the standard deviation completely (Berthier et al., 2007). We attribute high residual values to two factors: 1) the uncertainty in UAV photogrammetric reconstruction, i.e. lack of GCPs during the 2014 survey and issues related to GCP accuracy during the second. 2) The morphology of the coregistration area, i.e. the glacier outwash plain, which is still subject to significant changes owing to the inflow of glacier meltwater and sediment reworking. We have added a paragraph in the Discussion section that reads: "The uncertainty in UAV photogrammetric reconstruction also factored in the relatively high standard deviation still present after the coregistration between DEMs in areas outside the glacier (2.22 m between 2014 and 2016). Another important factor here is the morphology of the coregistration area, i.e. the outwash

plain, still subject to changes owing to the inflow of glacier meltwater and sediment reworking. The final accuracy of our UAV photogrammetric products was nevertheless adequate to investigate ice thickness changes over 2 years, while the integration with close-range photogrammetry was required to investigate hazards related to the collapse of the glacier terminus. "

RC Table 4: For the #, same comment as for Table 2. The i, ii, iii are not necessary here, or define them. Giving a volume as size is very uncommon and I suggest using area (m2). Consider merging this table with Table 5.

AC: We have replaced the # symbol with "number" and replaced i,ii and iii with the names of the techniques. We also indicate the area instead of the size. The table is now merged with table 5.

RC Table 5: Please specify that the mean and standard deviation is calculated with a function computing local point density. Same note for i, ii and iii as above. Merge with Table 4.

AC: We have replaced i,ii,iii with the names of the techniques and merged the table with table 4. The caption now reads: "Table 2: Area and number of points in each sample window on the Forni Glacier terminus, mean and standard deviation of local point density and number of points above the lower 12.5% percentile in each window.."

RC Table 6: Caption: As it is, the reader does not understand what the M3C2 is. Please define so that every image can be understood as stand-alone.

AC: We have modified the caption of this table as: "Statistics on distances between point clouds computed on the basis of M3C2 algorithm."

RC Table 7: The information of Table 8 is more useful in the sense that we can compare the the mean thickness change etc. over the same area of interest (of 0.32 km2). I would not include Table 7 in the manuscript.

AC: We have removed table 7 accordingly.

RC Table 8: Remove the last sentence. The reader will usually read the text if he/she wants more information ;-)

AC: We have removed the last sentence accordingly.

RC Short comments: The short comments are listed in a supplement .pdf file.

Please also note the supplement to this comment: https://www.nat-hazards-earth-syst-sci-discuss.net/nhess-2017-198/nhess-2017-198-    RC2-supplement.pdf    Interactive comment on Nat. Hazards Earth Syst. Sci. Discuss., https://doi.org/10.5194/nhess-2017-198, 2017.

RC Short comments:

RC Line 27 Page 1: Replace "on" with "in".

AC: We have replaced "on" with "in" accordingly while changing the sentence to focus on glacier hazards.

RC Line 33 Page 2: I would change this word or maybe say: "glacier and permafrost areas are shrinking". So something alike!

AC: The sentence was changed due to restructuring of the introduction section to: "Glacier and permafrost retreat, which have been reported in all sectors of the Alps (Smiraglia et al., 2015; Fischer et al., 2014; Gardent et al, 2014; Harris et al., 2009), are a major cause of slope instabilities which can result in debris flows, by debuttressing rock and debris flanks and promoting the exposure of unconsolidated and ice-cored sediments (Keiler et al., 2010; Chiarle et al., 2007). "

RC Lines 39-42 Page 2: hazards evolving in a downstream direction sounds not right. what about rephrasing like: " Rising temperatures generate land-surface instabilities and therefore increase the occurrence of geomorphological hazards in glacier and permafrost environments."?

AC: The sentence has removed to shorten the introduction.

RC Line 55 Page 2: Refer to Fig. 1 first.

AC: We have removed the reference to Fig.2.

RC Lines 91-99 Page 4: I think this does not need bullet points (as it is not information that you really want to highlight in your text) and can be listed in the text.

AC: We have removed this information to shorten the introduction section.

RC Line 104 Page 5: supraglacial lakes? if yes supraglacial is not needed.

AC: We have deleted this description to shorten the introduction section.

RC Line 114 Page 5: remove "and accuracy evaluation of point clouds".

AC: We have removed these words due to restructuring of the introduction section.

RC Line 121 Page 5: the reader reads this information later ;-)

AC: We have deleted the words accordingly.

RC Line 122 Page 5: Why a reference (and why this one?) here and after TLS but not after the UAV method? I would not put any references here as you are listing methods.

AC: We have deleted references accordingly

RC Line 123 Page 5: "The aims are:"

AC: we have replaced "with the aim of" with "our aims were:"

RC Line 125 Page 5: "which can represent a risk".

AC: we have removed the description here in view of your comment and the other reviewer's one. The specific hazards investigated in this study are described at the start of the paragraph as "In this study, we investigated a rapidly downwasting glacier in a protected area and highly touristic sector of the Italian Alps, Stelvio National Park. We focused on the glacier terminus and the hazards identified there, i.e., the formation of normal faults and ring faults. The former occur mainly on the medial moraines and

glacier terminus and are due to gravitational collapse of debris-laden slopes. The latter develop as a series of circular or semicircular fractures with stepwise subsidence, caused by englacial or subglacial meltwater creating voids at the ice-bedrock interface and eventually the collapse of cavity roofs. While often overlooked, these collapse structures are particularly hazardous for mountaineers and likely to increase under a climate change scenario (Azzoni et al., submitted). They are more dangerous than crevasses because of the larger size and because they could be filled with snow and rendered entirely or partly invisible to mountaineers. ".

RC Lines 128-131 Page 6: Here there is a lot of information that is not really necessary to know to understand the rest of the manuscript. Could you rephrase it? I suggest the following: ..."has an area of 11.34 km2 (based on the 2007 data of the Italian Glacier Inventory)",...

AC: We have modified the paragraph accordingly.

RC Lines 134 and 140-143 Page 6: "which is a typical evolution of valley glacier in the Alps". You can merge everything!

AC: We have merged this sentence with the previous one as suggested.

RC Lines 169-171 Page 7: delete this sentence

AC: we have deleted the sentence accordingly.

RC Line 172 Page 7: add "around midday".

AC: we have added "around midday" accordingly. We have also added "with 8/8 of the sky covered by stratocumulus clouds" as requested by reviewer 1.

RC Line 174 Page 7: Fig. 3a before 3b!

AC: The description of the 2014 survey has been placed before the 2016 survey, so the figure order is now correct.

RC Lines 175-179 Page 7:Could be condensed in: "Two different take-off and landing places were chosen in order to..." for instance.

AC: We have shortened the sentence accordingly.

RC Lines 194-195 Page 8: Here why not citing the original work on these methods?

AC: We have replaced the first reference with "Spetsakis and Aloimonos (1991)" and the second with "Furukawa and Ponce (2009)". The sentence was also moved to the description of the 2014 survey where the approach was first used in our study.

RC Lines 210-211 Page 9:Do you produce DEMs the same way than this study? If yes I would write: ... "to produce a DEM with the same method used in Immerzeel et al., 2014",... Otherwise the reader has to guess this, or misunderstand that this study is the first one to interpolate UAV point clouds to DEMs.

AC: We have modified the manuscript accordingly. This part has been moved to the description of the 2014 dataset, which is now at the top of the data section.

RC Line 224 Page 10: Is this relevant for later reading?

AC: we have deleted this part accordingly.

RC Line 252 Page 11: What does that mean? is it relevant for the reader?

AC: we have deleted this part accordingly.

RC Line 253 Page 11: Maybe put here a reference to a Fig. that show the location?

AC: we now show the location of the TLS standpoint in Figure 1 and added a reference in the text.

RC Lines 253-256 Page 11: I think this info might be better situated in the discussion, if you want to explain the advantages and drawbacks of this method!

AC: we have moved this part to the discussion section accordingly.

RC Line 262-265 Page 11: Similar comment than for L253 to L256.

AC: we have moved this part to the discussion section accordingly.

RC Line 267 Page 12: Rephrase as: "Prior the 2016 UAV surveys..." ?

AC: We have rephrased as "prior to the 2016 surveys" according to your comment.

RC Line 276 Page 12: place between brackets

AC: We have modified the manuscript accordingly.

RC Line 288 Page 13: 3b

AC: The image refers to figure 3a correctly now since the section about the 2014 dataset was moved to the top of the data section.

RC Lines 294-296 Page 13: L294 to 296 and L297 to 299 give the same information. I would recommend to remove this sentence.

AC: we have merged the two sentences at lines 294-298 as: "Early morning operations were preferred to avoid saturating camera pictures, as during this time of day the glacier is not yet directly illuminated by the sun, and to minimize blurring effects due to the UAV motion, since wind speed is at its lowest on glaciers during morning hours (Fugazza et al., 2015). "

RC Lines 298-299 Page 13: This breaks the link between the two other sentences. I suggest to remove it,

AC: we have removed this sentence accordingly.

RC Line 302-304 Page 13: I would remove this sentence as this is a well-known fact.

AC: we have removed the sentence accordingly.

RC Line 306-308 Page 13: I think it does not add value to the text to know the reason of a reduced surveyed area.

AC: we have removed the sentence accordingly.

RC Lines 330-340 Page 1-15: All these information have already been written in the previous sections. I think there is no need to duplicate the text.

AC: We have removed this part accordingly.

RC Line 360 Page 16: Be more precise to prepare the reader of the topic to come.

AC: We have replaced "Comparison between observations" with "Analysis of point clouds"

RC Line 360 Page 16: The highlighted information in this section is the size and the number of points generated per location. The number of point is depending on the size of the areas so I would prefer reading the the number of points per square meters (only) to be able to compare the different methods. This section, that has in the title the word "comparison", contains few information and the average reader probably expect more results. You could consider merging the 3.1 and 3.1.1.

AC: We have replaced the size with the area in the table, now merged with the table showing point density. However, we specify the absolute number of points (not per m2) to show the differences between sample locations.

RC Line 361 Page 16: Replace "data sets collected" with "point clouds generated"

AC: We have replaced the words accordingly.

RC Lines 365-366 Page 16: I suggest rephrasing as: "In our study, we refer to the work of Eltner et al., 2016 which applied criteria and metrics for comparing point clouds for different techniques, namely,..."

AC: we have deleted this sentence. We have rephrased the paragraph as: "The comparison between point clouds generated during the 2016 campaign had the aim of assessing their geometric quality before their application for the analysis of hazards. These evaluations were also expected to provide some guidelines for the organization

of future investigations in the field at the Forni Glacier and in other Alpine sites. Specifically, we analysed point density (points/m2) and completeness, i.e. % of area in the ray view angle. "

RC Line 367 Page 16: What about rephrasing such as: "that applies different criteria and metrics for point clouds generated from (i) UAV photogrammetry, (ii),..."?

AC: We have rephrased the paragraph as described in the previous comment.

RC Line 368 Page 16: criteria and metrics are vague terms. Can you please develop?

AC: The criteria used are those cited in the manuscript. We have therefore deleted this sentence.

RC Lines 372-375 Page 16: I think most people in this field of research know this. I would consider removing this sentence and the previous one.

AC: We have deleted this paragraph accordingly.

RC Lines 378-390 Page 16-17: This paragraph shows that the authors put effort in trying to explain the reader what the different point-cloud properties are. However, I think this is known from many people in the field and too detailed. My suggestions how to give the definition (in brakets) are below.

AC: We have deleted the paragraph accordingly.

RC Lines 391-392 Page 17: insert short description of criteria between brackets.

AC: we have replaced the description of point density and completeness as suggested. We have replaced the description of accuracy with a description of point cloud comparison in view of your major comment concerning the difference between accuracy and comparison. The paragraph now reads: "Specifically, we analysed point density (points/m2) and completeness, i.e. % of area in the ray view angle. Point density partly depends upon the adopted surveying technique, since it is controlled by the distance between sensor and surface and the obtainable spatial resolution. In SfM-

Photogrammetry, the latter property is affected by dense matching, while in TLS it can be set up as data acquisition input parameter. In this study, the number of neighbours N (inside a sphere of radius R=1 meter) divided by the neighbourhood surface was used to evaluate the local point density D in CloudCompare (www.cloudcompare.org). To understand the effect of point density dispersion (Teunissen, 2009), the inferior 12.5 percentile of the standard deviation of point density was also calculated. The use of these local metrics allowed to distinguish between point density in different areas, since this may largely change from one portion of surface to another. A further metric in this sense was point cloud completeness, referring to the presence of enough points to completely describe a portion of surface. In this study, the visual inspection of selected sample windows was used to identify occlusions and areas with lower point density. To analyse these properties, five regions were selected (see Fig. 5), located on the glacier topographic surface and characterized by different glacier features and the presence of hazards. The analysis of local regions was preferred to the analysis of the entire point clouds for the following reasons: 1) the incomplete overlap between point clouds obtained from different methods; 2) the opportunity to investigate the performances of the techniques in diverse geomorphological situations. Finally, we compared the point clouds in a pairwise manner within the same sample windows. Since no available benchmarking data set (e.g. accurate static GNSS data) was concurrently collected during the 2016 campaign, the TLS point cloud was used as a reference, as it less influenced by controlling factors (network geometry, object texture, lighting conditions). When comparing both photogrammetric data sets, the one obtained from UAV was used as reference because of the even distribution of point density within the sample windows. The presence of residual, non-homogenous geo-referencing errors in the data sets required a specific fine registration of each individual window, which was conducted in CloudCompare using the ICP algorithm (Pomerleau et al., 2016). Then, point clouds in corresponding sample windows were compared using the M3C2 algorithm implemented in CloudCompare (Lague et al., 2013). This solution allowed us to get rid of registration errors from the analysis, which could then be focused on the

capability of the adopted techniques to reconstruct the local geometric surface of the glacier in an accurate way. "

RC Line 393 Page 17: insert "are". I am not sure we can talk about "geomorphological properties" for something that looks more like "glacier features". If you don't like it, properties is the word to change ;-) And maybe hazard-prone areas? Remember that your paper is on hazards.

AC: we have removed both "are" to shorten the sentence. We have replaced "geomorphological properties" with "glacier features" and added "and the presence of hazards". The sentence now reads: "To analyse these properties, five regions were selected (see Fig. 5), located on the glacier topographic surface and characterized by different glacier features and the presence of hazards."

RC Lines 393-394 Page 17: This is a repetition of the first sentence.

AC: We have deleted this sentence accordingly.

RC Lines 394-397 Page 17: This is a repetition of the first two sentences.

AC: we have deleted the first sentence but kept the ones motivating the choice of analysing individual regions to answer the other Reviewer's comment.

RC Line 398 Page 17: Maybe change these words with "location" or synonym? sample window is not very clear.

AC: We have replaced "window" with "location" or "sample area" throughout the manuscript.

RC Line 410 Page 18: I would more refer to an area (m2).

AC: We now specify the area in Table 2.

RC Lines 411-413 Page 18: Here I would specify that not all location were surveyed (or only partially surveyed), by writing for instance "when available" or something alike.

Then the next two sentences are not needed anymore.

AC: We have deleted this sentence and added "where available" accordingly.

RC Lines 413-417 Page 18: This belongs to the discussion

AC: We have moved this part to the discussion section.

RC Lines 418-443 Pages 18-19: This would rather belong to a method section. It would be better for the reader to read a Method section first, were you detail all statistical calculation you will perform, and only display the results in the Results section.

AC: We have moved this part to the methods section and shortened it as suggested by reviewer and as described in the answer to your comment at line 391.

RC Line 444 Page 20: Here I would include what you actually see in this table (you wrote a full paragraph later in the text that can be summarized such as. "Although these values ranges from 103 to 2297 points/m2 depending on the surveying method, the density was sufficient for the reconstruction of the different surfaces (depicted on Fig. 7), except in the case of the location 5.") Figure 8 only displays few numbers of Table 5. So removing it would decrease your high number of Figures ;-)

AC: We have removed Figure 8 accordingly. We have rephrased the paragraph as: "The analysis of point density shows significant differences between the three techniques for point cloud generation (see Table 2). Values range from 103 to 2297 points/m2 depending on the surveying method, but the density was generally sufficient for the reconstruction of the different surfaces shown in Fig. 5, except for location 5. Terrestrial photogrammetry featured the highest point density, while UAV photogrammetry had the lowest. In relation to UAV photogrammetry, similar point densities were found in all sample locations, especially for the standard deviations that were always in the range 22-29 points/m2. Mean values were between 103-109 points/m2 in locations 2-4, while they were higher in location 5 (141 points/m2). Due to the nadir acquisition points, the 3D modelling of vertical/sub-vertical cliffs in location 1 was not possible. In

relation to TLS, a mean value of point density ranging from 141-391 points/m2 was found, with the only exception of location 5, where no sufficient data were recorded due to the position of this region with respect to the instrumental standpoint. Standard deviations ranged between 69-217 points/m2, moderately correlated with respective mean values."

RC Line 445-458 Page 20: From line 445, this belongs to the discussion.

AC: We have moved this part to the discussion section and shortened it.

RC Line 451 Page 20: You defined them already one time and using (i) are for enumerating a list and not a word. I would consider creating an accronym.

AC: we have removed ordinals accordingly.

RC Line 458 Page 20: This section could be shortened (written in a denser manner).

AC: we have shortened this section accordingly.

RC Lines 459-468 Pages 20-21: This might also go in a method section?

AC: This part was shortened and moved to the methods section as described in the answer to your comment at line 391.

RC Lines 469-476 Page 21: This section could be shortened and set around line 444. You can either put the numbers in the text or in a table (better) but not both, because this makes a repetition.

AC: We have deleted this part as numbers are already shown in the table

RC Lines 477- 485 Page 21: This paragraph belongs to the discussion and I think could be more concise.

AC: Part of this paragraph was kept in the results section as it only shows a numeric comparison. Relevant considerations were made in the discussion section.

RC Lines 486-492 Pages 21-22: Same as above paragraph. It belongs to the discussion. It also should be more concise.

AC: Part of this paragraph was kept in the results section as it only shows a numeric comparison. Relevant considerations were made in the discussion section.

RC Line 493 Page 22: Are the two Figures showing similar results but for two different location? If yes, I think only having one of them is enough and I would remove Fig. 10.

AC: We have removed Figure 10 accordingly.

RC Line 497 Page 22: Fig.11 should come first. Refer to another figure to understand a figure is not great. It means that one would be enough. Is Figure 12 really needed?Refer to another figure to understand a figure is not great. It means that one would be enough. Is Figure 12 really needed? What results? What did you do in this figure? How did you merged two point clouds from different methods? where they corresponding? How is that better? The first question should go in the method section, the second in the results and the third in the discussion ;-)

AC: We have removed Figure 12. The sentence "Results are also satisfying in gently sloped areas, as it can be observed in windows 2 and 3" has been removed. We now specify how the merging was performed in the methods section, and discuss the improvements of merging in the discussion section.

RC Line 499 Page 22: what terrestrial sensor? Inserting a camera in cavities and take pictures in the cavity? why this has not been possible with terrestrial photogrammetry?

AC: The sentence lacked clarity and has been deleted.

RC Lines 500-503 Page 22: This paragraph could be merged with the previous one. The outcome of the last 2-3 paragraphs is on the advantages and drawbacks of each methods. This should be written clearly and in the discussion section.

AC: The paragraph was shortened and merged in the new discussion section.

RC Lines 504-514 Pages 22-23: Same comment as above. And please it would be

nice if you select only the useful information for the reader.

AC: The paragraph has been shortened and merged in the new discussion section.

RC Line 516 Page 23: Here I understand that fractures and faults are not well reconstructed and therefore can be well detected. For what is this information? Can you specify? Otherwise I would think that where you have partial reconstruction is where you have fractures and faults and this is where the hazards are located.

AC: This sentence lacked clarity and has therefore been deleted.

RC Lines 518-522 Page 23: I think these two sentences are not necessary for the reader, which could get confused.

AC: We have removed this part accordingly.

RC Line 522 Page 23: Insert "such as".

AC: We have rewritten this sentence as: "Since no available benchmarking data set (e.g. accurate static GNSS data) was concurrently collected during the 2016 campaign, the TLS point cloud was used as a reference, as it less influenced by controlling factors (network geometry, object texture, lighting conditions)." in view of your major comment on accuracy.

RC Line 524 Page 23: Replace "to compare" with "for comparison"

AC: We have removed this part to shorten the manuscript.

RC Line 526 Page 23: the point clouds in a pairwise manner. (or reformulate in a similar way)

AC: We have rephrased this sentence as "Finally, we compared the point clouds in a pairwise manner within the same sample areas.".

RC Line 527 Page 23: selected location

AC: We have replaced "sample windows" with "selected locations" accordingly.

RC Line 530 Page 23: Is that the ICP from cloud compare as well?

AC: We have added "in CloudCompare" at the end of the sentence.

RC Lines 532-539 Pages 23-34: This algorithm has already been used in the studies of x and x for instance and proved to be suitable for ... . This is enough for the reader to know why you used that one. There is no need to go in details and explaning what the advantage of this algorithm is.

AC: We have deleted this part to shorten the manuscript.

RC Line 534 Page 23: I do not understand the meaning of "positive direction of distances" can you please explain or reword?

AC: This part has been deleted as suggested in your previous comment.

RC Line 541 Page 24: The second part of the sentence make the reader think that your reference point cloud is actually bad.

AC: We have deleted the second part of the sentence as it lacked clarity.

RC Lines 518-543 Pages 23-24: From L518 to here is information that could go in the method section.

AC: We have moved this information in the methods section accordingly, see the answer to your comment at line 391.

RC Line 546 Page 24: We can see this in the table.

AC: We have rephrased this paragraph as: "In terms of point cloud distance (see Table 3), the comparison between TLS and terrestrial photogrammetry resulted in a high similarity between point clouds, with no large differences between different sample windows. Conversely, the comparison between TLS and UAV photogrammetry and terrestrial and UAV photogrammetry provided significantly worse results, which may be summarized by the RMSEs in the range 21.1-37.7 cm and 20.7-30.4 cm, respectively.

The worse values were both obtained in the analysis of window 2, which mostly represents a vertical surface, while the best agreement was found within window 3 which is less inclined. As the UAV flight was geo-referenced on a set of GCPs with an RMSE of 40.5 cm, the ICP co-registration may have not totally compensated the existing bias."

RC Line 548 Page 24: departures? what does that mean in this context? or do you mean outliers?

AC: We have replaced the word "departures" with "differences".

RC Line 558 Page 24: This is italian-english ;-) Would "make use of" or "using" work in the sentence instead?

AC: We have replaced "recurring to" with "placing". The sentence has been also moved to the discussion section.

RC Line 561 Page 25: Not right word. see comment above.

AC: We have removed this sentence and moved considerations in the discussion section.

RC Line 571 Page 25: not only... there is also 2007 and 2014 as well as 2014 and 2016. Maybe change the title to fix this issue?

AC: we have changed the title to "glacier thickness change" accordingly.

RC Lines 574-576 Page 25: delete this part.

AC: We have deleted this part accordingly.

RC Line 576 Page 25: When different area of interest are used for computation, it is very hard to compare and make use of these results. Table 8 does it very well, so I would not include Table 7. AC: We have removed table 7 accordingly.

RC Lines 576-579 Page 25: No need to talk about maximum extension of DEMs. I think it is normal that authors display all data available

[Figure]

AC: We have removed reference to the maximum extension of DEMs.

RC Lines 579-581 Page 25: I think the reader understood this already from the previous paragraphs

AC: We have removed this sentence accordingly.

RC Lines 581-583 Pages 25-26: Delete this sentence

AC: We have shortened the paragraph, which now reads: "After DEM co-registration, the resulting shifts reported in Table 1 were applied to each 'slave' DEM, including the entire glacier area. Then the elevations of the 'slave' DEM were subtracted from the corresponding elevations of the 'master' DEM to obtain the so called DEM of Differences (DoD). Over a reference area common to all three DEMs (Fig. 1), we estimated the volume change and its uncertainty following the method proposed in Howat et al. (2008), which expresses the uncertainty of volume change as the combination of the standard deviation computed from the residual elevation difference over stable areas, and the truncation error implicit when substituting the integral in volume calculation with a finite sum, according to Jokinen and Geist (2010).", and moved it to the methods section.

RC Line 588 Page 26:So where are the results? Table 8?

AC: the results are provided in table 4 (former table 8) but since the part has been moved to the methods section the reference to the table is only provided in the results section.

RC Lines 589-591 Page 26:I would not include this. The reader does not get much information out of it. And "only lost 15m" is a point of view ;-)

AC: we have deleted this sentence accordingly.

RC Line 602 Page 26:This paragraph is unclear to me and the numbers are questionable. How can (L.595) an ice thickness change be of -40 to -50m over 2 years? And a

few lines below (L.598) have a glacier thinning of 10m over the same amount of time?

AC: "2014" at line 595 should have read "2007", while "10 m" at line 599 was lacking a minus sign. However, to improve clarity, we have rewritten the paragraph. We use minus signs when we employ the term "change" and no sign when we talk about thinning since the term already implies a loss. The paragraph now reads:"The Forni Glacier tongue was affected by substantial thinning throughout the observation period. Between 2007 and 2014, the largest thinning occurred in the eastern section of the glacier tongue, with changes persistently below –30 m, whereas the upper part of the central tongue only thinned by 10/18 m. The greatest ice loss occurred in correspondence with the normal faults localized in small areas at the eastern glacier margin (see Fig. 8a), with local changes generally below -50 m and a minimum of -66.80 m, owing to the formation of a lake. Conversely, between 2014 and 2016 the central and eastern parts of the tongue had similar thinning patterns, with average changes of -10 m. The greatest losses are mainly found in correspondence with normal faults, with a maximum change of -38.71 m at the terminus and local thinning above 25 m on the lower medial moraine. The ring fault at the left margin of the central section of the tongue also shows thinning of 20/26 m. In the absence of faults, little thinning occurred instead on the upper part of the medial moraine, where a thick debris cover shielded ice from ablation, with changes of -2/-5 m (see Fig. 8c). Considering a common reference area (see Fig. 1, table 4), an acceleration of glacier thinning seems to have occurred over recent years over the lower glacier tongue, from -4.55± 0.24 ma-1 in 2007-2014 to -5.20± 1.11 ma-1 in 2014-2016, also confirmed by the value of -4.76± 0.29 ma-1 obtained from the comparison between 2007 and 2016. Comparing the first two DoD, the trend seems to be caused by the increase in collapsing areas (Fig.8a,b). This equates to 13.46 ±0.14 million m3 of ice lost over the entire study period. "

RC Line 606 Page 27: replace "fused together" with "merged" and "merged" with "resulting.

AC: We have replaced "fused together" with "merged" and "merged" with "final". New

information has been added as described in the answer to your major comment concerning Subsection 3.3 (Lines 571 to 602)

RC Line 609 Page 27: Is that Figure 12? If yes please precise.

AC: The RGB colored point cloud can now be seen in Fig. 5

RC Line 635 Page 28:Would be nice to start talking about the upper transect, then middle, and finally lower. This paragraph gives new information and does not discuss or directly links to the results found. Please make it more clear why you now describe the glacier tongue in detail.

AC: The entire section has been deleted as not strictly relevant to this study.

RC Line 645 Page 28: Same comment than above. This new information has nothing to do in the discussion section.

AC:The entire section has been deleted as not strictly relevant to this study.

RC Line 648 Page 28: geometry?

AC: we have deleted the second part of the sentence. The sentence has also been moved to the introduction section where we specify why we mapped these specific hazards.

RC Line 665 Page 29:what is a repeat pace?

AC: we have rephrased the sentence as: "It is likely that the terminus will recede along the fault system on the eastern medial moraine and following the ring faults at the eastern and western margins, increasing the occurrence of hazardous phenomena in these areas."

RC Line 668 Page 29: two words

AC: we have deleted this part to shorten the manuscript.

RC Line 698 Page 30: roughness?

AC: we have restructured the conclusion section and thus the word has been deleted.

RC Line 715 Page 31: There is not much luck in science ;-) ... " due to higher camera location/ image angles"...

AC: We have removed this sentence as the conclusion section has been restructured.

RC Line 742 Page 32: For all references: Please, - check the spelling of the journals (uppercase or not). - put doi or DOI but not a mix - put the doi at the end of the citation (after the year) and without the webpage - check that all articles have a volume a page and a doi. - check that your proceedings references all have the same information in the right order (in the journal guidelines).A For all references, we have checked the journal spelling replacing lowercase with uppercase letters and kept the full name for consistency. We have added the doi were it was missing (always lowercase) and placed it always at the end of the reference, after the year. Now every entry has volume, pages and doi, except: Fioletti et al., 2016; Mugnier, 2005; Riccardi et al. 2010; Teunissen, 2009 and conference proceedings except for the ones from 2016 - No doi available Berthier et al., 2016; Gagliardini et al., 2011; Howat et al. 2008; Ryan et al., 2017; Urbini et al., 2017 - in Geophysical Research Letters, Annals of Geophysics and Frontiers in Earth Science the page number is not indicated as each article is numbered separately starting from 1. Only the letter or article number is reported and we have added that in the manuscript.

Please also note the supplement to this comment:
https://www.nat-hazards-earth-syst-sci-discuss.net/nhess-2017-198/nhess-2017-198-AC2-supplement.pdf
* * *
**Fig. 1.**

[Figure]

| Aircraft type | Swinglet CAM, Commercial platform |
|---|---|
| Digital Camera | Canon Ixus 127 HS |
| Camera technical features | 16 Megapixel, focal length 4.3 mm |
| GNSS antenna | GPS only |
| Weight (incl. payload) | 0.50 Kg |
| Battery time | 30 minutes |

[Figure]

| Aircraft type | Customized, with Tarot frame 650 size, VR Brain 5.2 Autopilot & APM Arducopter 3.2.1 Firmware |
|---|---|
| Digital camera | Canon Powershot ELPH 320 HS |
| Camera technical features | 16 Megapixel, focal length 4.3 mm |
| GNSS antenna | GPS+GLONASS (Galileo compatible) |
| Weight (incl. pay-load) | 2.75 Kg |
| Battery time | 20-25 minutes |

**Fig. 2.**

[Figure]

**Fig. 3.**

**Fig. 4.**

[Figure]

**Fig. 5.**

[Figure]

**Fig. 6.**

**Supplement:**

**Combination of UAV and terrestrial photogrammetry to assess rapid glacier evolution and map glacier hazards**

Fugazza, Davide1; Scaioni, Marco2; Corti, Manuel2; D'Agata, Carlo3; Azzoni, Roberto Sergio3; Cernuschi, Massimo4; Smiraglia, Claudio1; Diolaiuti, Guglielmina Adele3

1Department of Earth Sciences 'A.Desio', Università degli studi di Milano, 20133 Milano Italy

2Department of Architecture, Built Environment and Construction Engineering, Politecnico di Milano, 20133 Milano Italy

3Department of Environmental science and policy (DESP), Università degli studi di Milano, 20133 Milano Italy

4Agricola 2000 S.C.P.A., 20067 Tribiano (MI) Italy

Correspondence to: Marco Scaioni (marco.scaioni@polimi.it)

**Abstract**

Tourists and hikers visiting glaciers all year round face hazards such as the rapid formation of collapses at the terminus, typical of such a dynamically evolving environment. In this study, we analysed the potential of different survey techniques to analyze hazards of the Forni glacier, an important geo-site located in Stelvio Park (Italian Alps). We carried out surveys in the ablation season 2016 and compared point clouds generated from UAV, close range photogrammetry and terrestrial laser scanning (TLS). To investigate the evolution of glacier hazards and evaluate the glacier thinning rate, we also used UAV data collected in 2014 and a DEM from an aerial photogrammetric survey of 2007. We found that the integration between terrestrial and UAV photogrammetry is ideal to map hazards related to the glacier collapse, while TLS is affected by occlusions and logistically complex in glacial terrain. Photogrammetric techniques can therefore replace TLS for glacier studies and UAV-based DEMs hold potential to become a standard tool to investigate the glacier geodetic mass balance. Based on our datasets, an increase in the size of collapses was found over the study period, and the glacier thinning rates went from  $4.55 \pm 0.24$  ma-1 between 2007 and 2014 to  $5.20 \pm 1.11$  ma-1 between 2014 and 2016.

**1** Introduction**

Glacier and permafrost-related hazards can be a serious threat to humans and infrastructure in high mountain regions (Carey et al., 2014). The most catastrophic cryospheric hazards are generally related to the outburst of water, either through breaching of moraine- or ice-dammed lakes or from the englacial or subglacial system, causing floods and debris flows. Ice avalanches from hanging glaciers can also have serious consequences for downstream populations (Vincent et al., 2015), as well as debris flows caused by the mobilization of accumulated loose sediment on steep slopes (Kaab et al.,

| Style Definition: Normal: English (United Kingdom),      |
|----------------------------------------------------------|
| Border: Top: (No border), Bottom: (No border), Left: (No |
| border), Right: (No border), Between : (No border)       |

| Style Definition                                                  |
|-------------------------------------------------------------------|
| Style Definition                                                  |
| Style Definition                                                  |
| Style Definition: List Paragraph                                  |
| Style Definition: Caption                                         |
| Style Definition: Header                                          |
| Style Definition: Footer                                          |
| Style Definition: Comment Text                                    |
| Style Definition: Balloon Text                                    |
| Style Definition: RC items                                        |
| Style Definition: Revision                                        |
| Formatted                                                         |
| Formatted: Space After: 14 pt                                     |
| Deleted: conditions of                                            |
| Formatted: Italian (Italy)                                        |
| Formatted: Space Before: 0 pt, After: 14 pt                       |
| Formatted: Space After: 14 pt                                     |
| Formatted: Italian (Italy)                                        |
| Formatted: Space Before: 0 pt, After: 14 pt                       |
| Deleted: ), by describing local surface features           |
| Deleted: evaluating the glacier melting rate. The analyses |
| Deleted: and digital elevation models (DEMs) from two             |
| Deleted: tongue carried out                                       |
| Deleted: 2016 with Unmanned Aerial Vehicles (UAVs),               |
| Deleted: obtained in 2007                                         |
| Deleted: survey. On the area covered by the 2016 survey,   |
| Deleted: of -                                                     |
| Deleted: 15                                                       |
| Deleted: were found in                                            |
| Deleted: -2016, while the mean thickness change of the     |
| Deleted: -2016 was -10.40±2.60 m. UAV-based DEMs w                |
| Formatted: Font: Not Bold                                         |
| Deleted: ¶                                                        |
| Formatted: English (United Kingdom)                               |
| Moved (insertion) [1]                                             |
| Deleted: 49¶                                                      |

2005a). Less severe hazards, but still particularly threatening for mountaineers are the detachment of seracs (Riccardi et al., 2010) or the collapse of ice cavities (Gagliardini et al., 2011; Azzoni et al., submitted). While these processes are in part typical of glacial and periglacial environments, there is evidence that climate change is increasing the likelihood of specific hazards (Kaab et al., 2005a). In the European Alps, accelerated formation and growth of proglacial moraine-dammed lakes has been reported in Switzerland, amongst concern of possible overtopping of moraine dams provoked by ice avalanches (Gobiet et al., 2014). Lee avalanches themselves can be more frequent as basal sliding is enhanced by the abundance of meltwater in warmer summers (Clague, 2013). Glacier and permafrost retreat, which have been reported in all sectors of the Alps (Smiraglia et al., 2015; Fischer et al., 2014; Gardent et al, 2014; Harris et al., 2009), are a major cause of slope instabilities which can result in debris flows, by debuttressing rock and debris flanks and promoting the exposure of unconsolidated and ice-cored sediments (Keiler et al., 2010; Chiarle et al., 2007). Glacier downwasting is also increasing the occurrence of structural collapses and while not directly threatening human lives, sustained negative glacier mass balance can also cause shortages of water for industrial, agricultural and domestic use and energy production, negatively affecting even populations living away from glaciers. F
[revised manuscript text omitted]
to assess their ability to monitor glacier hazards: UAV photogrammetry, terrestrial photogrammetry and TLS. The aims were: (1) comparing UAV- and terrestrial photogrammetric products, acquired in 2016 against the TLS point cloud; (2) identifying glacier-related hazards and their evolution between 2014-2016 using the merged point cloud from UAV and terrestrial photogrammetry and UAV orthophotos; and 3) investigating ice thickness changes between 2014-2016 and 2007-2016 by comparing the two UAV DEMs and a third DEM obtained from stereo-processing of aerial photos captured in 2007.

**2 Study Area**

The Forni Glacier (see Fig. 1) has an area of 11.34 km2 based on the 2007 data from the Italian Glacier Inventory, (Smiraglia et al., 2015), an altitudinal range between 2501 and 3673 m a.s.l. and a North-North-Westerly aspect. The glacier, retreated markedly since the little ice age (LIA), when its area was 17.80 km2 (Diolaiuti & Smiraglia, 2010), with an acceleration of the shrinking rate in the last three decades, typical of valley glaciers in the Alps (Diolaiuti et al., 2012, D'Agata et al.; 2014). It has also undergone profound changes in dynamics in recent years, including the loss of ice flow from the eastern accumulation basin towards its tongue and the evidence of collapsing areas on the eastern tongue (Azzoni et al., submitted). One such area, hosting a large ring fault (see Fig. 2d) prompted an investigation carried out with Ground Penetrating Radar (GPR) in October 2015, but little evidence of

[revised manuscript text omitted]

|                                                                                                                                                                                                                                                                                                                                                                                                                                                                                                                                                                                                                                                                                                                                                                                                                                                                                                                                                                                                                                                                                                                                                                                                                                                                                                                                                                                                                                                                                                                                                                                                                                                                                                                                                                                                                                                                                                                                                                                                                                                                                                                               | operations see Schofield & Breach, 2007).                                                                                                                                                                                                                                                                                                                                                                                                                                                                                                                                                                        |
|-------------------------------------------------------------------------------------------------------------------------------------------------------------------------------------------------------------------------------------------------------------------------------------------------------------------------------------------------------------------------------------------------------------------------------------------------------------------------------------------------------------------------------------------------------------------------------------------------------------------------------------------------------------------------------------------------------------------------------------------------------------------------------------------------------------------------------------------------------------------------------------------------------------------------------------------------------------------------------------------------------------------------------------------------------------------------------------------------------------------------------------------------------------------------------------------------------------------------------------------------------------------------------------------------------------------------------------------------------------------------------------------------------------------------------------------------------------------------------------------------------------------------------------------------------------------------------------------------------------------------------------------------------------------------------------------------------------------------------------------------------------------------------------------------------------------------------------------------------------------------------------------------------------------------------------------------------------------------------------------------------------------------------------------------------------------------------------------------------------------------------|------------------------------------------------------------------------------------------------------------------------------------------------------------------------------------------------------------------------------------------------------------------------------------------------------------------------------------------------------------------------------------------------------------------------------------------------------------------------------------------------------------------------------------------------------------------------------------------------------------------|
| l                                                                                                                                                                                                                                                                                                                                                                                                                                                                                                                                                                                                                                                                                                                                                                                                                                                                                                                                                                                                                                                                                                                                                                                                                                                                                                                                                                                                                                                                                                                                                                                                                                                                                                                                                                                                                                                                                                                                                                                                                                                                                                                             | Deleted: Such                                                                                                                                                                                                                                                                                                                                                                                                                                                                                                                                                                                                    |
| l                                                                                                                                                                                                                                                                                                                                                                                                                                                                                                                                                                                                                                                                                                                                                                                                                                                                                                                                                                                                                                                                                                                                                                                                                                                                                                                                                                                                                                                                                                                                                                                                                                                                                                                                                                                                                                                                                                                                                                                                                                                                                                                             | Deleted: stable glacier areas or                                                                                                                                                                                                                                                                                                                                                                                                                                                                                                                                                                                 |
| 1                                                                                                                                                                                                                                                                                                                                                                                                                                                                                                                                                                                                                                                                                                                                                                                                                                                                                                                                                                                                                                                                                                                                                                                                                                                                                                                                                                                                                                                                                                                                                                                                                                                                                                                                                                                                                                                                                                                                                                                                                                                                                                                             | Deleted: (see Fig. 6).                                                                                                                                                                                                                                                                                                                                                                                                                                                                                                                                                                                           |
|                                                                                                                                                                                                                                                                                                                                                                                                                                                                                                                                                                                                                                                                                                                                                                                                                                                                                                                                                                                                                                                                                                                                                                                                                                                                                                                                                                                                                                                                                                                                                                                                                                                                                                                                                                                                                                                                                                                                                                                                                                                                                                                               | Formatted: Indent: First line: 0 cm, Space Before: 0 pt, After: 14 pt                                                                                                                                                                                                                                                                                                                                                                                                                                                                                                                                     |
| l                                                                                                                                                                                                                                                                                                                                                                                                                                                                                                                                                                                                                                                                                                                                                                                                                                                                                                                                                                                                                                                                                                                                                                                                                                                                                                                                                                                                                                                                                                                                                                                                                                                                                                                                                                                                                                                                                                                                                                                                                                                                                                                             | Deleted: ,                                                                                                                                                                                                                                                                                                                                                                                                                                                                                                                                                                                                       |
| l                                                                                                                                                                                                                                                                                                                                                                                                                                                                                                                                                                                                                                                                                                                                                                                                                                                                                                                                                                                                                                                                                                                                                                                                                                                                                                                                                                                                                                                                                                                                                                                                                                                                                                                                                                                                                                                                                                                                                                                                                                                                                                                             | Deleted: (                                                                                                                                                                                                                                                                                                                                                                                                                                                                                                                                                                                                       |
|                                                                                                                                                                                                                                                                                                                                                                                                                                                                                                                                                                                                                                                                                                                                                                                                                                                                                                                                                                                                                                                                                                                                                                                                                                                                                                                                                                                                                                                                                                                                                                                                                                                                                                                                                                                                                                                                                                                                                                                                                                                                                                                               | Deleted: to be used as reference point for GNSS surveys in the Forni Glacier region. The                                                                                                                                                                                                                                                                                                                                                                                                                                                                                                                  |
| l                                                                                                                                                                                                                                                                                                                                                                                                                                                                                                                                                                                                                                                                                                                                                                                                                                                                                                                                                                                                                                                                                                                                                                                                                                                                                                                                                                                                                                                                                                                                                                                                                                                                                                                                                                                                                                                                                                                                                                                                                                                                                                                             | Deleted: of this point were already known                                                                                                                                                                                                                                                                                                                                                                                                                                                                                                                                                                        |
| l                                                                                                                                                                                                                                                                                                                                                                                                                                                                                                                                                                                                                                                                                                                                                                                                                                                                                                                                                                                                                                                                                                                                                                                                                                                                                                                                                                                                                                                                                                                                                                                                                                                                                                                                                                                                                                                                                                                                                                                                                                                                                                                             | Deleted: geodetic/                                                                                                                                                                                                                                                                                                                                                                                                                                                                                                                                                                                               |
| ł                                                                                                                                                                                                                                                                                                                                                                                                                                                                                                                                                                                                                                                                                                                                                                                                                                                                                                                                                                                                                                                                                                                                                                                                                                                                                                                                                                                                                                                                                                                                                                                                                                                                                                                                                                                                                                                                                                                                                                                                                                                                                                                             | Deleted: frame                                                                                                                                                                                                                                                                                                                                                                                                                                                                                                                                                                                                   |
|                                                                                                                                                                                                                                                                                                                                                                                                                                                                                                                                                                                                                                                                                                                                                                                                                                                                                                                                                                                                                                                                                                                                                                                                                                                                                                                                                                                                                                                                                                                                                                                                                                                                                                                                                                                                                                                                                                                                                                                                                                                                                                                               | Deleted: and were used for geo-referencing all other points measured with GNSS                                                                                                                                                                                                                                                                                                                                                                                                                                                                                                                            |
| and a statement of the | Moved up [5]: took place on 28 th August 2014, using a SwingletCam fixed wing aircraft (see Fig. 3a).                                                                                                                                                                                                                                                                                                                                                                                                                                                                                          |
|                                                                                                                                                                                                                                                                                                                                                                                                                                                                                                                                                                                                                                                                                                                                                                                                                                                                                                                                                                                                                                                                                                                                                                                                                                                                                                                                                                                                                                                                                                                                                                                                                                                                                                                                                                                                                                                                                                                                                                                                                                                                                                                               | Deleted: This commercial platform developed by SenseFly, with basic technical features reported in Table 1, carries a Canon Ixus 127 HS compact digital camera. The UAV was flown in autopilot mode with a relative flying height of approximately 380 m above the average glacier surface, which resulted in an average GSD of 11.9 cm. The flight plan was organized by using the proprietary software eMotion, by which the aircraft follows predefined waypoints with a nominal along-strip overlap of 70%; sidelap was not regular because of the varying surface topography, but ranged around 60%. |
| /                                                                                                                                                                                                                                                                                                                                                                                                                                                                                                                                                                                                                                                                                                                                                                                                                                                                                                                                                                                                                                                                                                                                                                                                                                                                                                                                                                                                                                                                                                                                                                                                                                                                                                                                                                                                                                                                                                                                                                                                                                                                                                                             | Moved up [6]: Flight operations started at 07:44 AM and ended at 08:22 AM.                                                                                                                                                                                                                                                                                                                                                                                                                                                                                                                                |
|                                                                                                                                                                                                                                                                                                                                                                                                                                                                                                                                                                                                                                                                                                                                                                                                                                                                                                                                                                                                                                                                                                                                                                                                                                                                                                                                                                                                                                                                                                                                                                                                                                                                                                                                                                                                                                                                                                                                                                                                                                                                                                                               | Deleted: Early morning operations were preferred as during this time of day the glacier is not yet directly illuminated by the sun, thus diffuse illumination predominates over the glacier surface, and wind speed is at its lowest (Fugazza et                                                                                                                                                                                                                                                                                                                                                          |
| 1                                                                                                                                                                                                                                                                                                                                                                                                                                                                                                                                                                                                                                                                                                                                                                                                                                                                                                                                                                                                                                                                                                                                                                                                                                                                                                                                                                                                                                                                                                                                                                                                                                                                                                                                                                                                                                                                                                                                                                                                                                                                                                                             | Moved up [7]: ). Both the terminal parts of the central and                                                                                                                                                                                                                                                                                                                                                                                                                                                                                                                                               |
| 1                                                                                                                                                                                                                                                                                                                                                                                                                                                                                                                                                                                                                                                                                                                                                                                                                                                                                                                                                                                                                                                                                                                                                                                                                                                                                                                                                                                                                                                                                                                                                                                                                                                                                                                                                                                                                                                                                                                                                                                                                                                                                                                             | Deleted: The considerable difference in area covered duri                                                                                                                                                                                                                                                                                                                                                                                                                                                                                                                                                 |
| 1                                                                                                                                                                                                                                                                                                                                                                                                                                                                                                                                                                                                                                                                                                                                                                                                                                                                                                                                                                                                                                                                                                                                                                                                                                                                                                                                                                                                                                                                                                                                                                                                                                                                                                                                                                                                                                                                                                                                                                                                                                                                                                                             | Moved up [8]: Consequently, a global bias in the order of                                                                                                                                                                                                                                                                                                                                                                                                                                                                                                                                                 |
| 1                                                                                                                                                                                                                                                                                                                                                                                                                                                                                                                                                                                                                                                                                                                                                                                                                                                                                                                                                                                                                                                                                                                                                                                                                                                                                                                                                                                                                                                                                                                                                                                                                                                                                                                                                                                                                                                                                                                                                                                                                                                                                                                             | Deleted: 2.2 2014 UAV photogrammetric survey                                                                                                                                                                                                                                                                                                                                                                                                                                                                                                                                                                     |
| 1                                                                                                                                                                                                                                                                                                                                                                                                                                                                                                                                                                                                                                                                                                                                                                                                                                                                                                                                                                                                                                                                                                                                                                                                                                                                                                                                                                                                                                                                                                                                                                                                                                                                                                                                                                                                                                                                                                                                                                                                                                                                                                                             | Deleted: following the methods outlined in Sec. 2.1.1, wit                                                                                                                                                                                                                                                                                                                                                                                                                                                                                                                                                |
| ١                                                                                                                                                                                                                                                                                                                                                                                                                                                                                                                                                                                                                                                                                                                                                                                                                                                                                                                                                                                                                                                                                                                                                                                                                                                                                                                                                                                                                                                                                                                                                                                                                                                                                                                                                                                                                                                                                                                                                                                                                                                                                                                             | Formatted: Space Before: 0 pt, After: 14 pt                                                                                                                                                                                                                                                                                                                                                                                                                                                                                                                                                                      |
| 1                                                                                                                                                                                                                                                                                                                                                                                                                                                                                                                                                                                                                                                                                                                                                                                                                                                                                                                                                                                                                                                                                                                                                                                                                                                                                                                                                                                                                                                                                                                                                                                                                                                                                                                                                                                                                                                                                                                                                                                                                                                                                                                             | Deleted: (Compagnia Generale Riprese Aeree)                                                                                                                                                                                                                                                                                                                                                                                                                                                                                                                                                                      |
| 1                                                                                                                                                                                                                                                                                                                                                                                                                                                                                                                                                                                                                                                                                                                                                                                                                                                                                                                                                                                                                                                                                                                                                                                                                                                                                                                                                                                                                                                                                                                                                                                                                                                                                                                                                                                                                                                                                                                                                                                                                                                                                                                             | Deleted:                                                                                                                                                                                                                                                                                                                                                                                                                                                                                                                                                                                                         |
| 1                                                                                                                                                                                                                                                                                                                                                                                                                                                                                                                                                                                                                                                                                                                                                                                                                                                                                                                                                                                                                                                                                                                                                                                                                                                                                                                                                                                                                                                                                                                                                                                                                                                                                                                                                                                                                                                                                                                                                                                                                                                                                                                             | Deleted: coordinate                                                                                                                                                                                                                                                                                                                                                                                                                                                                                                                                                                                              |
| 1                                                                                                                                                                                                                                                                                                                                                                                                                                                                                                                                                                                                                                                                                                                                                                                                                                                                                                                                                                                                                                                                                                                                                                                                                                                                                                                                                                                                                                                                                                                                                                                                                                                                                                                                                                                                                                                                                                                                                                                                                                                                                                                             | Deleted: the                                                                                                                                                                                                                                                                                                                                                                                                                                                                                                                                                                                                     |
|                                                                                                                                                                                                                                                                                                                                                                                                                                                                                                                                                                                                                                                                                                                                                                                                                                                                                                                                                                                                                                                                                                                                                                                                                                                                                                                                                                                                                                                                                                                                                                                                                                                                                                                                                                                                                                                                                                                                                                                                                                                                                                                               |                                                                                                                                                                                                                                                                                                                                                                                                                                                                                                                                                                                                                  |

Deleted. (f

Mario datum (Mugnier, 2005). Heights were converted from ellipsoidal to geodetic using the official software for datum transformation in Italy (Verto ver. 3), which is distributed by the Italian Geographic Military Institute (IGMI). The final vertical accuracy reported by BLOM C.G.R. is  $\pm$  3 m. The only processing step performed within this study was the datum conversion to ITRS2000, using a seven-parameter similarity transformation based on a local parameter set provided by IGMI.

**4 Methods**

4.1 Analysis of point clouds from the 2016 campaign: UAV/terrestrial photogrammetry and TLS The comparison between point clouds generated during the 2016 campaign had the aim of assessing their geometric quality before their application for the analysis of hazards. These evaluations were also expected to provide some guidelines for the organization of future investigations in the field at the Forni Glacier and in other Alpine sites. Specifically, we analysed point density (points/m2) and completeness, i.e. % of area in the ray view angle. Point density partly depends upon the adopted surveying technique, since it is controlled by the distance between sensor and surface and the obtainable spatial resolution. In SfM-Photogrammetry, the latter property is affected by dense matching, while in TLS it can be set up as data acquisition input parameter. In this study, the number of neighbours N (inside a sphere of radius R=1 meter) divided by the neighbourhood surface was used to evaluate the local point density D in CloudCompare (www.cloudcompare.org). To understand the effect of point density dispersion (Teunissen, 2009), the inferior 12.5 percentile of the standard deviation  $\sigma$  of point density was also calculated. The use of these local metrics allowed to distinguish between point density in different areas, since this may largely change from one portion of surface to another. A further metric in this sense was point cloud completeness, referring to the presence of enough points to completely describe a portion of surface. In this study, the visual inspection of selected sample locations was used to identify occlusions and areas with lower point density.

To analyse these properties, five regions were selected (see Fig. 5), located on the glacier topographic surface and characterized by different glacier features and the presence of hazards: 1) Glacial cavity composed by subvertical and fractured surfaces over 20 m high, and forming a typical semicircular shape; 2) glacial cavity over 10 m high with the same typical semi-circular shape as location 1, covered by fine- and medium-size rock debris; 3) normal fault over 10 m high; 4) highly-collapsed area covered by fine- and medium-size rock debris and rock boulders; and 5) planar surface with a normal fault covered by fine- and medium-size rock debris and rock boulders. The analysis of local regions was preferred to the analysis of the entire point clouds for the following reasons: 1) the incomplete overlap between point clouds obtained from different methods; 2) the opportunity to investigate the performances of the techniques in diverse geomorphological situations.

Finally, we compared the point clouds in a pairwise manner within the same sample locations. Since no available benchmarking data set (e.g. accurate static GNSS data) was concurrently collected during the 2016 campaign, the TLS point cloud was used as a reference, as it less influenced by controlling factors (network geometry, object texture, lighting conditions). When comparing both photogrammetric data sets, the one obtained from UAV was used as reference because of the even distribution of point density within the sample locations. The presence of residual, non-homogenous geo-referencing errors in the data sets required a specific fine registration of each individual sample location, which was conducted in CloudCompare using the ICP algorithm (Pomerleau et al., 2016). Then, point clouds in corresponding sample areas were compared using the M3C2 algorithm implemented in CloudCompare (Lague et al., 2013). This solution allowed us to get rid of registration errors from the analysis, which could then be focused on the capability of the adopted techniques to reconstruct the local geometric surface of the glacier in an accurate way.

4.2 Merging of UAV and close-range photogrammetric point clouds

To improve coverage of different glacier surfaces, including planar areas and normal faults, photogrammetric point clouds from the 2016 campaign were merged. Prior to point cloud merging, a preliminary co-registration was performed on the basis of the ICP algorithm in CloudCompare. Regions common to both point clouds were used to minimize the distances between them and find the best co-registration. The point cloud from UAV photogrammetry, which featured the largest extension, was used as reference during co-registration, while the other was rigidly transformed to fit with it. After this task, both original point clouds resulted aligned into the same reference system. In order to get rid of redundant points and to obtain a homogenous point density, the merged point cloud (see Fig. 5) was subsampled keeping a minimum distance between adjacent points of 20 cm. The final size of this data set is approximately 4.4 million points, which represents a manageable data amount on up-to-date computers. The colour RGB information associated to each point in the final point cloud was derived by averaging the RGB information of original points in the subsampling volumes. While this operation resulted in losing part of the original RGB information, it helped provide a realistic visualization of the topographic model, which can aid the interpretation of glacier hazards.

**4.2 Glacier hazard mapping**

The investigation of glacier hazards was conducted by considering datasets from 2014 and 2016. In 2014, only the point cloud and UAV orthophoto were available, while in 2016 the point cloud obtained by merging UAV and close-range photogrammetric data sets was used in combination with the UAV orthophoto. In this study, we focused on ring faults and normal faults, which were manually delineated by using geometric properties from the point clouds while color information from orthophotos was used as a cross-check. On point clouds, mapping is based on visual inspection of vertical displacements following faulting or subsidence. On orthophotos, both types of structures also generally appear as linear features in contrast with their surroundings. A
[revised manuscript text omitted]

| Deletea: | reconstruction |
|----------|----------------|
| Deleted: | window         |
| Deleted: | The limit      |

**Deleted:** the inferior 12.5% percentile was between 49-62 **Deleted:** 49¶ from 141-391 points/m2 was found, with the only exception of location 5, where no sufficient data were recorded due to the position of this region with respect to the instrumental standpoint. Standard deviations ranged between 69-217 points/m2, moderately correlated with respective mean values. The analysis of the completeness of surface reconstruction also revealed some issues related to the adopted techniques (see Fig. 6). Specifically, TLS suffered from severe occlusions which prevented acquisition of data in the central part of the sample area, while UAV photogrammetry was able to reconstruct the upper portion of the sample area but not the vertical cliff. Only terrestrial photogrammetry acquired a large number of points in all areas,

In terms of point cloud distance (see Table 3), the comparison between TLS and terrestrialphotogrammetry resulted in a high similarity between point clouds, with no large differences between different sample areas. Conversely, the comparison between TLS and UAV photogrammetry and terrestrial and UAV photogrammetry provided significantly worse results, which may be summarized by the RMSEs in the range 21.1-37.7 cm and 20.7-30.4 cm, respectively. The worse values were both obtained in the analysis of location 2, which mostly represents a vertical surface, while the best agreement was found within location 3 which is less inclined. As the UAV flight was geo-referenced on a set of GCPs with an RMSE of 40.5 cm, the ICP co-registration may have not totally compensated the existing bias.

**5.2 Glacier-related hazards and risks**

The tongue of Forni glacier hosts a variety of hazardous structures. While most collapsed areas are normal faults, two large ring fault systems can be identified: the first, located in the eastern section (see Fig. 2d and 7), covered an area of  $25.6 \times 10^3$  m2 and showed surface lowering up to 5 m in 2014. This area was not surveyed in 2016, since field observation did not show evidence of further subsidence. Conversely, the ring fault that only emerged as a few semi-circular fractures in 2014 grew until cavity

[revised manuscript text omitted]

In this study, the uncertainty of the 2016 UAV dataset (40.5 cm RMSE on GCPs and 21.1-37.7 cm RMSE when compared against TLS) was slightly higher than previously reported in high mountain glacial environments (Immerzeel et al., 2014; Gindraux et al., 2017; Seier et al., 2017). Contributing factors might include the sub-optimal distribution and density of GCPs (Gindraux et al., 2017), the delay between the UAV surveys as well as between UAV and other surveys and the lack of coincidence between GCP placement and the UAV flights. This means the UAV photogrammetric reconstruction was affected by ice ablation and glacier flow, which on Forni Glacier range between 3-5 cm day-1 (Senese et al., 2012) and 1-4 cm day-1, respectively (Urbini et al., 2017). We thus expect a combined 3-day uncertainty on the 2016 UAV dataset between 10 and 20 cm, and lower on GCPs considering reduced ablation owing to their placement on boulders. A further contribution to the error budget of GCPs might stem from the intrinsic precision of GNSS/theodolite measurements and image resolution. The comparison between close-range photogrammetry and TLS, was less affected by glacier change as data were collected one day apart and the RMSE of 6-10.6 cm is in line with previous findings by Kaufmann and Landstaedter (2008). To improve the accuracy of UAV photogrammetric blocks, a better distribution of GCPs or switching to an RTK system should be considered, while closerange photogrammetry could benefit from measuring a part of the photo-stations as proposed in Forlani et al. (2014), instead of placing GCPs on the glacier surface.

The uncertainty in UAV photogrammetric reconstruction also factored in the relatively high standard deviation still present after the coregistration between DEMs in areas outside the glacier (2.22 m between 2014 and 2016). Another important factor here is the morphology of the coregistration area, i.e. the outwash plain, still subject to changes owing to the inflow of glacier meltwater and sediment reworking. The final accuracy of our UAV photogrammetric products was nevertheless adequate to investigate ice thickness changes over 2 years, while the integration with close-range photogrammetry was required to investigate hazards related to the collapse of the glacier terminus.

We conducted UAV surveys under different meteorological scenarios, and obtained adequate results with early-morning operations with 0/8 cloud cover and midday flights with 8/8 cloud cover. Both scenarios can provide diffuse light conditions allowing to collect pictures suitable for photogrammetric processing, but camera settings need to be carefully adjusted beforehand (O'Connor et al., 2017). If early morning flights are not feasible in the study area for logistical reasons or when surveying east-exposed glaciers, the latter scenario should be considered.

In our pilot study, we covered part of the Forni glacier tongue, and only investigated hazards related to the glacier collapse. Our maps can help identify safer paths where mountaineers and skiers can visit the glacier and reach the most important summits. However, the increase in collapse structures owing to climate change requires multi-temporal monitoring. A comprehensive risk assessment should also cover the entire glacier, to investigate the probability of serac detachment and provide an estimate of the glacier mass balance with the geodetic method. While our integrated approach using a multicopter and terrestrial photogrammetry should be preferred to investigate small individual ice bodies, fixed-wing UAVs, ideally equipped with an RTK system and ability to tilt the camera off-nadir, might be the platform of choice to cover large distances (see e.g. Ryan et al., 2017), potentially reducing the number of flights and solving issues with GCP placement. Such platforms could help collect sufficient data for hazard management strategies up to the basin scale in Stelvio National Park and other sectors of the Italian Alps, eventually replacing aerial LiDAR surveys. Cost analyses (Matese et al., 2015) should also be performed to evaluate the benefits of improved spatial resolution and DEM accuracy of UAVs compared to aerial and satellite surveys and choose the best approach for individual cases.

**7 Conclusions**

In our study, we compared point clouds generated from UAV photogrammetry, close-range photogrammetry and TLS to assess their quality and evaluate the potential in mapping and describing

glacier hazards such as ring faults and normal faults, by carrying out a specific campaign in summer 2016. In addition, we employed orthophotos and point clouds from a UAV survey conducted in 2014 to analyze the evolution of glacier hazards and a DEM from an aerial photogrammetric survey conducted in 2007 to investigate glacier thickness changes between 2014 and 2016. The main findings of our study include:

- UAVs and terrestrial photogrammetric surveys provide reliable performances in glacial environments, outperform TLS in terms of logistics and costs, and are more flexible in relation to meteorological conditions.
- UAV and terrestrial photogrammetric blocks can be easily integrated providing more information than individual techniques to help identify glacier hazards.
- UAV-based DEMs can be employed to estimate thickness changes but improvements are necessary in terms of area covered and accuracy to calculate the glacier geodetic mass balance of large glaciers.
- The Forni Glacier is rapidly collapsing with an increase in ring faults size, providing evidence of climate change in the region.
- The glacier thinning rate increased owing to collapses to 5.20 ma-1 between 2014 and 2016.

The maps produced from the combined analysis of UAV and terrestrial photogrammetric point clouds can be made available through GIS web portals of Stelvio National Park or Lombardy region (http://www.geoportale.regione.lombardia.it/). A permanent monitoring programme should be setup to help manage risk in the area, issuing warnings and assisting mountain guides in changing hiking and ski routes as needed. The analysis of glacier thickness changes suggests a feedback mechanism which should be further analysed, with higher thinning rates leading to increased occurrence of collapses, with additional release of meltwater. Glacier downwasting is also of relevance for risk management in

the protected area, providing valuable data to assess the increased chance of rockfalls and to improve forecasts of glacier meltwater production.

While our test was conducted on one of the largest glaciers in the Italian Alps, the integrated photogrammetric approach is easily transferrable to similar sized and much smaller glaciers, where it would be able to provide a comprehensive assessment of hazards and mass balance and become useful in decision support systems for natural hazard management. In larger regions, UAVs hold the potential to become the platform of choice but their performances and cost-effectiveness compared to aerial and satellite surveys need to be further evaluated.

**Competing interests**

The authors declare that they have no conflict of interest.

**Acknowledgements**

This study was funded by DARAS, the department for autonomies and regional affairs of the presidency of the council of the Italian government. The authors acknowledge the central scientific committee of CAI (Club Alpino Italiano – Italian Alpine Club) and Levissima San Pellegrino S.P.A. for funding the UAV quadcopter. The authors also thank Stelvio Park Authority for the logistic support and for permitting the UAV surveys and IIT Regione Lombardia for the provision of the 2007 DEM. Acknowledgements also go to the GICARUS lab of Politecnico Milano at Lecco Campus for providing the survey equipment. Finally, the authors would also like to thank Tullio Feifer, Livio Piatta, and Andrea Grossoni for their help during field operations.

Azzoni, R.S Deleted: 49¶

|                                                                                                                                                                                              | Formatted                                             |          |
|----------------------------------------------------------------------------------------------------------------------------------------------------------------------------------------------|-------------------------------------------------------|----------|
|                                                                                                                                                                                              | Deleted: Zerboni, A., Maugeri, M., Smiraglia, C. and  |   |
| Aleradi I. Chichards E. Cress, N. Linner, A.M. Needs, E. and Grand, A. UAM sheets surrouter                                                                                                  | Formatted                                             |          |
| with oblique images: first analysis on data acquisition and processing. International Archives of the                                                                                        | Formatted                                             |          |
| Photogrammetry, Remote Sensing and Spatial Information Sciences, XXIII ISPRS Congress, Prague,                                                                                               | Deleted: Barazzetti L., Remondino F., Scaioni M. and  |          |
| Czech Republic, 12–19 July 2016, 41-B1, 835-842, 2016, doi: 10.5194/isprs-archives-XLI-B1-835-                                                                                               | Formatted                                             |          |
| 2016.                                                                                                                                                                                        | Formatted                                             |          |
| Arrani B.S. Fusarra D. Zamara M. Zuseli M. D'Asata C. Marsana D. Camusahi M.                                                                                                                 | Deleted: sensing                                      |          |
| Smiraglia C, and Diolajuti, G A.: Recent structural evolution of Forni Glacier tongue (Ortles-Cevedale                                                                                       | Formatted                                             |          |
| Group, Central Italian Alps), submitted to Journal of Maps.                                                                                                                                  | Deleted: environment                           |          |
|                                                                                                                                                                                              | Formatted                                             |          |
| Berthier, E., Arnaud, Y., Kumar, R., Ahmad, S., Wagnon, P. and Chevallier, P.: Remote sensing                                                                                                | Formatted                                             |          |
| estimates of glacier mass balances in the Himachal Pradesh (Western Himalaya, India), Remote                                                                                                 | Formatted                                             |          |
| Sensing of Environment , 108, 527-558, 2007 , doi: 10.1010/j.rse.2000.11.017 .                                                                                                 | Formatted                                             |          |
| Berthier, E., Cabot, V., Vincent, C. and Six, D.: Decadal Region-Wide and Glacier-Wide Mass                                                                                                  | Formatted                                             |   |
| Balances Derived from Multi-Temporal ASTER Satellite Digital Elevation Models.Validation over the                                                                                            | Deleted: sensing                                      |          |
| Mont-Blanc Area, Frontiers in Earth Science, 4, 63, 2016, doi: 10.3389/feart.2016.00063.                                                                                                     | Formatted                                             | _ (      |
| Phardusi A. Sam I. Akankaha Martin Torres, E.L. and Kumar, D.: UAVs as remote sensing                                                                                                        | Deleted: environment                                  |          |
| platform in glaciology: Present applications and future prospects. Remote Sensing of Environment.                                                                                            | Formatted                                             |          |
| 175, 196-204, 2016 , doi: 10.1016/j.rse.2015.12.029.                                                                                                                           | Deleted: 2016a                                        |          |
|                                                                                                                                                                                              | Formatted                                             |          |
| Blasone G., Cavalli M. and Cazorzi F.: Debris-Flow Monitoring and Geomorphic Change Detection                                                                                                | Deleted: Bhardwaj, A., Sam, L., Bhardwaj, A. and Mart | ín       |
| Combining Laser Scanning and Fast Photogrammetric Surveys in the Moscardo Catchment (Eastern
Italian Alas) in: Lolling G. Arattano M. Bineldi M. Giustolici O. Moreabel J.C. and Grant G. | Formatted                                             |          |
| (eds) Engineering Geology for Society and Territory 3. Springer, Cham. 51-54, 2015, doi:                                                                                                     | Moved up [9]: J.:                                     |          |
| 10.1007/978-3-319-09054-2_10                                                                                                                                                                 | Deleted: LiDAR remote sensing of the cryosphere: Pres | en       |
|                                                                                                                                                                                              | Deleted: ). In                                        |          |
| Carey, M., McDowell, G., Huggel, C., Jackson, M., Portocarrero, C., Reynolds, J.M. and Vicuna, L.:                                                                                           | Deleted: JC.,                                         |          |
| in: Haeberli, W. and Whiteman, C. (Eds.). Snow and Ice-related Hazards, Risks and Disasters, Elsevier.                                                                                       | Deleted: - Volume                                     |          |
| 219-261, 2014, doi: 10.1016/B978-0-12-394849-6.00008-1.                                                                                                                                      | Deleted: .                                            |          |
|                                                                                                                                                                                              | Deleted: .                                            |          |
| Clague, J.: Glacier Hazards, in: Bobrowski, P. (Ed.), Encyclopedia of Natural Hazards, Springer, 400-                                                                                        | Deleted: , 2015                                       |          |
| 403, 2013, doi: 10.1007/770-1-4020-4399-4_130.                                                                                                                                               | Moved (insertion) [11]                                |          |
| Chiarle, M., Iannotti, S., Mortara, G. and Deline, P.: Recent debris flow occurrences associated with                                                                                        | Formatted                                             |          |
| glaciers in the Alps, Global and Planetary Change, 56, 123-136, 2007, doi:                                                                                                                   | Deleted: Bocchiola, D. and Diolaiuti, G. Evidence of  |          |
| 10.1016/j.gloplacha.2006.07.003.                                                                                                                                                      | Formatted                                             |          |
| Colomina, I. and Molina, P.: Unmanned aerial systems for photogrammetry and remote sensing: A                                                                                                | Formatted                                             |          |
| review, ISPRS Journal of Photogrammetry and Remote Sensing, 92, 79-97, 2014, doi:                                                                                                            | Formatted                                             |          |
| 10.1016/j.isprsjprs.2014.02.013                                                                                                                                                              | Formatted                                             |          |
| D'Agata, C., Bocchiola, D., Maragno, D., Smiraglia, C. and Diolaiuti, G.A.; Glacier shrinkage driven                                                                                         | Deleted: Theor Appl Cimatol                           |   |
| by climate change during half a century (1954–2007) in the Ortles-Cevedale group (Stelvio National                                                                                           | Formatted                                             | _        |
| Park, Lombardy, Italian Alps), Theoretical and Applied Cimatology, 116, 169-190, 2014, doi:                                                                                                  | Formatted                                             |          |
| 10.1007/s00704-013-0938-5                                                                                                                                                                    | Polotodi 400                                          |   |
|                                                                                                                                                                                              |                                                       |          |

Moved down [10]: ., Senese, A.,

.... .... ....

Dewez, T.J.B., Leroux, J. and Morelli, S.: Cliff collapse hazard from repeated multicopter UAV acquisitions: return on experience, The International Archives of the Photogrammetry, Remote Sensing and Spatial Information Sciences, XXIII ISPRS Congress, Prague, Czech Republic, 12–19 July 2016, 41-B5, 805-811, 2016, doi: 10.5194/isprs-archives-XLI-B5-805-2016

Diolaiuti, G.A. and Smiraglia, C.: Changing glaciers in a changing climate: how vanishing geomorphosites have been driving deep changes in mountain landscapes and environments, Géomorphologie : Relief, Processus, Environnement, 2, 131-152, 2010, doi: 10.4000/geomorphologie.7882.

Diolaiuti, G.A., Bocchiola, D., D'Agata, C. and Smiraglia, C.: Evidence of climate change impact upon glaciers' recession within the Italian Alps, Theoretical and Applied Climatology, 109,429-445, 2012, doi: 10.1007/s00704-012-0589-y

Eltner, A., Kaiser, A., Castillo, C., Rock, G., Neugirg, F. and Abellán, A.: Image-based surface reconstruction in geomorphometry – merits, limits and developments. *Earth Surface Dynamics*, 4, 359-389, 2016, doi: 10.5194/esurf-4-359-2016.

Fioletti, M., Bonetti, M., Smiraglia, C., Diolaiuti, G.A., Breganze, C., dal Toso, M. and Facco, L.: Indagini radar per lo studio delle caratteristiche endoglaciali del ghiacciaio dei Forni in alta Valtellina, Neve e Valanghe, 87, 40-45, 2016

Fischer, M., Huss, M., Barboux, C. and Hoelzle, M.: The new Swiss Glacier Inventory SGI2010: relevance of using high-resolution source data in areas dominated by very small glaciers, Arctic, Antarctic and Alpine Research, 46,933-945, 2014, doi: 10.1657/1938-4246-46.4.933.

Fischer, M., Huss, M. and Hoelzle, M.: Surface elevation and mass changes of all Swiss glaciers 1980–2010, The Cryosphere, 9, 525-540, 2015, doi: 10.5194/tc-9-525-2015

Forlani, G., Pinto, L., Roncella, R. and Pagliari, D.: Terrestrial photogrammetry without ground control points, Earth Science Informatics, 7, 71-81, 2014, doi: 10.1007/s12145-013-0127-1.

Fugazza, D., Senese, A., Azzoni, R.S., Smiraglia, C., Cernuschi, M., Severi, D. and Diolaiuti, G.A.: High-resolution mapping of glacier surface features. The UAV survey of the Forni glacier (Stelvio national park, Italy), Geografia Fisica e Dinamica Quaternaria, 38, 25-33, 2015, doi: 10.4461/GFDQ.2015.38.03,

Furukawa, Y. and Ponce, J.: Accurate Camera Calibration from Multi-View Stereo and Bundle Adjustment, International Journal of Computer Vision, 84, 257–268, 2009, doi: 10.1007/s11263-009-0232-2.

Garavaglia, V., Diolaiuti, G.A., Smiraglia, C., Pasquale, V. and Pelfini, M.: Evaluating Tourist Perception of Environmental Changes as a Contribution to Managing Natural Resources in Glacierized areas: A Case Study of the Forni Glacier (Stelvio National Park, Italian Alps), Environmental Management, 50, 1125-1138, 2012, doi: 10.1007/s00267-012-9948-9,

|           |                                                          |   |
|-----------|----------------------------------------------------------|----------|
|           | Formatted                                                | [        |
|           | Formatted                                                | []       |
|           | Deleted: relief, processus, environnement                |          |
|           | Formatted                                                | [        |
|           | Deleted: .                                               |          |
|           | Formatted                                                | ٢.       |
|           | Formatted                                                |   |
|           | Deleted: Theor. Appl. Climatol.,                         |          |
|           | Formatted                                                | []       |
|           | Deleted: DOI                                             |          |
|           | Formatted                                                | [        |
|           | Deleted: , 2012                                          |          |
|           | Formatted                                                | ٦.       |
|           | Formatted                                                |   |
|           | Formatted                                                | [        |
|           | Formatted                                                | [        |
|           | Deleted: 2016,                                    | -        |
| $\langle$ | Formatted                                                | [        |
|           | Formatted                                                |   |
|           | Formatted                                                |   |
|           | Formatted                                                | [        |
|           | Formatted                                                | (        |
|           | Formatted                                                |   |
|           | Formatted                                                |   |
|           | Deleted: Fonstad, M.A., Dietrich, J.T., Courville, B.C., | (        |
|           | Formatted                                                | [        |
|           | Deleted: Geogr. Fis. Dinam. Quat                         |          |
|           | Formatted                                                | [        |
|           | Deleted: DOI                                             |          |
|           | Formatted                                                | [        |
|           | Deleted: , 2015                                          |          |
|           | Formatted                                                | [        |
|           | Formatted                                                | [        |
|           | Deleted: Fugazza, D., Senese, A., Azzoni, R.S., Maugeri, | [        |
|           | Formatted                                                | [        |
|           | Deleted: management                                      |          |
|           | Formatted                                                | (        |
|           | Formatted                                                | (        |
|           | Formatted                                                | [        |
|           | Deleted: 49¶                                             |          |

[remaining 31,511 characters of this post omitted]

---

## Author Response (AR2)

Dear Editor,

We have carefully revised the manuscript taking into account all suggestions by the reviewers.

Specifically, we the manuscript has been proofread by a professional mother tongue consultant to improve its readability. We have clarified the advantages of point cloud merging through a figure and a paragraph in the results section; we have also revised and numerically quantified the uncertainty of UAV volume calculations and revised the text with a consistent use of 'uncertainty' in place of 'accuracy' and 'thickness changes' in place of 'mass balance'; we have added explanations for the algorithms used in point cloud comparison. Finally, we have restructured the discussion section and added references to the articles suggested by one of the reviewers. In addition, we have improved the figures and tables thanks to the reviewers' suggestions.

We think that the manuscript has improved and that the final result is clearer and more readable. We are grateful to the reviewers for their detailed and helpful comments.

You will find in the following text the detailed responses to the reviewers' suggestions and comments with relevant changes made to the manuscript directly reported in our answers. Finally, a marked up version of the manuscript with all changes is provided.

We hope that the revised version of the manuscript can now meet the reviewers' expectations and can be accepted for publication; otherwise, we are open to new improvements.

Best regards, Davide Fugazza & coauthors. We have prepared a point by point response to the reviewer's comments. In the following text, reviewer's comments are reported as RC and highlighted in italics, our answers as AC in plain text while our changes to the text are in bold black.

**RC Review of the manuscript:**

Combination of UAV and terrestrial photogrammetry to assess rapid glacier evolution and conditions of glacier hazards. Fugazza et al.

**General comments:**

In this manuscript, the authors describe and analyse geomorphological features on the tongue of a hazard-prone glacier in the Italian alps with the help of three different (close-range) remote sensing methods. They found that UAV- and terrestrial photogrammetry are the best surveying techniques to assess ice thickness change and map glacier hazards. Compared to the first manuscript, the authors reduced significantly the length of the manuscript which was preconized by both reviewers, set the aims more clearly and reorganized the different sections in a much better way. However, there are still several points of the paper that require substantial improvements, such as:

RC 1) This version of the paper in my opinion, is poorly written. I had large difficulties understanding several sections of the manuscripts, due to the use of unconventional words and wrong sentence construction. As this is not my mother-tongue either, I could not give suggestions for every case and did not highlighted all of them. A English proof reading is surely needed before considering this paper for publication.

AC: The manuscript has been proofread by a professional mother tongue consultant. Since minor errors were found, we have not reported all changes to the manuscript here, but they can be seen in the tracked manuscript version.

RC 2) The individual methods and comparison of the methods used as suggested in this version of the manuscript are not new. The only aspect that could have been interesting and relatively new, is the merging of different datasets. However, the authors only merged the terrestrial and UAV photogrammetry point clouds, without giving a quantitative explanation of how this merged point cloud is much better than the terrestrial one alone. Looking at Figure 6, and the very sparse point cloud they obtained from UAV photogrammetry compared to terrestrial photogrammetry, I doubt that the merging of both point clouds did make a big difference. It would however be good if the authors could give more information on this point. Moreover in this study, this merged point cloud is only used to map the hazards (along with the orthophoto), which I guess, could have also been done with the point cloud from one method only (the terrestrial photogrammetry). So I am not sure if this merged point cloud was really necessary. I think the authors should emphasis more on the scientific value of this study.

AC: The reason why we chose to compare UAV and terrestrial SfM-photogrammetry was because they were the two lower cost techniques, and we have added this information in the text. While terrestrial photogrammetry has the highest point density, it only covers those portions of the glacier surfaces that could be depicted from ground-based photo-stations. The point cloud obtained from terrestrial photogrammetry covers the terminus and makes it possible to investigate vertical and sub-vertical areas. The point cloud derived from UAV-

photogrammetry covers a larger part of the glacier and allows to investigate sub-horizontal areas such as the ring faults on the central and eastern tongue. In general, the UAV and terrestrial photogrammetry point clouds only partially overlap. Therefore, merging these point clouds enables to cover a larger area than with individual point clouds and investigate different types of hazards. We have added a figure showing the glacier area covered with both techniques and discuss this aspect in the results section, where we have added a paragraph which reads: "In terms of spatial coverage, considering the entire surface examined using each technique outside the sample locations, the UAV survey extended over the widest area (0.59 km2), including part of the proglacial plain, the entire terminus and the glacier tongue up to the collapsed area on the central part, but with data gaps on the vertical and sub-vertical walls (see Fig. 6a). The point cloud obtained from terrestrial photogrammetry covered approximately a third of the area surveyed with the UAV (see fig. 6b), including the full glacier terminus at very high spatial resolution, with the exception of a few obstructed parts, while the TLS point cloud covered the terminus, although with some holes due to the obstructions.". In addition, we proposed here the use of such a methodology because we think it may have a larger applications in other case studies related to Alpine glaciers.

RC 3) I think that the authors wrote a lot about the differences between the methods they used, but very little refer to other literature. There are a lot of papers (below only a few for example) comparing point clouds and DEMs generated from different surveying methods.

a. Baltsavias, 1999: A comparison between photogrammetry and laser scanning
 b. Rayburg et al. 2009: A comparison of digital elevation models generated from different data sources

c. Naumann et al. 2013: Accuracy comparison of digital surface models created by unmanned aerial systems imagery and terrestrial laser scanner

In most sections of the results and discussion, the authors report results that have already been found in other studies. For instance in the discussion section, where the advantages and drawbacks from all methods are explained, the authors cite publications that found the same results (but several years ago). As several papers are already stating these, I think this information should not be discussed anymore, but taken as granted. I suggest the authors to integer older and newer publications related to the comparison of point clouds and DEM specifically.

AC: The publications we have cited in the discussion section related to the point density/completeness comparison of different techniques are from 2012, 2016 (2) and 2017, therefore on average rather recent, and mainly describe surveys of non-glacial environments. Thus, we have decided to keep the references to these publications, to provide confirmation for the findings and conclusions drawn there. We have now restructured the discussion section separating it into four sub-sections 1) point density and completeness; 2) point cloud uncertainty; 3) logistics and costs; 4) additional remarks. In the last section, we have added references to Rayburg et al. (2009) and Naumann et al. (2013) among others and also compare the techniques we used against ALS and aerial photogrammetry in terms of logistics, cost, accuracy and spatial resolution but have not cited Baltsavias (1999), as it is a much older publication.

RC 4) To my point of view, the authors are often making statements such as: "The UAVbased DEMs hold the potential to become a standard tool to investigate geodetic mass balance", but the authors did not try to do this and the publications that succeeded to do it are very rare. Another one: "The final accuracy of our UAV photogrammetric products was nevertheless adequate to investigate ice thickness change over two years,...". The reader don't know for what it is "adequate", as the authors found DEM errors over two meters (for these two years). They do not give any percentage error that would mean on the total melt. Moreover, the authors do not state why they need ice thickness data. We only know that ice thickness change is related to an increase of natural hazards.

I think in general, the manuscript needs to be more carefully written, with results presented in a more quantitative robust way.

AC: We have replaced 'mass balance' with 'ice thickness', which is what we investigated. As concerns the uncertainty in ice thickness change, in all three cases it is below 3% of the total ice volume change (see answer to your comment at line 325). As the accuracy depends on the application of the data, we have decided to avoid using this term and we have added percentage values of volume uncertainty in Table 4. We further comment them in section 5.3, where we have added: **"In all DoDs, the uncertainty in ice thickness change results in less than 3% of the respective volume change (see Table 4)**." at the end of the section. Finally, we have deleted the word 'adequate' and specified the percentage uncertainty. Thus, we have replaced 'The final accuracy of our UAV photogrammetric products was nevertheless adequate to investigate ice thickness changes over 2 years' with 'UAV photogrammetric products permitted us to investigate ice volume changes over 2 years with an uncertainty of 2.60%'.

The ice thickness data are useful for hazard management because glacier downwasting is a hazard per se, as noted by Kaab et al. (2005). In fact, changes in glacier length, area and volume cause variations in water resources and their availability for human consumption, hydropower generation and irrigation. We have better clarified why glacier downwasting contributes to natural hazards, by replacing: "Glacier downwasting is also increasing the occurrence of structural collapses and while not directly threatening human lives, sustained negative glacier mass balance can also cause shortages of water for industrial, agricultural and domestic use and energy production, affecting even populations living away from glaciers." with "Glacier downwasting causes changes in water resources, with an initial increase in discharge due to enhanced melt and a long-term reduction, affecting drinking water supply, irrigation and hydropower production (Kaab et al., 2005b) and is also increasing the occurrence of structural collapses (Azzoni et al., 2017).".

Besides, we have explained that ice thickness data are useful for hazard management in the introduction, where have replaced "In particular, the advantages of UAV and terrestrial SfM-Photogrammetry, and the possibility of data fusion to support hazard management strategies in glacial environments needs to be investigated and assessed." with "In particular, the advantages of UAV and terrestrial SfM-photogrammetry and the possibility of data fusion and volume change estimation to support hazard management strategies in glacial environments needs to be investigated and assessed." With "In particular, the advantages of UAV and terrestrial SfM-photogrammetry and the possibility of data fusion and volume change estimation to support hazard management strategies in glacial environments needs to be investigated and assessed."

RC: I think the manuscript requires again major revision before being considered for publication in NHESS. More specific and short comments are reported in the supplementary material as .pdf.

Comments on Figures and Tables:

**RC Figure 1:**

- I would zoom in more as there is a lot of space under the reference area, and eventually show the positions of the terrestrial pictures.

- It's hard to make the difference between the green triangle and the green points. Can you change the colour? If you do this, I think you can also remove the " (in 2016 two different...)" in the caption, because we can see it on the map.

- Is the number after +/- based on 1 standard deviation? Or 2?

AC:

- We have zoomed in the figure but have chosen not to show the terrestrial pictures as there are too many of them and they would make the map too cluttered.
- We have changed the color of the green point which is now brown
- It is not clear what the last comment refers to, as there are no numbers with +/- in this or other figures.

**RC Figure 2:**

- It is very hard for the reader to have an idea about the size of the feature. Could you insert a scale?

- This Figure shows the hazardous features on the glacier but these are not the ones that you survey and studied. So I am wondering what kind of information the reader gets out of this Figure. I would maybe recommend to put them on the right side of Fig.1 where there is some space left before the caption's end, and delete the Figure 2.

AC: These photographs were taken in 2016 during the field campaign between 29th August and 1st September. They do represent features that we have surveyed and studied in our analysis of glacier hazards (section 5.2). Figure 1 already provides a lot of information and we believe that adding more makes it less clear. We have therefore decided to keep this Figure on its own. Unfortunately, these figures were taken without a scale (person and other object). While we could add a vertical scale from the analysis of point clouds, this information belongs to the results, thus we have decided not to introduce it here. The height of the features is reported in the results section 5.2. In the previous version of the manuscript, we referenced panels c and d. We have added references to the other three panels (a, b and e) in section 5.2 to help the reader understand the size of these features.

**RC Figure3:**

- Very clean Figure
- I would only centre the b in the white case

AC: We have centered the 'b' as suggested.

RC Figure 4:

- I think this figure does not give much information to the reader. My suggestion is that you either put the location of the images on Fig 1., or that you merge a and b.

- Caption: Small question in (a). In the text you say that you took 134 pictures. Are they all displayed here? I have the feeling that they don't, so did they all align in the software?

AC: All photos have been displayed here and all of them have been aligned. As explained in the answer to your previous comment to Figure 1, there are too many images to be shown on Figure 1. The reason why it seems that fewer photos are displayed is that some photos were captured from very close locations and they look as from a single camera station. Besides, some photos were collected from the same position but with the camera rolled 90 degrees to provide a more suitable configuration for camera self-calibration. This are standard rules in close-range photogrammetry (see Luhmann et al., 2014, Close-range Photogrammetry).

Following your comment at line 134, we have chosen to delete this figure and we now show the coverage of UAV, terrestrial photogrammetry and the merged point cloud in a new figure (fig.6)

**RC Figure 5:**

- The background image is very dark. It would be nice to see more lighter colours

- Also here on the image it is hard to see how big these features are. As you don't have much space, I suggest you make a similar scale everywhere and that you add it on top of the 100 scale bar and you show the number here.

AC: We have changed the background, showing lighter colors. We have further added a vertical approximate scale on each sample location panel representing a height of 10 m and have added this bar on top of the horizontal distance bar

**RC Figure 6:**

- Caption: Maybe you could mention that the scale bars don't have the same scale?

AC: We have followed the suggestion of reviewer 2 by producing a uniform scale for each panel, although this makes it impossible to understand how different features are represented by each technique. We kindly ask the editor to choose which version is best suited for point density analysis.

**RC Figure 7:**

- Caption: "L" in Location

- In the caption and in the main text you use orthomosaic and orthophotos. Please stick to one term.

- I would add after (a) and (b), situation in 2014 (situation in 2016) or something similar, so that you don't start with a year.

AC: - 'L' is now uppercase.

- We now use the term '**orthophoto'** throughout the manuscript.
- We have added 'Collapse structures in' after each letter and before the year.

RC Figure 8:

- Maybe consider to change the total ice thickness change! In the manuscript you are stating values to -30 and -50m that we don't see on the map! The reader needs to calculate if he/she has the yearly values.

AC: The purpose of Fig. 8 is to enable a comparison between different DEM pairs, which is only possible if the DoDs are normalized, using yearly rates instead of absolute values. To help the reader switch between absolute and yearly values, we have added the yearly rates between parentheses in the text (see answers to your comments at lines 375, 377, 382 and 384.

**RC Table 1:**

- In the caption, it would be nice if you could state something like:" DEM 2007 from aerial multispectral survey, DEM2014 and DEM 2016 from UAV photogrammetry." So that the reader do not need to go back to the text to remember which DEM is which.

AC: We have added this information in the caption as suggested.

RC Table 2:

- This table definitely need some adjustments, because it's not easy to read.

1. The "sample window" text could be rotated 90°, and use all space above the numbers (merge cells).

2. The meaning of k should be explained in the caption

3. You could choose 3 abbreviations (UAV, TP and TLS) in the table and explain them in the caption, so that the text is less squeezed.

4. Watch that you use capital letter at the beginning or you text everywhere.

5. The numbers with 1645+/- 54 need to be in one line! Otherwise the reader asks: what is the number below it? There seems to be a bit of space left on the right side of your table to enlarge it (till the level of your caption right).

AC:

- 1) We have rotated the text by 90° and merged cells to use all spaces above the numbers.
- 2) The meaning of k is now explained in the caption as "k stands for thousands of points".
- 3) We have used the suggested abbreviations and explained them in the caption.
- 4) The first letter of each column title is now uppercase
- 5) The numbers separated by +- signs are now on the same line.

We have rewritten the caption as: "Table 2: Area and number of points in each sample window on the Forni Glacier terminus, mean and standard deviation of local point density and number of points above the lower 12.5% percentile in each window. k stands for thousands of points. UAV refers to UAV photogrammetry, TP to terrestrial photogrammetry and TLS to terrestrial laser scanning."

RC Table 3:

1. The "sample window" text could be rotated 90°, and use all space above the numbers.

- 2. Same comment as above with the abbreviations
- 3. Explain what is "-" and Ref. in the caption?
- 4. The caption could be a bit more elaborated!

AC: We have modified the table and caption as suggested. The caption now reads: "Table 3: Statistics on distances between point clouds computed on the basis of the M3C2 algorithm, showing mean, standard deviation and root mean square error (RMSE) of each point cloud pair. UAV refers to UAV photogrammetry, TP to terrestrial photogrammetry and TLS to terrestrial laser scanning. Ref. stands for reference and "-" means no comparison was performed."

RC Table 4:

- The text could be all set on left side of the cell
- Caption: Could you explain what is sigma?

AC: We have replaced sigma with '**standard deviation**' and moved text to the left as suggested.

**Minor Comments**

**RC Line 1: are abbreviations authorized in the title?**

AC: There are a number of manuscripts in the same special issue as this article with 'UAV' abbreviated in the title. https://www.nat-hazards-earth-syst-sci.net/special\_issue859.html We therefore assumed this is acceptable. We kindly ask the editor to confirm this.

**RC Line 14: What is a geo-site?**

AC: We meant geosite without hyphen. According to the encyclopedia of geomorphology, geosites are "portions of the geosphere that present a particular importance for the comprehension of Earth history. They are spatially delimited and from a scientific point of view clearly distinguishable from their surroundings. More precisely, geosites are defined as geological or geomorphological objects that have acquired a scientific (e.g. sedimentological stratotype, relict moraine representative of a glacier extension), cultural/ historical (e.g. religious or mystical value), aesthetic (e.g. some mountainous or coastal landscapes) and/or social/economic (e.g. aesthetic landscapes as tourist destinations) value due to human perception or exploitation." (Reynard, 2004, p.440).

Reynard, E., Geosite, in Goudie, A.S. (ed), Encyclopedia of geomorphology, volume 1, 2004, Routledge, London, UK.

RC Line 16: Explain the abbreviation.

AC: We have added 'unmanned aerial vehicle' as suggested.

**RC Line 18: Explain the abbreviation**

AC: We have added 'digital elevation model' as suggested.

RC Line 22: Did you investigate glacier geodetic mass balance? If not, I think this is a very strong (and not founded) assumption. And it would keep it on the natural hazard topic.

AC: we have replaced 'mass balance' with 'thickness changes'.

RC Line 44: I would put here reference.

AC: We have added: 'Azzoni et al. (2017)' as suggested.

RC Lines 44-46: I would remove his sentence because you are talking about glaciers and natural hazards in this section, and that this sentence come a bit out of the blue and covers very general topics (that we generally find at the very beginning of the introduction).

AC: While these topics are general, they demonstrate why glacier volume change is important in connection to glacier hazards, i.e. because it is related to changes in water resources. We have replaced this sentence with "Glacier downwasting causes changes in water resources, with an initial increase in discharge due to enhanced melt followed by a long-term reduction, affecting drinking water supply, irrigation and hydropower production (Kaab et al., 2005b)" to better clarify this, as explained in the answer to your major comment.

RC Line 51: I find the english really heavy here. You could replace it by: such as

AC: We have replaced 'owing to the ability to generate' with 'such as' as suggested.

RC Lines 58-59: And what about aerial photogrammetric surveys? Aerial LIDAR surveys? I think your assumption is wrong.

AC: While the spatial resolution that can be obtained with conventional aerial surveys (higher flying altitude compared to UAVs) is generally lower, we now mention them as well. We have therefore added: "via aerial laser scanner/photogrammetric surveys (Vincent et al., 2010; Janke, 2013)" and the relative entries in the bibliography.

RC Line 70: for the monitoring of glacier or for monitoring glaciers

AC: We have replaced 'for monitoring of glaciers' with 'for monitoring glaciers'

RC Lines 81-83: Crevasses can also be filled with snow and be invisible... I would remove this sentence.

AC: We have removed the sentence as suggested.

RC Line 82: their

AC: we have replaced 'the' with 'their' as suggested.

RC Lines 84-85: I found this not very clear. What about: ...2014. In summer 2016 the glacier was survey with three different techniques allowing for the generation of pt-cloud, DEM and orthomosaic. The aims were: (1) compare the different methods and select the "better" one for monitoring glacier hazards (2)...

AC: We have rephrased the sentence from: "then, through a dedicated field campaign 85 carried out in summer 2016, we compared different platforms and techniques for point cloud, DEM and orthomosaic generation to assess their ability to monitor glacier hazards: UAV photogrammetry, terrestrial photogrammetry and TLS. The aims were: (1) comparing UAV- and terrestrial photogrammetric products acquired in 2016 against the TLS point cloud;" to "in summer 2016, the glacier was surveyed using three different techniques for the generation of point clouds, DEMs and orthophotos. The aims were: (1) to compare the different methods and select the most appropriate ones for monitoring glacier hazards"

RC Lines 89-90: The reader is not ready for this information. The merged pt cloud is only described later! I would remove.

AC: we have removed this part of the sentence as suggested.

RC Line 97: If you don't use little ice age later in the text, there is no need to add an abbreviation to it.

AC: we have removed the abbreviation as suggested.

RC Line 100: such as

AC: we have replaced 'including' with 'such as' as suggested.

RC Lines 102-105: I think the first sentence is grammatically wrong and to my point of view, this information is not needed, as you are not discussing the processes triggering ice collapses later. I would remove.

AC: we have removed the sentence as suggested.

RC Lines 107-108: To my point of view this information is not relevant. I think it's enough to say it's touristic!

AC: we have removed this part of the sentence as suggested.

RC Line 121: ...70%. In our study, sidelap,...

AC: we have modified the text as suggested.

RC Line 122: I would say between 7 and 9am. Knowing the precise time is not relevant.

AC: the sentence now reads: "Flight operations started around 07:30 and ended around 08:30."

RC Line 128: just? what does this word bring? I would remove it, as it almost sounds negative.

AC: we have removed this word as suggested.

RC Line 129: not needed, we are redirected on the map.

AC: we have removed 'lake Rosole, close to Branca Hut' as suggested

RC Lines 132-133: These guys are not the ones that created the SfM algorithm of Agisoft Photoscan. They didn't even wrote the first SfM algorithm. Can you please explain why you cited this paper? Same comment as before. Why this one?

AC: Agisoft Photoscan is closed source, therefore the exact algorithm that is used is unknown. Both articles are cited by Westoby et al. (2012) in their description of the SfM-workflow. We have replaced the references with Westoby et al. (2012), which is more recent.

RC Line 134: The reader do not know what it is and what it is for at that point... I suggest you move the part on the GNSS survey as section 3.1.1 and state what is a GCP there (Ground Control Point).

AC: We have considered this suggestion. However, placing the GNSS subsection at the start would disrupt the chronological flow previously recommended by you and the other reviewer. We have therefore added a paragraph at the start of section 3, including information about GCPs and the workflow used for photogrammetric processing (see e.g. comment at line 158). The paragraph reads:"In this study, we took advantage of a UAV survey performed in 2014 (Fugazza et al., 2015). Then, through a field campaign in 2016, we conducted different surveys using a UAV, terrestrial photogrammetry and TLS. In the 2014 UAV survey, no ground control points (GCPs) were collected, while in 2016 we specifically set up a control network for geo-referencing purposes. Processing of the UAV and terrestrial images was carried out using Agisoft Photoscan version 1.2.4 (www.agisoft.com), implementing a SfM algorithm for image orientation followed by a multi-view dense-matching approach for surface 3D reconstruction (Westoby et al., 2012). In addition, we employed a DEM from an aerial survey of 2007 to calculate glacier thickness changes over a period of 7 to 9 years."

RC Line 138: You used the SfM method. Is Immerzeel using a different one? Or did you followed his workflow in Agisoft Photoscan. If yes, please specify.

AC: we have deleted this sentence as we now explain that Photoscan was used at the start of section 3, as suggested by you, e.g. in your comment at line 158.

RC Line 142: Two UAV surveys...The reader do not know or do not remember that you already talked about them.

AC: we have modified the text as suggested.

RC Lines 147-149: I think this sentence do not give more information than the map. I suggest to remove

AC: we have removed the sentence as suggested.

RC Line 151: To have parallel flights you need two of them no? So why individual? And how can you do parallel flights in zig zag? I suggest rephrasing.

AC: we have replaced 'several individual parallel flights' with 'several flights'.

RC Line 158: 2014 as well... so maybe you can say somewhere that all UAV data were processed with the same software and version?

AC: We have added a paragraph at the start of section 3.1 including this information, see comment at line 134, and removed it from this sentence.

RC Line 160: the GCP cannot have a rmse. The error of their positioning maybe. Please correct

AC: we have modified the text as suggested.

RC Line 167: I would put the reference after "subvertical surfaces" because otherwise the reader expects to see a UAV and a camera looking downwards

AC: we have moved the figure reference as suggested.

RC Line 170: Can you put the location of the pictures on the map maybe?

AC: As there are 134 pictures, placing them on the map would make it less clear. We therefore preferred to keep a separate figure showing the picture location.

RC Line 174: that's the third time you mention this software. See above comment, I would mention it only once.

AC: We have added a paragraph at the start of section 3.1 including this information, see comment at line 134, and removed it from this sentence.

RC Line 183: For me these are results already. You show only the reconstruction of the terrestrial survey because it's the best I guess. However, Figure 6 already shows the comparison. So I do not understand what this Figure brings to the reader. Consider removing.

AC: We have moved this paragraph to the results section where we now also show the full spatial coverage of UAV and terrestrial photogrammetry and have removed this figure accordingly.

RC Line 186: I think the location alreadygives an idea of the angle of measurement. I would remove also because it does not sound right as the adverb is at the wrong place.

AC: we have removed the word 'frontally' as suggested.

RC Line 195: The glacier is not a room ;-) "at the glacier vicinity" or "outside the glacier extent" or "on the periglacial area"

AC: we have replaced 'outside the glacier' with 'on the periglacial area'.

RC Line 196: Define GNSS

AC: we have added global navigation satellite system between parentheses.

RC Line 196: The location of the data? what is this? Can you rephrase?

AC: we have replaced 'at their location' with 'at the target location'.

RC Lines 197-199: I don't understand your sentence. Please rephrase.

AC: we have rephrased this sentence as: "GCPs were used 1) to geo-reference UAV data directly, by identifying the targets on the images in Photoscan; 2) to register theodolite measurements for georeferencing terrestrial photogrammetry and TLS."

RC Line 199: ... consisted of a square white fabric (80x80cm),... ?

AC: We have replaced 'consisted in a piece of white fabric 80x80 cm wide' with 'consisted in a square piece of white fabric (80 x 80 cm)'

RC Line 205: precise point beside the Branca Hut, with coordinates...

AC: we have modified the text as suggested.

RC Lines 208-2012: I would shorten such as: "but due to the local topography preventing the radio link and mobile phone services (for RTK), fixed points with measurement time of approx. 12 min were surveyed."

AC: only a few points were measured in static mode. We have rephrased the sentence as: "but due to the local topography preventing the radio link and the lack of mobile phone services (for RTK), some points were measured in static mode with measurement time of approximately 12 minutes"

RC Lines 212-213: I think there should not be theories on accuracy here ;-) The device or the post-processing software should give you a pretty good approximation of your points error. Can you find them? AC: sometimes terms related to accuracy are not used in a common way by scientists coming from different fields. In Geodesy, the term "theoretical accuracy" refers to the estimated accuracy of estimated values obtained on the basis of observed data processing. Typically, it is the case of estimated accuracy contained in a covariance matrix output after least squares adjustment. This value is obtained from variance-covariance propagation, but also it takes into considerations the quality of adopted observations, as can be found in books about Least Squares. In such a case, what we termed as "theoretical accuracy" does not come from theory, but it is just the estimated value output by the RTK-GNSS processing software.

Thus this is exactly what the Reviewer would like to see. We slightly modified this sentence to make this point clearer and have replace the word 'accuracy' with 'uncertainty' as we preferred to avoid using the term 'accuracy' as indicated in the previous author's response:

**"The theoretical uncertainty of GCPs provided by the processing code was in the order of 2-3 cm."**

RC Line 215: by the

AC: we have added 'the' as suggested.

RC Line 2016: space

AC: we have added a space as suggested

RC Lines 218-224: I would shorten the whole as: for instance: "...2x2 m, with a +/-3m accuracy. We converted the DEM from the "Gause Boaga" to the "ITRS2000" datum and the heigth from ellipsoidal to geodetic using the official software for datum transformation in Italy (Verto ver.3)

AC: we have rephrased the sentence as suggested.

RC Lines 224-225: The 7 parameter transformation is done in the software is it? If yes this information is a bit too detailed I think.

AC: we have deleted this sentence as suggested.

RC Line 226: In the introduction you state that "possibility of data fusion needs to be investigated", then there is a methodology section which is called "Merging UAV and close-range photogrammetric point clouds" that comes a bit out of the blue (why not a combination of other methods and only these two?)

AC: The reason why we chose to compare the two photogrammetric techniques is because they are less costly than TLS. We have added this explanation at the start of the paragraph, in a sentence that reads: "We chose to avoid TLS and employed the two lower cost techniques (Chandler and Buckley, 2016) to assess their potential for combined future use."

RC Line 232: as well as

AC: we have replaced 'and' with 'as well as' as suggested.

RC Line 233: The point density is controlled by the obtainable spatial resolution? What does it mean? Please rephrase.

**AC: We have replaced "and the obtainable spatial resolution" with "and determines spatial resolution"**

RC Line 233: I think you can use "the former, the later" only when you listed sth before. You only talked about point density, so I would use "it" or re-write "point density" instead of "the latter property".

AC: we have replaced 'the latter property' with 'point density' as suggested.

RC Lines 233-234: I think not only. If your images are taken on a surface with little contrast, the dense matching tool will not be able to do anything. Please rephrase the whole sentence.

AC: We have rephrased this sentence as: "point density is affected by image texture, sharpness and resolution, which affect the performance of dense matching algorithms (Dall'Asta et al., 2015)"

RC Line 253: Finally can be used if you use firstly, secondly, ... it's not the case and surprises the reader that ask himself if he has not missed sth. Please change.

AC: We have rephrased the sentence as: "Within the same sample locations, we compared the point clouds in a pairwise manner."

RC Lines 255-256: But TLS is influences by atmospheric condition and angle of survey. So it also has errors. If there is no other criteria of why you chose TLS I would not say anything else. Just say "TLS point cloud was used as reference". You say in the abstract that UAV outperforms TLS, so why not using UAV as reference then? To me it is contradictory

AC: We have modified the manuscript as suggested. TLS is generally regarded as more accurate compared to UAV photogrammetry (see also your suggested reference Naumann et al., 2013), while the main findings of our work are that UAV is superior to it in terms of coverage, logistics and cost and thus should be preferred in glacial environments unless obtaining absolute accuracy is paramount.

**RC Line 261: Which does what?**

AC: we have extended and better explained this part since Reviewer 2 asked to provide more details. We have therefore added the following text: **"As discussed in Fey and Wichmann (2016), the distance between a pair of point clouds is often evaluated by comparing elevations at corresponding nodes of DEMs, after resampling of the original data. This approach works properly when both point clouds are**  approximately aligned along the same planar direction, but not when there are structures with different alignments as in the case of the glacier surfaces under investigation. In fact, the M3C2 algorithm does not always evaluate the distance between two point clouds along the same directional axis, but computes a set of local normals using points within a radius *D* depending on the local roughness, which is directly estimated from the point cloud data, and also considering the uncertainty of preliminary local registration refinement using ICP. In this case, a radius *D*=20 cm and a pre-registration uncertainty of 5 cm were considered, the latter obtained from ICP residuals"

RC Line 262: this is English slang :-) A synonym would be better

AC: We have replaced 'get rid of' with 'remove' as suggested.

RC Lines 262-263: what could then be focused? This solution? the registration errors from the analysis? I do not understand the sentence. Please rewrite.

AC: we have replaced 'which could then be focused' with 'and focus'.

RC Line 265: UAV is also a close-range remote sensing technique. Maybe change to UAV and terrestrial?

AC: we have replaced 'close-range' with 'terrestrial' as suggested.

RC Line 267: please define (UAV and terrestrial) for the reader (that probably do not remember what you used in 2016).

AC: We have added '(UAV and terrestrial)'.

RC Line 268: You use ICP a lot in your text afterward and use it for registration. So maybe it would be useful at a point that you explain what ICP means and what it does?

AC: We have spelled out the ICP acronym by adding '(iterative closest point)' between parentheses and added a description of the algorithm, which reads: "ICP iteratively matches a source point cloud to a reference point cloud in Euclidean space and calculates the necessary rotation and translation to align the source point cloud to the reference based on minimization of a distance metric (usually point-to-point)."

RC Line 272: They were not in the same reference system before? I think what you want to say is: "After many iterations, both point clouds were aligned based on the best solution found by the ICP". Solution is not the right word but something alike.

AC: Thanks for the suggestion. It appears that 'solution' is a widely used term to describe the optimal alignment (e.g. Low, 2004), and thus we have rephrased this sentence from: "After this task, both original point clouds resulted aligned into the same reference system." to "After many iterations, both point clouds were aligned based on the best solution found by the ICP" as suggested.

Low, K-L: Linear Least-Squares Optimization for Point-to-Plane ICP Surface Registration, Technical Report TR04-004, Department of Computer Science, University of North Carolina at Chapel Hill, February 2004, available from:

http://citeseerx.ist.psu.edu/viewdoc/download?doi=10.1.1.298.4533&rep=rep1&type=pdf

RC Line 273: Same comment as before: Better find a synonym like "remove" or "delete"

AC: we have replaced 'get rid of' with 'remove' as suggested.

RC Line 275: this merged data set?

AC: we have added 'merged' as suggested.

RC Lines 275-276: I think the concept of up-to-date changes every day. What does it mean? I suggest either remove the sentence or put the model of your computer there.

AC: we have removed this part of the sentence.

RC Line 279: I suggest to replace: and therefore

AC: we have replaced 'and aid in the interpretation of glacier hazards' with 'and therefore interpret glacier hazards'.

RC Line 282: only sounds negativ

AC: we have deleted 'only' as suggested.

RC Line 283: See my comment above: change with "terrestrial"

AC: we have replaced 'close-range' with 'terrestrial' as suggested.

RC Line 284: You could shorten the section here as: " The investigation of glacier hazards was conducted using the point cloud and orthophoto from the 2014 UAV dataset as well as the merged (UAV and terrestrial) point cloud and orthophoto from 2016).

AC: We have rephrased the sentence as suggested.

RC Line 284: I would insert here "using visual inspection", so that you don't need the next sentence

AC: We have rephrased the sentence from: 'In this study, we focused on ring faults and normal faults, which were manually delineated by using geometric properties from the point clouds' to 'In this study, we focused on ring faults and normal faults, which were identified by visually inspecting their geometric properties in the point clouds and manually delineated'.

RC Lines 286-287: delete the sentence

AC: we have deleted this sentence as suggested.

RC Lines 287-288: I don't understand. Please check if this is relevant and if yes rephrase.

AC: We have rephrased from: "On orthophotos, both types of structures also generally appear as linear features in contrast with their surroundings" to: "On orthophotos, both types of structures generally appear as dark linear features owing to shadows projected by fault scarps"

RC Lines 288-292: This second part of the paragraph has more its place in the results section for me, because you already describe the form and the location of the structures. I would move it.

AC: We have considered this suggestion. However, location and orientation are criteria used to discriminate normal faults and ring faults from crevasses, which were not included in this study, therefore we have decided to keep the sentence in the method section.

RC Line 321: over a common glacier area

AC: we have replaced 'over a reference area common to all three DEMs' with 'over a common glacier area' as suggested.

RC Line 324: I don't understand. Can you please rewrite?

AC: we have rephrased as: "the truncation error caused by the use of a discrete sum (sum of DEM difference at each pixel multiplied by pixel area) in place of the integral in volume calculation (Jokinen and Geist, 2010)."

RC Line 325: It is not clear what kind of other information than the standard deviation can this

AC: according to Jokinen and Geist (2010), the volume between two surfaces at times  $t_1$  and  $t_2$  can be expressed as:

$$\Delta V = \int_{\Omega} \int (z(x, y, t_2) - z(x, y, t_1)) \, dx \, dy$$

Where  $(x, y) \in \Omega \subset R^2$

while in the case of DEMs, this formula is approximated as:

$$\Delta V = \sum_{k=1}^{K} \bar{d}_k A_k$$

Where  $\bar{d}_k$  is the average of differences  $\bar{d}(x_i, y_i) = z(x_i, y_i, t_2) - z(x_i, y_i, t_1)$  at the vertices of  $\Omega_k$  and  $A_k$  is the area of  $\Omega_k$

The uncertainty in volume change calculated from DEM differencing can then be expressed as a combination of two factors: 1) errors in elevation propagating to the elevation difference

and volume calculation and 2) truncation error, as the integral in the first equation is replaced by a finite sum in the second equation.

We have revised our calculations, using the approach by Rolstad et al. (2009) to calculate factor 1).

This approach takes into account spatial autocorrelation of elevation differences over bedrock. Thus, the standard deviation of DoD over bedrock  $\sigma_{\Delta h}$  is scaled to account for the effective correlated area,  $A_{cor}$ .

 $A_{cor}$  is calculated as  $A_{cor} = \pi \times L^2$ , where *L* is the radius of a circular area, and is equal to the correlation length.

We identified this correlation length by looking at the semivariograms of the DoDs in R software and found a mean value of 50 m for the three DoDs.

The standard deviation  $\sigma_{cor}$  is then calculated as  $\sigma_{cor} = \sqrt{\sigma_{\Delta h}^2 \frac{A_{cor}}{5 \cdot A}}$ , where *A* is the glacier

area.

Finally, the uncertainty on volume change is expressed as  $\sigma_{\Delta V} = \sigma_{cor} \times A$ , considering the error as entirely correlated.

As regards the truncation error  $E_T(\Delta V)$ , we calculated it following the approach by Jokinen and Geist (2010), i.e. using the formula

$$|E_T(\Delta V)| \leq \frac{h^4}{12} \sum_{k=1}^K \max(x, y) \in \Omega_k \left| \frac{\partial^2}{\partial x^2} d(x, y) + \frac{\partial^2}{\partial y^2} d(x, y) \right|$$

Where *h* is grid spacing and  $\frac{\partial^2}{\partial x^2} d(x, y) + \frac{\partial^2}{\partial y^2} d(x, y)$  is the Laplacian operator of d(x, y)

RC Line 330: who says that this is enough? You or did you find a publication that found out the minimum points needed to get a certain accuracy? Can you please specify?

AC: from a rigorous point of view, this is simply an application of the "sampling theory." If you have a planar surface, three points would be enough to estimate the fitting plane. If you have a surface with a more complex shape, you need more points. An exact correlation between point sampling and surface approximation accuracy can be obtained by applying a Fourier analysis. Here we followed a simplified approach: considering that a minimum point density of approximately 100 points/m2 was found (i.e., one point every one square decimeter on the glacier surface), we retained that the errors in the reconstructed surface were lower than the local surface roughness and noise.

RC Line 332: It it clear from Fig. 6 that the terrestrial photogrammetry is the method that produces the best results in term of point density and UAV the "worse". I did not understand how much in brings you to merge the point clouds from both techniques if the UAV point cloud is so sparse! What difference did it make?

AC: as explained in the answer to your major comment, and now reported in the results section, the two point clouds cover different areas. We have added a figure in the results section showing this and added a paragraph explaining the different coverage of the techniques.

RC Line 349: larger deviations

AC: We have replaced 'worse values' with 'greater deviations'.

RC Line 357 and Line 361: Fig.7

AC: we have written 'Fig.7' as suggested

RC Line 375: On Fig.8a, we can see loss between 0 to -9m. Where are these -30 meters taking place?

AC: Fig. 8a reports thickness change rates instead of thickness changes to allow a comparison between the three DEM pairs. We have added the change rate between parentheses here and throughout the paragraph.

RC Line 377: Same as before, I don't see these numbers on the map in Fig.1

AC: see the answer to your comment above.

RC Line 382: 20 to 26m?

AC: we have replaced '20/26 m' with '20 to 26 m' as suggested.

RC Line 384: -2 to -5m

AC: we have replaced '-2/-5 m' with '-2 to -5 m' as suggested.

RC Line 392: You write about geodetic mass balance in the abstract and in the discussion part, but none of your results deals with glacier mass balance. This is very surprising for the reader that is wondering what's going on. Maybe consider removing it.

AC: we have replaced 'geodetic mass balance' with 'thickness changes'.

RC Line 404: I think you should write this as suggestion, as you did not tested it.

AC: we have replaced 'is due' with 'might be due'.

RC Line 416: That's right. The literature has already shown that UAV are better on flat surfaces and terrestrial photogrammetry and TLS on steeper topography. So what do your results bring more to this knowledge?

AC: as discussed in the answer to your major comment, these publications are rather recent and mainly describe surveys of non-glacial environments. We have kept the references to these publications to provide a review of their conclusions in a different environment, and restructured the discussion section adding a further paragraph discussing why our approach should be preferred to TLS and a comparison between our techniques, ALS and aerial photogrammetry. RC Lines 450-451: You say it's adequate. What does it mean? When looking at Table 1, at the elevation differences between the DSMs after co-registration, I see that the standard deviation of the elevation differences between both UAV DSMs is of 2.2 m. Considering that you have a melt of several meters (I'm guessing 6m per year, so 12 in two years), I'm not sure you can say that this technique is adequate. Can you please explain in a more quantitative way what "adequate" means to you?

AC: As explained in the answer to your major comment, the ice thickness change uncertainty results in 2.60% of the volume change uncertainty for the 2014-2016 comparison. We have replaced this sentence with: **"UAV photogrammetric products permitted to investigate ice volume changes over 2 years with an uncertainty of 2.60%**"

RC Line 459: this sounds negative. I would remove

AC: we have removed 'only' as suggested.

RC Line 459: You investigated different techniques to map/monitor glacier hazards.

AC: we have rephrased as: "different techniques to map/monitor hazards related to the glacier collapse"

RC Line 465: what about TLS?

AC: We integrated terrestrial and UAV photogrammetry, while TLS was used for comparison only. We now clarify that UAV and close-range photogrammetry should be preferred to TLS, and have rephrased the sentence as: "While our integrated approach using a multicopter and terrestrial photogrammetry should be preferred to TLS for the investigation of small individual ice bodies"

RC Line 470: Aerial LiDAR surveys? Did you have some for the Stelvio National Park? Why is this technique suddenly comming up here?

AC: We now mention ALS and aerial photogrammetry in the introduction and discussion to provide a more comprehensive comparison of different techniques and their advantages/drawbacks. We have replaced 'aerial LiDAR surveys' with 'higher altitude aerial surveys' to include both ALS and aerial photogrammetry and clarify we are not referring to UAVs.

RC Lines 482-483: I don't think the UAVs in the mountains are more flexible in terms of meteorological conditions...

AC: we have removed this part of the sentence

RC Line 487: You did not try to measure mass balance, so please remove.

AC: we have removed this part of the sentence.

RC Line 493: Your maps were produced with ortophotos right?

AC: The base layers of the maps in Figure 7 are indeed orthophotos but the analysis was mainly conducted based on the point clouds. We have added '**and orthophotos**' after 'point clouds'.

RC Line 499: This is also new... You never talked about this before. Why in the conclusion?

AC: we have deleted this part of the sentence.

RC Line 504: delete 'mass balance'.

AC: we have replaced 'mass balance' with 'thickness changes'.

We have prepared a point by point response to the reviewer's comments. In the following text, reviewer's comments are reported as RC and highlighted in italics, our answers as AC in plain text while our changes to the text are in bold black.

**Review**

The authors have benefited from two sets of very thorough reviews on their original manuscript. I have read these reviews and the authors' responses, which are mostly appropriate and well-justified. Importantly, the authors have shortened many sections and undertaken some restructuring which has improved the flow and clarity of the manuscript.

I have provided a few additional points for consideration prior to publication. These are minor and should not take the authors very long to address.

Dear Reviewer,

Thank you for your comments. We have provided our answers to your points below.

RC Line 50 – 'sensing', not 'Sensing'

AC: the word is now in lower case as suggested.

RC Line 63 and throughout – 'photogrammetry', not 'Photogrammetry'

AC: 'photogrammetry' is now lower case throughout the manuscript.

RC Line 69 – remove 'slowly'

AC: We have removed this word as suggested. This part of the sentence has now been changed as: "**is on the rise**"

RC Line 74 – would be useful to quantify 'rapidly downwasting' here – approximately how much surface lowering/terminus retreat is there per year?

AC: We have added "(almost 5 ma-1 water equivalent, Senese et al., 2012)" to clarify the amount of surface lowering per year.

RC Line 102 – is it appropriate in NHESS for authors to list articles that have been 'submitted'? 'In press' or 'Accepted' – yes; 'submitted' – I'd perhaps not be comfortable with including this. This requires an Editorial decision.

AC: The article that was listed as 'submitted' has now been published on the Journal of Maps.

We have replaced '**submitted**' with '**2017**' and changed the entry in the reference list accordingly.

RC Line 160 – I assume that no independent check data were acquired to truly test the 'accuracy' of the UAV-SfM data during the model generation stage? If so, the wording needs changing here – the RMSE for the PhotoScan 'markers' in this situation is simply a reflection

of the internal project consistency (i.e. how well the software can shift/rotate/transform the data to fit the user-placed markers). It provides no true measure of accuracy, for which an independent set of check points is required. I would strongly suggest authors clarify this briefly, or change their terminology. I note that there has already been some discussion about the appropriateness of the term 'accuracy' elsewhere in the manuscript (see p24 of the author response), and request that the authors consider this a little further in this paragraph.

AC: Thanks for your comment. We have changed the terminology here, replacing "which can be used as an indicator of accuracy for the geo-referencing of the photogrammetric block." with "which can be used as an indicator of the internal consistency of the photogrammetric block", and deleted a similar comment in section 3.2 Terrestrial Photogrammetry, i.e. "which can be considered as the accuracy of absolute geo-referencing". We have further decided to avoid using the term accuracy in the manuscript, as the accuracy depends on the use of the data, as explained in the previous author response. Thus, we have replaced "The final accuracy of our UAV photogrammetric products was nevertheless adequate to investigate ice thickness changes over 2 years" with "UAV photogrammetric products permitted to investigate ice volume changes over 2 years with an uncertainty of 2.60%" in the discussion section and replaced the term accuracy with uncertainty throughout the manuscript, providing uncertainty values where available.

RC Line 261 – these sentences are a vast oversimplification of the M3C2 algorithm, which is complex and requires some more explanation – i.e. what was the radius for surface normal estimation, what was the value of the registration that was specified prior to analysis? A sentence explaining why you chose this method over others (e.g. 2.5D raster subtraction) would be useful – e.g. was the topography complex enough to require its use?

AC: Although keeping the discussion short, we have better explained the motivations for using M3C2 algorithm. Also we briefly reviewed the main features of this algorithm, and reported the values of the adopted parameters. The new text is reported below:

"As discussed in Fey and Wichmann (2016), the distance between a pair of point clouds is often evaluated by comparing elevations at corresponding nodes of DEMs, after resampling of the original data. This approach works properly when both point clouds are approximately aligned along the same planar direction, but not when there are structures with different alignments as in the case of the glacier surfaces under investigation. In fact, the M3C2 algorithm does not always evaluate the distance between two point clouds along the same directional axis, but computes a set of local normals using points within a radius *D* depending on the local roughness, which is directly estimated from the point cloud data, and also considering the uncertainty of preliminary local registration refinement using ICP. In this case, a radius *D*=20 cm and a pre-registration uncertainty of 5 cm were considered, the latter obtained from ICP residuals."

RC Line 371: 'thickness', not 'Thickness'

AC: the word is now lower case as suggested.

RC Table 1 – column 1 - please add some annotation to help the reader understand the source of each DEM – e.g. (TLS) / (UAV-SfM) etc

AC: Thanks for the suggestion. The information about the source of each DEM is now written in the caption, as suggested by the other reviewer.

**RC Figure 4 – requires a distance scale on both or either of the panels**

AC: We have deleted this figure as suggested by reviewer 1, and replaced it with a figure showing the full spatial coverage of UAV and terrestrial photogrammetric point clouds to better clarify the differences between the two and the advantages of merging the point clouds.

RC Figure 6 – I take issue with the different scales on each of these panels – it is impossible to compare like-for-like. Strongly suggest authors modify this so that the point densities between panels are directly comparable.

AC: We have produced a new figure with uniform color scales so point densities between panels are directly comparable. However, terrestrial photogrammetry has a much higher point density than the other techniques, whose panels now mostly show one single colour. This makes it impossible to understand how different features are represented by individual techniques (e.g. vertical cliffs vs horizontal features in UAV photogrammetry). We have provided the new figure but kindly ask you and the editor to choose which version is best suited for point density analysis.

**Combination of UAV and terrestrial photogrammetry to assess rapid glacier evolution and map glacier hazards**

- Fugazza, Davide1; Scaioni, Marco2; Corti, Manuel2; D'Agata, Carlo3; Azzoni, Roberto Sergio3;
   Cernuschi, Massimo4; Smiraglia, Claudio1; Diolaiuti, Guglielmina Adele3
- 5 1Department of Earth Sciences 'A.Desio', Università degli studi di Milano, 20133 Milano Italy
- 2Department of Architecture, Built Environment and Construction Engineering, Politecnico di Milano, 20133 Milano
   Italy
- 8 3Department of Environmental science and policy (DESP), Università degli studi di Milano, 20133 Milano Italy
- 9 4Agricola 2000 S.C.P.A., 20067 Tribiano (MI) Italy
- 10

12

11 Correspondence to: Marco Scaioni (marco.scaioni@polimi.it)

**Abstract**

[revised manuscript text omitted]

We conducted our first UAV survey of the glacier in 2014; then, through a dedicated field campaign 97 98 earried out in summer 2016, we compared the glacier was surveyed using three different platforms 99 and techniques for point eloud, DEM and orthomosaie the generation to assess their ability to 100 monitor glacier hazards: UAV photogrammetry, terrestrial photogrammetry and TLS.of point 101 clouds, DEMs and orthophotos. The aims were: (1) comparing UAV- and terrestrial 102 photogrammetric products acquired in 2016 against the TLS point cloud; to compare the different 103 methods and select the most appropriate ones for monitoring glacier hazards (2) identifyingto 104 identify glacier-related hazards and their evolution between 2014-2016 using the merged point eloud from UAV and terrestrial photogrammetry and UAV orthophotos; and 3) investigating to 105 106 investigate changes in ice thickness changes 
[revised manuscript text omitted]
 GCPs were used 1) to geo-referencing of reference UAV 229 230 data the GCPs were directly visible, by identifying the targets on the quadcopter images, for in 231 Photoscan; 2) to register theodolite measurements for geo-referencing terrestrial photogrammetry 232 and TLS-they were adopted for the registration of theodolite measurements... The targets consisted 233 in a square piece of white fabric (80 x 80 cm-wide,), with a circular marker in red paint chosen to provide contrast against the background. Except for the one GCP located at the highest site, such 234 GCPs were positioned on large, flat boulders to provide a stable support and reduce the impact of 235 236 ice ablation between flights.-

237 GNSS data were acquired by means of a pair of Leica Geosystems 1200 geodetic receivers working in RTK (Real-Time Kinematics) mode (see Hoffman-Wellenhof, 2008). One of them was set up as 238 239 master on a boulderprecise point beside Branca Hut, where a monument had been establishedhut, 240 with known coordinates in the mapping reference system ITRS2000 / UTM 32N. The second receiver was used as a rover, communicating via radio link with the master station. The maximum 241 242 distance between master and rover was less than 1.5 km, but due to the local topography prevented 243 broadcastingpreventing the differential corrections in a few zones of radio link and the glacier. 244 Unfortunately, no lack of mobile phone services were available and consequently the internet 245 network could not be accessed, precluding the use of the regional GNSS real-time positioning service. Non-(for RTK), some points were processed measured in fast-static mode, requiring a 246

247 longer\_with measurement time of approx.approximately 12 minutes. The theoretical
248 accuracyuncertainty of GCPs provided by the processing code was estimated inon the order of 2-3
249 cm.

**250 **3.5 2007 DEM**

[revised manuscript text omitted]

341 it helped to provide a realistic visualization of the topographic model, which can aid the
 342 interpretation of and therefore to interpret glacier hazards.

**343 4.2 Glacier hazard mapping**

344 The investigation of glacier hazards was conducted by considering datasets from 2014 and 2016. In 2014, only using the point cloud and UAV orthophoto were available, while in 2016 from the 2014 345 346 UAV dataset as well as the merged (UAV and terrestrial) point cloud obtained by merging UAV and elose-range photogrammetric data sets was used in combination with the UAV and orthophoto from 347 2016. In this study, we focused on ring faults and normal faults, which were manually delineated by 348 349 using identified by visually inspecting their geometric properties from in the point clouds and 350 manually delineated, while colorcolour information from orthophotos was used as a cross-check. 351 On point clouds, mapping is based on visual inspection of vertical displacements following faulting 352 or subsidence. On orthophotos, both types of structures also generally appear as dark linear features in contrast with their surroundingsowing to shadows projected by fault scarps. As these structures 353 354 may look similar to crevasses, further information concerning their orientation and location needs to 355 be assessed for discrimination. The orientation of fault structures is not coherent with glacier flow, 356 with ring faults also appearing in circular patterns. Their location is limited to the glacier margins, 357 medial moraines and terminus, whereas crevasses can appear anywhere on the glacier surface 358 (Azzoni et al., submitted2017). After delineation, we also analysed the height of vertical facies 359 using information from the point clouds.-

**360 **4.3 DEM eoCo**-registration for glacier thickness change estimation**

[revised manuscript text omitted]

- 856 Spetsakis, M. and Aloimonos, J.Y.: A multi-frame approach to visual motion perception,
  857 International Journal of Computer Vision, 6, 245-255, 1991, doi: 10.1007/BF00115698.
- 858 | Teunissen, P.J.G.: Testing theory. An introduction. Series on Mathematical Geodesy and859 Positioning, VSSD Delft, The Netherlands, 2009.

Urbini, S., Zirizzotti, A., Baskaradas, J.A., Tabacco, I.E., Cafarella, L., Senese, A., Smiraglia, C.,
Diolaiuti, G.: Airborne radio echo sounding (RES) measures on alpine glaciers to evaluate ice
thickness and bedrock geometry: Preliminary results from pilot tests performed in the ortlescevedale group (Italian alps), Annals of Geophysics, 60, G0226, doi: 10.4401/ag-7122, 2017.

- 864 Vincent, C., Auclair, S. and Le Meur, E.: Outburst flood hazard for glacier-dammed Lac de
  865 Rochemelon, France, Journal of Glaciology, 56, 91-100, 2010, doi: 10.3189/002214310791190857
- 866 Vincent, C., Thibert, E., Harter, M., Soruco, A. and Gilbert, A.: Volume and frequency of ice
  avalanches from Taconnaz hanging glacier, French Alps, Annals of Glaciology, 56, 17-25, 2015,
  doi: 10.3189/2015AoG70A017.
- Westoby, M.J., Brasington, J., Glasser, N.F., Hambrey, M.J. and Reynolds, J.M.: Structure-fromMotion' photogrammetry: A low-cost, effective tool for geoscience applications, Geomorphology,
  179, 300-314, 2012, doi: 10.1016/j.geomorph.2012.08.021.
- Winkler, M., Pfeffer, W.T. and Hanke, K.: Kilimanjaro ice cliff monitoring with close range
  photogrammetry, International Archives of the Photogrammetry, Remote Sensing and Spatial
  Information Sciences, XXII ISPRS Congress, Melbourne, Australia, 25 August-1 September 2012,
  39-B5, 441-446, 2012.
- 876

**Tables**
* * *
| DEM pair   | Elevation
differences without                                    | Co-regist | ration shifts | Elevation
differences with co-                                |  |
|------------|---------------------------------------------------------------------|-----------|---------------|------------------------------------------------------------------|--|
|            | co-registration shifts $(\mu_{\Delta H} \pm \sigma_{\Delta H})$ [m] | X [m]     | Y [m]         | registration shifts $(\mu_{\Delta H} \pm \sigma_{\Delta H})$ [m] |  |
| 2007-2014_ | 1.96±2.60                                                           | 1.11      | -1.11         | 0.00±1.70                                                        |  |
| 2007-2016  | -0.43±3.48                                                          | 2.44      | -1.11         | $0.00{\pm}2.60$                                                  |  |
| 2014-2016  | -2.92±3.21                                                          | -0.20     | -1.30         | 0.00±2.22                                                        |  |

*Table 1: Statistics of the elevation differences between DEM pairs before and after the*

*application of co-registration shifts.* DEM 2007 from aerial multispectral survey, DEM 2014 and

*DEM 2016 from UAV photogrammetry.*

| S
a
m
p
l
e
W
i
n
d
o
w | Area
(m 2 ) | number Number of points in sample windows |                                                                                         | Mean and standard deviation of
point density [points/m 2 ] |                                              |                                                                           | Number of <del>point</del> points
above the lower 12.5%
percentile |                            |                                                      |     |
|----------------------------------------------------------|---------------------------|--------------------------------------------------|-----------------------------------------------------------------------------------------|--------------------------------------------------------------------------|----------------------------------------------|---------------------------------------------------------------------------|---------------------------------------------------------------------------------|----------------------------|------------------------------------------------------|-----|
| -                                                        |                           | UAV
<del>photogra</del>
<del>mm.</del>     | <del>Terrestri</del>
<del>al</del>
<del>Photogra</del>
<del>mm.TP</del> | TLS                                                                      | UAV
<del>Photogra</del>
<del>mm.</del> | <del>Terrestrial</del>
<del>Photogram</del>
<del>m.TP.</del> | TLS                                                                             | UAV
Phot
ogra
mm. | Terr
estri
al
Pho
togr
am
m.T
P | TLS |
| 1                                                        | 2793                      | -                                                | 1984k                                                                                   | 141k                                                                     | -                                            | 1654±637                                                                  | 226±100                                                                         | -                          | 880                                                  | 26  |
| 2                                                        | 1806                      | 76k                                              | 2175k                                                                                   | 130k                                                                     | 109±29                                       | 2297±708                                                                  | 391±217                                                                         | 61                         | 881                                                  | 0   |
| 3                                                        | 495                       | 43k                                              | 712k                                                                                    | 25k                                                                      | 103±27                                       | 1978±606                                                                  | 151±60                                                                          | 49                         | 766                                                  | 31  |
| 4                                                        | 672                       | 62k                                              | 557k                                                                                    | 33k                                                                      | 108±22                                       | 1384±530                                                                  | 141±69                                                                          | 62                         | 324                                                  | 2   |
| 5                                                        | 3960                      | 406k                                             | 810k                                                                                    | -                                                                        | 141±22                                       | 485±227                                                                   | -                                                                               | 97                         | 31                                                   | -   |

*Table 2: Area and number of points in each sample window on the Forni Glacier terminus, mean*886 *and standard deviation of local point density and number of points above the lower 12.5%*887 *percentile in each window. k stands for thousands of points.* UAV refers to UAV photogrammetry.
888 *TP to terrestrial photogrammetry and TLS to terrestrial laser scanning.*

| S am       |           | Means and Std. Dev.s of M3C2 distances
[cm] |                                              |                                              | RMSE of M3C2 distances [cm]                                              |                                              |                                                                          |
|-------------------|-----------|------------------------------------------------|----------------------------------------------|----------------------------------------------|--------------------------------------------------------------------------|----------------------------------------------|--------------------------------------------------------------------------|
| sam
ple
Win | Ref.      | TLS                                            | TLS                                          | UAV
<del>Photogramm</del>
<del>.</del> | TLS                                                                      | TLS                                          | UAV
<del>Photogram</del>
<del>m.</del>                             |
| -                 | Slav
e | Terrestrial
Photogramm
- TP       | UAV
<del>Photogramm</del>
<del>.</del> | Terrestrial
Photogramm
- TP     | <del>Terrestrial</del>
<del>Photogram</del>
<del>m.TP</del> | UAV
<del>Photogramm</del>
<del>.</del> | <del>Terrestrial</del>
<del>Photogram</del>
<del>m.TP</del> |
| 1                 |           | 4.5±7.4                                        | -                                            | -                                            | 8.7                                                                      | -                                            | -                                                                        |
| 2                 |           | -1.1±10.5                                      | 14.8±34.7                                    | -14.5±26.7                                   | 10.6                                                                     | 37.7                                         | 30.4                                                                     |
| 3                 |           | 8.4±4.1                                        | 14.7±15.1                                    | -8.5±18.9                                    | 9.4                                                                      | 21.1                                         | 20.7                                                                     |
| 4                 |           | 2.8±5.3                                        | 9.4±22.2                                     | -2.3±24.9                                    | 6.0                                                                      | 24.0                                         | 25.0                                                                     |
| 5                 |           | -                                              | -                                            | -8.5±25.3                                    | -                                                                        | -                                            | 26.7                                                                     |

Table 3: Statistics on distances between point

- *clouds computed on the basis of the M3C2 algorithm, showing mean, standard deviation and root*
- *mean square error (RMSE) of each point cloud pair. UAV refers to UAV photogrammetry, TP to*
- *terrestrial photogrammetry and TLS to terrestrial laser scanning. Ref. stands for reference and "-"*
- *means no comparison was performed.*

| DEM pair  | Mean thickness change [m] | Mean thinning rates
[ma -1 ] | Volume Change [10 6
m 3 ] |
|-----------|---------------------------|--------------------------------------------|----------------------------------------------------|
| 2007-2014 | $-31.91 \pm 1.70$         | $-4.55 \pm 0.24$                           | $-10.00 \pm 0.1217$
(1.74%)                     |
| 2007-2016 | $-42.86 \pm 2.60$         | $-4.76 \pm 0.29$                           | $-13.46 \pm 0.1420$
(1.47%)                     |
| 2014-2016 | $-10.41 \pm 2.22$         | $-5.20 \pm 1.11$                           | $-3.29 \pm 0.0508$
(2.60%)                      |

899 Table 4: Average ice thickness change, thinning rates and volume loss from DEM differencing
900 over a common reference area of 0.32 km2 for all DEM pairs. Uncertainty of thickness change
901 expressed as +one standard deviation of residual elevation differences over stable areas after
902 DEM co-registration.

---

## Author Response (AR3)

Dear Editor,

We thank you and the reviewer for your comments. We have modified the manuscript as suggested. Specifically, we have changed the figures and added two paragraphs at the end of sub-section 5.2. to clarify why the Forni glacier is an important site to monitor and what features can be mapped.

We have provided the answers (marked as AC, in plain text) to your comments (marked as EC, in italic), below. Changes to the text are in bold black.

We hope that the revised manuscript can meet your expectations and be considered for publication. Otherwise, we are open to new improvements.

Sincerely,

Davide Fugazza & coauthors

**Editor comments**

*Comments to the Author:*

*Dear author,*

*Your paper has been further revised by one reviewer, the one who raised major review, and by myself. The reviewer proposed at this stage minor, however highlighting the fact that the paper looks like a case study. I deeply read the paper again, and according to the goals of the Special Issue (where you submitted the work) on "The use of remotely piloted aircraft systems (RPAS) in monitoring applications and management of natural hazards", i think that this research, even if case study, is welcome. It is absolutely in line with the special issue purpose.*

*Summarizing: you provided a suitable reply to all the comments (major and minor) raised at the previous review stage. From my side, I think that only minor issues remain in work, mainly related to the section 5.2, and the figure quality and clearness. Here the details:*

*EC - section 5.2. Please improve (adding just 3-4 sentences at the end of the session), the discussion about hazards, and implications of this study. A reader should understand why it is so important to survey such environment and which kind of hazards can be mapped and monitored.*

AC: We have modified subsection 5.2. throughout to enhance its clarity. We have added a sentence at the start to clarify which structures can be monitored. The start of the first paragraph now reads:

"**The tongue of Forni glacier hosts several hazardous structures, including crevasses, normal faults and ring faults. In this study, we focused on the latter two due to their relationship with glacier downwasting.**"

Based on your suggestion, we have also added two paragraphs at the end of the subsection to explain why this environment is a source of hazards and why it should be continuously monitored, i.e. to understand the development of future collapses and avoid risks for hikers. In fact, more collapses have already been documented in summer 2017. The paragraph reads:

"**The presence of cavities at the terminus, which is easily reached in a 45 minutes' walk from Branca hut, is particularly hazardous for tourists, because of 1) the danger of cavity collapse and 2) the potential fall of large boulders or blocks of ice from the ice cliffs. Other fracture systems located higher up glacier are presumably reached by more experienced hikers. Most ring faults however show evidence of vertical and horizontal expansion, and further cavity collapse would imply a severe risk of injury and death if hikers were involved.**

**The location of structures in 2016 suggests that the glacier terminus will recede through ice cliff backwasting and cavity collapse along the fault system on the eastern medial moraine and along the ring faults at the eastern and western margins, potentially compromising access to the glacier. Since this system of fractures has been developing very rapidly, and new collapses have already been documented in September 2017, the risk for people walking on the glacier tongue during summer should be carefully evaluated. Although surface features may be visually detected, the availability of detailed 3D models that depict the entire outer surface of the glacier is a great advantage because it allows quickly capturing the glacier topography remotely, helping predict the possible development of new collapses and understand their mechanisms of formation.**"

*EC: - "Meters" in the maps should be "m".*

AC: We have replaced "meters" with "m" in all maps and uniformed font style of text in all figures (now Arial).

*EC - Fig. 5. It should be improved in the clearness and font size of the numbers/letters. I suggest to reset the figure and set up it in a new layout (you can use any GIS), where the legend and scale should be one, background white, and number/letter font sizes larger.*

AC: We have modified the figure as suggested.

*EC - Fig. 6. I printed it; it is very dark, very difficult to read. I think you can play with the colour options and light as you wish in GIS. The scale should be one. Please avoid to insert letter in box, just simply add it as (a)(b)(c)*

AC: we have modified the figure as requested by using one single scale. In addition, we now avoid using square boxes enclosing the letters in all figures. The letters are now enclosed in brackets "()" when they represent different panels of the figure, and always placed on the top left corner. In case of a non-white background, we have further added a semi-transparent halo around the text to enhance readability against the background.

*EC- Fig. 7. Here please add just one legend and one scale, with glaciers outlines in two colours.*

AC: we have modified the figure as suggested.

*EC- Fig. 8. This can be improved. Legend, scale and north arrow should be outside the figure. Otherwise, it is difficult to feel about the background. Make clearer the glaciers outlines: make the width size larger, try to use different colours (blue with blue on the ground is difficult to read)*

AC: Thanks for the suggestion. We have increased the width of glacier outlines and changed their colours to enhance readability. We have also placed legend, scale and north arrow outside the figure.

[revised manuscript text omitted]

Point Density - Samp

Terrestrial Laser S

Terrestrial Photogra

UAV Photogramm

Auto, Line spacing:  Double

[Figure]

*Figure 6: Spatial coverage of UAV- and terrestrial photogrammetry point clouds and merged point*

*cloud from the two techniques. a) UAV photogrammetry point cloud; b) terrestrial photogrammetry*

*point cloud; c) merged point cloud.*

Point Density -

Terrestrial Laser

Terrestrial Photog

UAV Photogra

*Figure 5 (alternative):  Maps of point density in sample
location 2.¶*

[Figure]

[Figure]

*Figure 7: Location of collapse structures, i.e. normal faults and ring faults and trails crossing the*

*Forni Glacier. (a) Collapse structures in 2014, with 2014 UAV ortophoto as basemap. The red box*

*marks the area surveyed in 2016. (b) Collapse structures in 2016, with 2016 UAV orthophoto as*

*basemap. Trails from Kompass online cartography at https://www.kompass-1039 italia.it/info/mappa-*

*online/.*

[Figure]

*Figure 8: Ice thickness change rates from DEM differencing over (a) 2007-2014; (b) 2007-2016; (c)*

*2014-2016. Glacier outlines from 2014 and 2016 are limited to the area surveyed during the UAV*

*campaigns. Base map from hillshading of 2007 DEM.*